# Preconditioned DeltaNet: Curvature-aware Sequence Modeling for Linear Recurrences

**Neehal Tumma** [* 1 2]  **Noel Loo** [* 1 2]  **Daniela Rus** [1 2]

## Abstract

To address the increasing long-context compute limitations of softmax attention, several sub-quadratic recurrent operators have been developed. This work includes models such as Mamba-2, DeltaNet, Gated DeltaNet (GDN), and Kimi Delta Attention (KDA). As the space of recurrences grows, a parallel line of work has arisen to taxonomize them. One compelling view is the test-time regression (TTR) framework, which interprets recurrences as performing online least squares updates that learn a linear map from the keys to values. Existing delta-rule recurrences can be seen as first-order approximations to this objective, but notably ignore the *curvature* of the least-squares loss during optimization. In this work, we address this by introducing *preconditioning* to these recurrences. Starting from the theory of online least squares, we derive equivalences between linear attention and the delta rule in the exactly preconditioned case. Next, we realize this theory in practice by proposing a diagonal approximation: this enables us to introduce preconditioned variants of DeltaNet, GDN, and KDA alongside efficient chunkwise parallel algorithms for computing them. Empirically, we find that our preconditioned delta-rule recurrences yield consistent performance improvements across synthetic recall benchmarks and language modeling at the 340M and 1B scale. Code can be found here: https://github.com/ntumm120/preconditioned-deltanet.

## 1. Introduction

Scaling LLMs to longer context lengths is increasingly limited by the quadratic compute requirements of softmax attention (Vaswani et al., 2017; Amini et al., 2025). This has driven a rapid evolution of subquadratic recurrent operators, many of which fall under the umbrella of linear attention (Katharopoulos et al., 2020). One such example is Mamba-2 (Dao & Gu, 2024) which can be expressed as follows: $\mathbf{S}_t = \alpha_t \mathbf{S}_{t-1} + \boldsymbol{v}_t \boldsymbol{k}_t^\top$ where $\alpha_t$ represents a scalar between 0 and 1. A drawback of this functional form is that it does not model the varying importance of different key-value associations: if the model needs to forget specific key-value pairs in its associative map $\mathbf{S}_t$, all such pairs are equally decayed. To address this, DeltaNet (Yang et al., 2024) models the recurrence using the *delta rule* (Schlag et al., 2021): $\mathbf{S}_t = \mathbf{S}_{t-1}(\mathbf{I} - \beta_t \boldsymbol{k}_t \boldsymbol{k}_t^\top) + \beta_t \boldsymbol{v}_t \boldsymbol{k}_t^\top$. Yang et al. (2024) showed that delta-style updates improve performance while also admitting an efficient chunkwise parallel form. Subsequent work in Gated Deltanet (Yang et al., 2025), Kimi Delta Attention (Team et al., 2025), Comba (Hu et al., 2025), and RWKV-7 (Peng et al., 2025a) refined the DeltaNet recurrence with a decay factor $\boldsymbol{\alpha}_t$, resulting in significant improvements on recall-driven language tasks.

Concurrently, a more theoretical line of work characterizes these linear-time sequence models using the test-time regression (TTR) framework (Wang et al., 2025; Liu et al., 2024): the recurrent state implements an online map from keys to values that approximately minimizes an in-context least-squares objective given by $||\mathbf{S}\boldsymbol{k}_t - \boldsymbol{v}_t||_2^2$. Notably, these delta-style models (e.g. DeltaNet, GDN, KDA, RWKV-7, Comba) can be interpreted as performing online gradient descent steps on the least squares objective, acting as first-order approximates to a convex, second-order optimization problem. From this perspective, the existing delta-rule recurrences do not account for the curvature that governs the regression geometry, and thus can be viewed as coarse approximations to the underlying least-squares solution. In contrast, Mesalayer (von Oswald et al., 2024) and its more efficient form MesaNet (von Oswald et al., 2025) attempt to solve least squares exactly. The former does so via the direct inverse space recurrence, which suffers from a lack of parallelizability. The latter does so using conjugate gradient steps, which are prone to numerical instability. In this work, we study linear recurrences through the TTR lens and introduce *approximate preconditioning* to their updates, allowing us to retain both efficiency and stability, while benefiting

*Equal contribution [1]MIT [2]Liquid AI. Correspondence to: Neehal Tumma <ntumma@mit.edu>.

*Proceedings of the 43rd International Conference on Machine Learning*, Seoul, South Korea. PMLR 306, 2026. Copyright 2026 by the author(s).

from second-order information.

Our main technical contributions can be summarized as follows: **(i)** We derive an equivalence between linear attention and DeltaNet in the *exactly preconditioned* case. **(ii)** We leverage this insight to uncover efficient chunkwise parallel forms for **Preconditioned Linear Attention** (PLA) and **Preconditioned DeltaNet** (PDN) using an *approximate, diagonal preconditioner*. **(iii)** We efficiently implement and train PDN (and its decayed variants), finding that they improve performance on synthetic tasks and language modeling at the 340M/1B scale.

## 2. Background and preliminaries

**Notation.** We index tokens by $t \in \{1, \dots, T\}$ and chunk the sequence into contiguous blocks of length $C$ (so chunk $i$ contains tokens $t \in \{iC+1, \dots, (i+1)C\}$, assuming $C \mid T$ for simplicity). For any per-token quantity $\boldsymbol{x}_t$, we denote its chunked matrix by $\mathbf{X}_{[i]} \in \mathbb{R}^{C \times d}$ whose rows are the vectors $\boldsymbol{x}_t^\top$ within chunk $i$ (with $d$ being the feature dimension of $\boldsymbol{x}_t$). In particular, $\mathbf{Q}_{[i]}, \mathbf{K}_{[i]} \in \mathbb{R}^{C \times d_k}$ and $\mathbf{V}_{[i]}, \mathbf{O}_{[i]} \in \mathbb{R}^{C \times d_v}$ stack the per-token queries, keys, values, and outputs in chunk $i$, respectively. We also use $\mathbf{S}_{[i]} := \mathbf{S}_{iC}$ to denote the recurrent state at the end of chunk $i$.

### 2.1. Chunkwise parallel linear attention

Linear attention maintains a running key–value summary $\mathbf{S}_t \in \mathbb{R}^{d_v \times d_k}$ via $\mathbf{S}_t = \mathbf{S}_{t-1} + \boldsymbol{v}_t \boldsymbol{k}_t^\top$ and outputs $\boldsymbol{o}_t = \mathbf{S}_t \boldsymbol{q}_t$. As written, this computation is sequential over $t$, which limits training throughput. For efficient training, Yang et al. (2023) derive an exact chunkwise parallel form which allows for an interpolation between the recurrent and quadratic forms of the model. In particular, we can split the sequence into chunks of length $C$. Then the chunkwise parallel form is given by

$$\mathbf{S}_{[i+1]} = \mathbf{S}_{[i]} + \mathbf{V}_{[i]}^\top \mathbf{K}_{[i]},$$
$$\mathbf{O}_{[i]} = \underbrace{\mathbf{Q}_{[i]} \mathbf{S}_{[i]}^\top}_{\text{inter-chunk output}} + \underbrace{\left( \left( \mathbf{Q}_{[i]} \mathbf{K}_{[i]}^\top \right) \odot \mathbf{M} \right) \mathbf{V}_{[i]}}_{\text{intra-chunk output}}.$$

where $\mathbf{M} \in \{0, 1\}^{C \times C}$ is a causal mask. Note that the intra-chunk portion of the output can be computed in parallel across all chunks. Furthermore, the intra-chunk computations are matrix multiplications, which trigger the tensor cores on GPUs, improving wall-clock time despite performing more FLOPs.

### 2.2. Chunkwise parallel delta-rule

DeltaNet replaces the write term $\boldsymbol{v}_t \boldsymbol{k}_t^\top$ in linear attention with the delta rule update (Yang et al., 2024) $(\boldsymbol{v}_t - \mathbf{S}_{t-1} \boldsymbol{k}_t) \boldsymbol{k}_t^\top$. As in linear attention, a chunkwise parallel form can be derived for DeltaNet, enabling efficient training

(Yang et al., 2024). Using the same notation as above:

$$\mathbf{S}_{[i+1]} = \mathbf{S}_{[i]} + \left( \mathbf{U}_{[i]} - \mathbf{W}_{[i]} \mathbf{S}_{[i]}^\top \right)^\top \mathbf{K}_{[i]},$$
$$\mathbf{O}_{[i]} = \mathbf{Q}_{[i]} \mathbf{S}_{[i]}^\top + \left( \mathbf{Q}_{[i]} \mathbf{K}_{[i]}^\top \odot \mathbf{M} \right) \left( \mathbf{U}_{[i]} - \mathbf{W}_{[i]} \mathbf{S}_{[i]}^\top \right),$$
$$\mathbf{T}_{[i]} = \left( \mathbf{I} + \mathrm{tril} \left( \mathrm{diag}(\boldsymbol{\beta}_{[i]}) \mathbf{K}_{[i]} \mathbf{K}_{[i]}^\top, -1 \right) \right)^{-1} \mathrm{diag}(\boldsymbol{\beta}_{[i]}),$$
$$\mathbf{W}_{[i]} = \mathbf{T}_{[i]} \mathbf{K}_{[i]}, \qquad \mathbf{U}_{[i]} = \mathbf{T}_{[i]} \mathbf{V}_{[i]}.$$

Here $\mathrm{diag}(\cdot)$ maps a length-$C$ vector to a $C \times C$ diagonal matrix, and $\mathrm{tril}(\cdot, -1)$ keeps the strict lower-triangular part. To compute $\mathbf{U}_{[i]}, \mathbf{W}_{[i]}$ efficiently, Yang et al. (2024) use the UT transform. We refer the reader to the DeltaNet paper for the intuition behind this derivation.

### 2.3. Test-time regression

Test-time regression is a framework for studying sequence modeling layers and views sequence mixers as performing regression inside their forward passes (Wang et al., 2025). Given a prefix of key–value pairs $\{(\boldsymbol{k}_i, \boldsymbol{v}_i)\}_{i=1}^t$ and a query $\boldsymbol{q}_t$, the layer constructs a memory map $m_t$ by minimizing a regression objective over the prefix, and then applies the resulting map to the query to produce an output. TTR organizes this design space using three axes: (i) how associations are weighted, (ii) the function class used for the memory map, and (iii) the optimization algorithm used as a solver at test time (i.e. the *solve* axis).

**The solve axis.** To isolate (iii), for simplicity, we consider an unweighted objective and restrict the memory map to be linear, $m(\boldsymbol{k}) = \mathbf{S}\boldsymbol{k}$ with $\mathbf{S} \in \mathbb{R}^{d_v \times d_k}$. Weights and nonlinear memory maps can be incorporated into the TTR framework with modifications, but we defer this discussion to Appendix B. Under these assumptions, the TTR objective reduces to the following least-squares problem:

$$\mathbf{S}_t \in \arg \min_{\mathbf{S} \in \mathbb{R}^{d_v \times d_k}} \frac{1}{2} \sum_{i=1}^t \|\mathbf{S}\boldsymbol{k}_i - \boldsymbol{v}_i\|_2^2 + \frac{\lambda}{2} \|\mathbf{S}\|_F^2, \quad (1)$$

with layer output given by $\boldsymbol{o}_t = \mathbf{S}_t \boldsymbol{q}_t$. If we let $\mathbf{K}_t \in \mathbb{R}^{t \times d_k}$ and $\mathbf{V}_t \in \mathbb{R}^{t \times d_v}$ denote the matrices formed by stacking keys and values as rows, then the closed-form solution at time step $t$ is characterized by the normal equations

$$\mathbf{S}_t \underbrace{\left( \mathbf{K}_t^\top \mathbf{K}_t + \lambda \mathbf{I} \right)}_{\mathbf{G}_t} = \underbrace{\mathbf{V}_t^\top \mathbf{K}_t}_{\mathbf{C}_t}. \quad (2)$$

where $\mathbf{G}_t \in \mathbb{R}^{d_k \times d_k}$ is the key Gram matrix and $\lambda \mathbf{I}$ denotes the ridge term. There are two common approaches to solving Eq. (1) at test time. One can compute an *offline*, batch solution by treating $\mathbf{C}_t$ as an approximation of $\mathbf{S}_t$ in Eq. (2). Alternatively, one can maintain an *online* solution by approximately updating $\mathbf{S}_t$ sequentially as new pairs arrive via online gradient descent. The choice of an offline versus online solve yields the recurrences for linear attention and DeltaNet, respectively. Specifically, DeltaNet can be

interpreted as performing an approximate online solve to the least squares solution by doing a single SGD step on the per-token squared loss $L_t(\mathbf{S}) = \frac{1}{2}\|\mathbf{S}\boldsymbol{k}_t - \boldsymbol{v}_t\|_2^2$ with step size given by $\beta_t$:

$$\mathbf{S}_t = \mathbf{S}_{t-1} - \beta_t \nabla_{\mathbf{S}} L_t(\mathbf{S}_{t-1})$$
$$= \mathbf{S}_{t-1} - \beta_t(\mathbf{S}_{t-1}\boldsymbol{k}_t - \boldsymbol{v}_t)\boldsymbol{k}_t^\top.$$

Crucially, both choices only *approximately* solve Eq. (2). In section 3, we will show that by preconditioning these solves using the key Gram matrix $\mathbf{G}_t$, linear attention and DeltaNet both recover the exact solution to Eq. (2).

## 3. Preconditioning linear recurrences

### 3.1. Unifying linear attention and DeltaNet with preconditioning

As discussed in Section 2.3, linear attention and DeltaNet can be viewed as approximate solvers for the least squares problem with loss $\sum_{i=1}^{t}\|\mathbf{S}\boldsymbol{k}_i - \boldsymbol{v}_i\|_2^2$. Here, we show that each recurrence admits a simple modification that recovers the *exact* least-squares solution. This reveals an instance where linear attention and DeltaNet are two realizations of the same operator. Define the key Gram and cross-covariance

$$\mathbf{G}_t := \sum_{i=1}^{t} \boldsymbol{k}_i \boldsymbol{k}_i^\top + \lambda \mathbf{I}, \qquad \mathbf{C}_t := \sum_{i=1}^{t} \boldsymbol{v}_i \boldsymbol{k}_i^\top, \qquad \mathbf{P}_t := \mathbf{G}_t^{-1},$$

and let $\mathbf{S}_t^\star := \mathbf{C}_t \mathbf{P}_t$ denote the least squares map derived in Equation 2.

**Exact solve via linear attention updates.** Linear attention maintains $\mathbf{C}_t$ via the recurrence $\mathbf{C}_t = \mathbf{C}_{t-1} + \boldsymbol{v}_t \boldsymbol{k}_t^\top$. The exact least squares readout is obtained by applying the inverse Gram to the query:

$$\boldsymbol{o}_t = \underbrace{(\mathbf{C}_t\mathbf{P}_t)}_{\text{precondition } \mathbf{C}_t} \boldsymbol{q}_t = \mathbf{C}_t \underbrace{(\mathbf{P}_t\boldsymbol{q}_t)}_{\text{"apply-to-query" (}\mathbf{ATQ}\text{)}} = \mathbf{C}_t\, \tilde{\boldsymbol{q}}_t \tag{3}$$

This aligns with the observation in TTR that linear attention corresponds to an offline solve, in which the Gram is effectively treated as $\mathbf{I}$: applying the inverse key Gram recovers the normal equations. Another interpretation of the offline solve is a single step of full batch gradient descent (BGD). In particular, we can consider linear attention as doing a single step of offline BGD with no preconditioner; in contrast, we can interpret the exact solve shown in Eq. (3) as doing a single step of offline BGD with a perfect preconditioner (i.e. the inverse Gram) on the key dimension of the state (Wang et al., 2025).

In practice, we *realize* the exact solution by applying the inverse Gram transform via pre-multiplication on the query (as opposed to post-multiplication on the state). We will refer to the application of an arbitrary preconditioner to the query vector by $\mathbf{ATQ}$ (apply-to-query) as shown in Eq. (3). An instance of the ATQ formulation is shown in MesaNet

(von Oswald et al., 2025), which applies an iterative transform on the queries to solve the least squares objective via conjugate gradients.

**Exact solve via DeltaNet updates.** DeltaNet performs an online update of the form (Yang et al., 2024)

$$\mathbf{S}_t = \mathbf{S}_{t-1} - \beta_t(\mathbf{S}_{t-1}\underbrace{\boldsymbol{k}_t}_{\text{"read key"}} - \boldsymbol{v}_t)\underbrace{\boldsymbol{k}_t^\top}_{\text{"write key"}}, \qquad \boldsymbol{o}_t = \mathbf{S}_t\boldsymbol{q}_t.$$
$$\tag{4}$$

which, as mentioned in Section 2.2, can be interpreted as a single SGD step on the per-token squared loss $L_t(\mathbf{S}) = \frac{1}{2}\|\mathbf{S}\boldsymbol{k}_t - \boldsymbol{v}_t\|_2^2$. We now show that DeltaNet also admits an exact least squares realization when the key Gram inverse is incorporated into the recurrence. Define

$$\mathbf{P}_{t-1}^{\text{norm}} := \frac{\mathbf{P}_{t-1}}{1 + \boldsymbol{k}_t^\top \mathbf{P}_{t-1}\boldsymbol{k}_t}, \qquad \tilde{\boldsymbol{k}}_t := \mathbf{P}_{t-1}^{\text{norm}}\boldsymbol{k}_t$$

Then the modified DeltaNet update can be written as a preconditioned SGD step:

$$\mathbf{S}_t = \mathbf{S}_{t-1} - \left(\underbrace{(\mathbf{S}_{t-1}\boldsymbol{k}_t - \boldsymbol{v}_t)\boldsymbol{k}_t^\top}_{\nabla_{\mathbf{S}}L_t(\mathbf{S}_{t-1})}\right)\mathbf{P}_{t-1}^{\text{norm}}$$
$$= \mathbf{S}_{t-1} + (\boldsymbol{v}_t - \mathbf{S}_{t-1}\boldsymbol{k}_t)\underbrace{\left(\mathbf{P}_{t-1}^{\text{norm}}\boldsymbol{k}_t\right)^\top}_{\text{"apply-to-key"}(\mathbf{ATK})} \tag{5}$$
$$= \mathbf{S}_{t-1} + (\boldsymbol{v}_t - \mathbf{S}_{t-1}\boldsymbol{k}_t)\tilde{\boldsymbol{k}}_t^\top, \qquad \boldsymbol{o}_t = \mathbf{S}_t\boldsymbol{q}_t.$$

We claim that this preconditioned DeltaNet recurrence recovers the exact least squares solution (see Theorem 3.1). In practice, we realize this by introducing a second key, the write key, $\tilde{\boldsymbol{k}}_t$, in addition to the standard read key, $\boldsymbol{k}_t$. This differs from the standard case in DeltaNet, where both keys are the same. We refer to the application of an arbitrary preconditioner to the write key as $\mathbf{ATK}$ (apply-to-key).

We formalize this equivalence between preconditioned linear attention (**PLA**) and preconditioned DeltaNet (**PDN**) in the case where the preconditioner is the inverse key Gram as follows:

**Theorem 3.1.** *Let $\mathbf{C}_t, \mathbf{G}_t, \mathbf{P}_t$ be defined as above and initialize $\mathbf{S}_0 = \mathbf{0}$. Then the PDN recursion Eq. (5) satisfies $\mathbf{S}_t = \mathbf{C}_t\mathbf{P}_t$ for all $t$. Consequently, for all queries $\boldsymbol{q}_t$, the outputs match:*

$$\boldsymbol{o}_t^{\text{PDN}} = \mathbf{S}_t\boldsymbol{q}_t = (\mathbf{C}_t\mathbf{P}_t)\boldsymbol{q}_t = \mathbf{C}_t(\mathbf{P}_t\boldsymbol{q}_t) = \boldsymbol{o}_t^{\text{PLA}}.$$

The proof (see Appendix C.1) follows from Sherman–Morrison applied to $\mathbf{P}_t = (\mathbf{G}_{t-1} + \boldsymbol{k}_t\boldsymbol{k}_t^\top)^{-1}$.

### 3.2. Preconditioned chunkwise parallel forms

Recall that linear attention and DeltaNet admit chunkwise parallel forms that enable efficient training. Suppose we have access to the per-token within-chunk preconditioner $\mathbf{P}_{[i]}$; then we can construct a chunkwise parallel form for

PLA as follows. For a complete derivation, refer to Appendix F.

$$\mathbf{S}_{[i+1]} = \mathbf{S}_{[i]} + \mathbf{V}_{[i]}^\top \mathbf{K}_{[i]},$$

$$\mathbf{O}_{[i]} = \underbrace{\tilde{\mathbf{Q}}_{[i]}\mathbf{S}_{[i]}^\top}_{\text{inter-chunk output}} + \underbrace{\left(\left(\tilde{\mathbf{Q}}_{[i]}\mathbf{K}_{[i]}^\top\right) \odot \mathbf{M}\right)\mathbf{V}_{[i]}}_{\text{intra-chunk output}}.$$

Here we leverage the ATQ realization described earlier to transform $\mathbf{Q}_{[i]} \to \mathbf{Q}_{[i]}\mathbf{P}_{[i]}^\top = \tilde{\mathbf{Q}}_{[i]}$.

Analogously, for PDN, we can use the ATK realization on the write key to construct the following chunkwise parallel form. Again, for a complete derivation, refer to Appendix F.

$$\mathbf{S}_{[i+1]} = \mathbf{S}_{[i]} + \left(\mathbf{U}_{[i]} - \mathbf{W}_{[i]}\mathbf{S}_{[i]}^\top\right)^\top \tilde{\mathbf{K}}_{[i]},$$

$$\mathbf{O}_{[i]} = \mathbf{Q}_{[i]}\mathbf{S}_{[i]}^\top + \left(\mathbf{Q}_{[i]}\tilde{\mathbf{K}}_{[i]}^\top \odot \mathbf{M}\right)\left(\mathbf{U}_{[i]} - \mathbf{W}_{[i]}\mathbf{S}_{[i]}^\top\right),$$

$$\mathbf{T}_{[i]} = \left(\mathbf{I} + \text{tril}\left(\text{diag}(\boldsymbol{\beta}_{[i]})\,\mathbf{K}_{[i]}\tilde{\mathbf{K}}_{[i]}^\top, -1\right)\right)^{-1}\text{diag}(\boldsymbol{\beta}_{[i]}),$$

$$\mathbf{W}_{[i]} = \mathbf{T}_{[i]}\mathbf{K}_{[i]}, \qquad \mathbf{U}_{[i]} = \mathbf{T}_{[i]}\mathbf{V}_{[i]}.$$

$$\tag{6}$$

So, if we let $\mathbf{P}_{[i]} = (\mathbf{G}_{[i]})^{-1}$ denote the per-chunk inverse Gram (or normalized per-chunk inverse Gram for PDN), then PLA/PDN recover the exact least squares map, provided $\mathbf{P}_{[i]}$ is available. Unfortunately, this requires computing $\mathbf{P}_{[i]}$ without introducing a sequential dependence on all previous time steps. Exact inverse Gram updates follow the Sherman-Morrison recursion which depends on $\mathbf{P}_{t-1}$ in a way that cannot be summarized into a chunk-local affine transition, precluding a chunkwise parallel form from being constructed (see Appendix C.4 for more details). Thus, Mesalayer (von Oswald et al., 2024), which employs the naive, sequential form of the recurrence, is quite inefficient.

One way to address this is shown in MesaNet (von Oswald et al., 2025) which iterates upon Mesalayer by using conjugate gradients (CG) to iteratively approximate the key Gram. This avoids transforming the problem into inverse space, so the recurrence remains linear, enabling a chunkwise parallel form. The drawback of this approach is that it requires several steps of CG, which is prone to numerical instabilities during training (see Appendix E.5 for further discussion).

### 3.3. Approximating the key Gram

As discussed above, solving for the least squares solution exactly in Mesalayer and MesaNet presents either speed or stability issues. On the other end of the spectrum, we have linear attention and DeltaNet which entirely ignore the key Gram curvature, resulting in efficient, stable algorithms at the cost of entirely ignoring second-order information. For our approach, we attempt to interpolate between these two extremes and approximate second-order information by considering a diagonal approximation to the key Gram as

follows:

$$\mathbf{A}_t = \mathbf{A}_{t-1} + \boldsymbol{k}_t \odot \boldsymbol{k}_t. \tag{7}$$

where $\mathbf{A}_t \in \mathbb{R}^{d_k}$ stores the coordinate-wise second moments of the keys. This induces the diagonal preconditioner $\mathbf{P}_t = \text{Diag}(\mathbf{A}_t)^{-1}$. Notably, unlike the exact Gram, its diagonal approximation admits a chunkwise parallel form because its state update is an additive prefix-sum (applied elementwise across dimensions) and inversion is elementwise. Define the shorthand

$$\mathbf{K}_{[i]}^{(2)} := \mathbf{K}_{[i]} \odot \mathbf{K}_{[i]}, \qquad \mathbf{M}_< := \mathbf{M}\mathbf{Z},$$

where $\mathbf{M}$ is the causal mask and $\mathbf{Z}$ is the one-step shift (so $\mathbf{M}_<$ is the strict-causal mask). Then we have that

$$\mathbf{A}_{iC} = \mathbf{A}_{(i-1)C} + \mathbf{1}^\top\mathbf{K}_{[i]}^{(2)}, \quad \mathbf{A}_{[i]}^{\text{pre}} = \mathbf{A}_{(i-1)C} + (\mathbf{M}_<\mathbf{K}_{[i]}^{(2)}) + \underbrace{\boldsymbol{\Lambda}}_{\text{ridge}},$$

$$\tilde{\mathbf{K}}_{[i]} = \underbrace{\text{diag}\left(\left(\mathbf{1} + (((\mathbf{A}_{[i]}^{\text{pre}})^{-1} \odot \mathbf{K}_{[i]}^{(2)})\mathbf{1}))^{-1}\right)\right)}_{\text{normalization term}}\left((\mathbf{A}_{[i]}^{\text{pre}})^{-1} \odot \mathbf{K}_{[i]}\right),$$

$$\tilde{\mathbf{Q}}_{[i]} = \left(\mathbf{A}_{(i-1)C} + (\mathbf{M}\mathbf{K}_{[i]}^{(2)}) + \boldsymbol{\Lambda}\right)^{-1} \odot \mathbf{Q}_{[i]}.$$

$$\tag{8}$$

where $\mathbf{A}_{iC}$ are chunk-boundary diagonal states in $\mathbb{R}^{d_k}$. Note that this enables us to construct both the preconditioned keys and preconditioned queries efficiently. And using the ATK and ATQ realizations, we can instantiate diagonal PGDN and diagonal PLA, respectively, by composing the chunkwise parallel form of the diagonal preconditioner recurrence with the chunkwise parallel form of the main recurrence. Importantly, once we approximate the key gram, Theorem 3.1 no longer holds: namely, the ATK realization of DeltaNet is not equal to the ATQ realization of linear attention (see Appendix C.2 for details). Because of this, not only is the use of a preconditioner a lever in the recurrence design space, but in the case of approximate ones, it also matters where you apply it.

*Figure 1.* In these plots, we analyze the key Gram matrix in an arbitrary layer of a pretrained DeltaNet-340M model from HuggingFace. **Left.** Plot of sorted eigenspectrum in various instances of the key Gram. **Right.** Eigenvalue-weighted average of the $\ell_\infty$ norm of the key Gram eigenvectors. A value of 1 indicates perfect axis-alignment and a value of $\frac{1}{\sqrt{d}}$ indicates perfect misalignment.

To understand how well a diagonal approximation captures the exact key Gram, we analyze a pretrained DeltaNet model (i.e. a linear recurrence without preconditioning) as shown in Figure 1. In particular, we observe two things that lend

support to a diagonal approximation: (i) the eigenspectrum of the key Gram and diagonal key Gram are highly corrrelated and (ii) the eigenvectors of the key Gram are fairly axis-aligned.

From the perspective of optimization, a diagonal approximation is commonly used in optimizers like Adam (Kingma & Ba, 2017), which analogously accumulates only the diagonal of the second-order term. Given the connection between these recurrences and gradient descent, drawing inspiration from preconditioners used in optimization is a natural extension of our work. In our case, a diagonal preconditioner corrects for certain directions in key space being overly amplified or dampened, promoting the learning of a state whose recurrent dynamics are well-conditioned. In Section 5 and Appendix A, we also discuss how preconditioning fits into the online-convex programming view presented in (Liu et al., 2024).

### 3.4. Generalizing to recurrences with decay

So far, we have only considered linear attention and DeltaNet (and their preconditioned variants). Drawing inspiration from prior work that introduces a decay term to these recurrences, (Dao & Gu, 2024; Yang et al., 2025), we analogously add one to ours. Note that linear attention with scalar decay $\alpha_t$ corresponds to Mamba-2 and linear attention with diagonal decay $\boldsymbol{\alpha}_t$ corresponds to GLA. Similarly, DeltaNet with scalar decay corresponds to Gated DeltaNet (GDN), and DeltaNet with diagonal decay corresponds to Kimi Delta Attention (KDA). Fortunately, the preconditioner axis we have introduced into the design space is orthogonal to the decay axis. As a result, we can perform the ATQ transform $\mathbf{Q}_{[i]} \rightarrow \mathbf{Q}_{[i]}\mathbf{P}_{[i]}^\top = \tilde{\mathbf{Q}}_{[i]}$ and substitute this into the chunkwise parallel forms of Mamba-2 and GLA to form preconditioned Mamba-2 and preconditioned GLA, respectively. Analogously, we can perform the ATK transform $\mathbf{K}_{[i]} \rightarrow \mathbf{K}_{[i]}\mathbf{P}_{[i]}^\top = \tilde{\mathbf{K}}_{[i]}$ and substitute this into the chunkwise parallel forms of GDN and KDA to form preconditioned GDN (**PGDN**) and preconditioned KDA (**PKDA**), respectively. See Appendix F for details.

Given that in the case of an approximate preconditioner, linear attention and DeltaNet (as well as their decayed forms) are no longer equivalent, a natural question to ask is which unpreconditioned base recurrence is a better approximation of the exact least squares recurrence. We answer this question with the following result (see Appendix C.5 for proof):

**Theorem 3.2.** *Assume the keys $\{\boldsymbol{k}_t\}$ and queries $\{\boldsymbol{q}_t\}$ are i.i.d. and distributed uniformly on the unit sphere, and that $\{\boldsymbol{q}_t\}$ is independent of $\{\boldsymbol{k}_t\}$. Let $\boldsymbol{o}_t^{\mathrm{LA}}, \boldsymbol{o}_t^{\mathrm{DN}}, \boldsymbol{o}_t^\star$ denote the outputs of linear attention, DeltaNet, and the exact least-squares recurrence at time t, respectively, and assume $\mathbf{S}_0 = \mathbf{0}$. Then there exists $t_0 = t_0(d, \beta)$ such that for all*

*Figure 2.* Distributions of the diagonal Gram preconditioner before (**left**) and after (**right**) squashing where $x = 1.5$ in $\mathbf{B}_t$. Note that the left plot shows distribution in log-spaced buckets, demonstrating the log-normality of $\mathbf{A}_t$, which justifies the log-space parameterization in (9).

$t \geq t_0$,

$$\mathbb{E}\left\|\boldsymbol{o}_t^{\mathrm{DN}} - \boldsymbol{o}_t^\star\right\|_2^2 \leq \mathbb{E}\left\|\boldsymbol{o}_t^{\mathrm{LA}} - \boldsymbol{o}_t^\star\right\|_2^2,$$

Additionally, we find that the ATK transform on the DeltaNet recurrence empirically outperforms the ATQ transform on linear attention (see Appendix E.3). As such, for our empirical study, we will restrict our scope to the PDN, PGDN, and PKDA recurrences, and leave further exploration of the ATQ axis to future work (see Section 5 for additional discussion).

### 3.5. Improving the trainability of the preconditioner

In practice, we find that using the raw form of the diagonal preconditioner recurrence in Eq. (7) is suboptimal from the perspective of trainability. In particular, it has the following drawbacks: **(i)** The empirical distribution of $\mathbf{A}_t$ is log-normal as it represents the accumulation of a positive semi-definite matrix, which leads to heavy right tails as shown in Figure 2. **(ii)** It forces us to align with the Sherman-Morrison form, namely the use of the normalization term from Eq. (8) as well as the previous time-step $\mathbf{A}_{t-1}$ as the preconditioner. **(iii)** It necessitates the explicit formation of the inverse of $\mathbf{A}_t$ to precondition the key, which is prone to numerical instability. To address these issues, we propose the following recurrence for PDN:

$$
\begin{aligned}
\mathbf{A}_t &= \alpha_t^P \mathbf{A}_{t-1} + \beta_t^P (\boldsymbol{k}_t \odot \boldsymbol{k}_t), \\
\boldsymbol{r}_t &= \log(\mathbf{A}_t) - \mu, \qquad \boldsymbol{s}_t = \boldsymbol{r}_t \oslash (\mathbf{1} + |\boldsymbol{r}_t|), \\
\mathbf{B}_t &= \exp\bigl(-\log(x)\boldsymbol{s}_t\bigr), \qquad \tilde{\boldsymbol{k}}_t = \mathbf{B}_t \odot \boldsymbol{k}_t, \\
\mathbf{S}_t &= \mathbf{S}_{t-1} - \beta_t(\mathbf{S}_{t-1}\boldsymbol{k}_t - \boldsymbol{v}_t)\tilde{\boldsymbol{k}}_t^\top, \qquad \boldsymbol{o}_t = \mathbf{S}_t\boldsymbol{q}_t.
\end{aligned}
\tag{9}
$$

where $\oslash$ denotes elementwise devision, $\mu > 0$ is a per-head learnable parameter and $x$ is a fixed hyperparameter. Here, $\mathbf{B}_t$ is our preconditioner, which approximately models the inverse of $\mathbf{A}_t$, but is modified for stability. Note that this parameterization squashes $\mathbf{B}_t$ onto the interval $[\frac{1}{x}, x]$ (as shown in Figure 2) and maintains the per-dimension monotonicity induced by $(\mathbf{A}_t)^{-1}$. As such, to guarantee stability (i.e. recurrent matrix eigenvalues $\leq 1$), it suffices that $x \leq 2$. See Appendix E for details.

The recurrences for PGDN and PKDA are derived by simply

*Figure 3.* Throughput results measuring tokens-per-second during training on 8-H100s using DDP. Models are 340M parameters (D = 1024, H = 8, layers = 24).

replacing $\mathbf{S}_{t-1}$ with $\alpha_t \mathbf{S}_{t-1}$ and $\boldsymbol{\alpha}_t \mathbf{S}_{t-1}$ in Eq. (9), respectively. For additional commentary on the choice of this parameterization as well as ablations on the functional form, refer to Appendix E.3.

# 4. Empirical study

## 4.1. Efficiency analysis

We implement custom Triton kernels for the PGDN and PKDA recurrences by modifying the GDN and KDA kernels from the `flash-linear-attention` (FLA) repository (Yang & Zhang, 2024) using the chunkwise parallel form presented in Eq. (6). Additionally, we write kernels for computing the preconditioner recurrence shown in Eq. (9) and then compose the two for the end-to-end recurrence. For the PDN recurrence, since we do not scale the model up to 1B parameters in this work, we simply reuse the PGDN kernel by fixing the decay term to 1. For implementation details and pseudocode, refer to Appendix D.

**Profiling.** In Figure 3, we profile the training throughput of the models we explore in this work at the 340M parameter scale. We see that the PGDN and PKDA recurrences incur a roughly $10\%$ overhead relative to their GDN and KDA counterparts as a result of the diagonal preconditioner recurrence. This indicates a potential direction for future work where we learn the preconditioned key $\tilde{\boldsymbol{k}}_t$ directly to avoid the time complexity of computing the preconditioner. Of course, the drawback of this approach that it introduces significant parametric overhead; in contrast, the parametric overhead in our recurrence is minimal, coming mostly from small projection layers used to learn $\alpha_t^P, \beta_t^P$. See Section 4.2 for more details. Additionally, we observe that PGDN is more than $20\%$ faster than MesaNet, demonstrating the efficiency advantage of using an approximate preconditioner.

**A new form within the DPLR class.** We additionally note that there exists a kernel in FLA for computing recurrences of the general diagonal-plus-low-rank (DPLR) form, as follows: $\mathbf{S}_t = \mathbf{S}_{t-1}\big(\mathbf{D}_t - \boldsymbol{a}_t \boldsymbol{b}_t^\top\big) + \boldsymbol{v}_t \boldsymbol{k}_t^\top$. Although this functional form supports all the preconditioned delta-rule re-

currences discussed to this point, its generality comes at the cost of speed and memory consumption. We see in Figure 3 that DPLR-GDN, which refers to naively composing our preconditioner recurrence with the DPLR kernel, is significantly slower than our kernel. DPLR-KDA is not pictured as it OOMs with 32K tokens, demonstrating the exorbitant memory requirements of the DPLR kernel (which is used by RWKV-7).

*Figure 4.* Execution time of kernels for varying sequence lengths. Batch size is set to 1, number of heads to 8, and head dimension to 128. Without $\mathbf{P}_t$ refers to the exclusion of the preconditioner recurrence from the computation. $\alpha_t$ and $\boldsymbol{\alpha}_t$ denote scalar and diagonal decay respectively.

Recall our PDN recurrence is given by $\mathbf{S}_t = \mathbf{S}_{t-1} + \beta_t(\boldsymbol{v}_t - \mathbf{S}_{t-1}\boldsymbol{k}_t)\tilde{\boldsymbol{k}}_t^\top$ in which we have a distinct read key and write key. This corresponds to a constrained form of DPLR where $\boldsymbol{a_t}$ is free (in practice we use $\boldsymbol{a_t} = \mathbf{B}_t \boldsymbol{k}_t$, but can be generalized to the $\boldsymbol{a_t}$ free case) and $\boldsymbol{b_t} = \boldsymbol{k_t}$. This acts as an interpolation between the maximally constrained DPLR form used in DeltaNet, GDN, and KDA where $\boldsymbol{a_t} = \boldsymbol{b_t} = \boldsymbol{k_t}$, and the general DPLR form where they are all untied. The nice property of the tying scheme we propose is that having a distinct read and write key does not incur the same overhead that the general DPLR form does. This is shown in Figure 4, which plots the speed of the PGDN and PKDA kernels (with the preconditioner recurrence removed) relative to the speeds of the GDN, KDA, and DPLR kernels. We observe that the overhead is effectively zero, pointing to the scalability of delta-style recurrences with distinct read and write keys. For further discussion, refer to Appendix D.2.

## 4.2. Neural architecture

We follow the same neural architecture presented in (Yang et al., 2024) when instantiating our models. In particular, we preprocess $\boldsymbol{q}, \boldsymbol{k}, \boldsymbol{v}$ with a short convolution to capture token-wise shifts and a SILU activation to approximate the exponential kernel in the softmax attention (Wang et al., 2020). Before the output projection, we employ a headwise RMSNorm. In the KDA and PKDA recurrences only, a data-dependent gating mechanism is also used before the output projection, as this operation is fused into its kernel in `flash-linear-attention`.

*Table 1.* Zero-shot performance of 340M and 1B models trained on the SlimPajama dataset. Colors correspond to the better-performing model within (DN, PDN), (GDN, PGDN), and (KDA, PKDA) pairs. All tasks are evaluated using the `lm-evaluation-harness`. The in-context retrieval tasks follow (Arora et al., 2024b) with 2K input tokens. We note that the 340M KDA/PKDA models are actually 355M, since the KDA kernel in `flash-linear-attention` forces the use of an output gate. As such, decreasing the head dimension to make the models DN, GDN, and KDA isoparametric reduces the throughput of KDA shown in Figure 3. Thus, we keep the head dimension at 128, which results in a slightly larger KDA/PKDA model. See Appendix E.1 for further discussion and additional results.

| Model | Commonsense Reasoning ↑ | | | | | | | | | | In-context retrieval ↑ | | | | | |
|---|---|---|---|---|---|---|---|---|---|---|---|---|---|---|---|---|
| | LAMB. ppl↓ | Wiki ppl↓ | $\text{ARC}_e$ acc | $\text{ARC}_c$ $\text{acc}_n$ | HellaS $\text{acc}_n$ | Lamb. acc | PIQA acc | WinoG acc | BoolQ acc | SciQ acc | Avg acc | FDA acc | SWDE acc | SQuAD acc | TQA acc | DROP acc | Avg acc |
| *340M params with 15B training tokens and 0.5M batchsize tokens* | | | | | | | | | | | | | | | | | |
| DN | 45.76 | 29.04 | 44.02 | 23.55 | 34.08 | 29.58 | 65.07 | 50.91 | 56.80 | 75.30 | 47.41 | 8.53 | 27.09 | 28.32 | 31.20 | 17.80 | 22.59 |
| **PDN** | 43.05 | 29.05 | 44.70 | 23.52 | 33.83 | 30.48 | 65.14 | 52.41 | 58.50 | 75.10 | **47.96** | 14.16 | 29.17 | 28.89 | 30.60 | 16.40 | **23.84** |
| GDN | 31.39 | 27.08 | 44.49 | 24.32 | 35.96 | 34.50 | 65.83 | 51.30 | 56.30 | 78.20 | 48.86 | 13.61 | 29.43 | 29.83 | 33.80 | 17.20 | 24.77 |
| **PGDN** | 28.43 | 26.86 | 47.81 | 24.21 | 36.19 | 35.55 | 64.83 | 52.97 | 60.10 | 78.00 | **49.96** | 14.25 | 32.24 | 31.76 | 35.20 | 17.40 | **26.17** |
| KDA | 31.37 | 26.18 | 45.45 | 22.70 | 36.06 | 34.04 | 66.00 | 52.25 | 57.00 | 79.50 | 49.12 | 13.88 | 34.11 | 29.69 | 34.80 | 16.90 | 25.88 |
| **PKDA** | 25.33 | 25.98 | 46.97 | 22.95 | 37.01 | 37.22 | 65.68 | 51.46 | 59.10 | 78.70 | **49.89** | 14.25 | 34.65 | 31.39 | 35.50 | 18.00 | **26.76** |
| *1B params with 50B training tokens and 0.5M batchsize tokens* | | | | | | | | | | | | | | | | | |
| GDN | 13.46 | 18.05 | 56.10 | 27.30 | 48.91 | 46.42 | 70.08 | 54.22 | 55.80 | 84.70 | 55.44 | 19.69 | 49.86 | 37.70 | 41.60 | 20.50 | 33.87 |
| **PGDN** | 13.09 | 18.01 | 56.66 | 26.96 | 49.66 | 46.49 | 70.73 | 54.54 | 59.90 | 86.10 | **56.38** | 18.33 | 50.32 | 41.19 | 44.30 | 20.80 | **34.99** |
| KDA | 14.52 | 17.95 | 54.55 | 26.88 | 48.95 | 45.16 | 70.29 | 54.22 | 57.60 | 85.80 | 55.43 | 23.23 | 50.14 | 37.00 | 42.70 | 19.70 | 34.55 |
| **PKDA** | 11.86 | 17.82 | 57.24 | 26.96 | 49.05 | 48.52 | 70.62 | 55.49 | 58.80 | 86.40 | **56.64** | 21.69 | 48.79 | 38.61 | 44.60 | 20.70 | **34.88** |

In practice, we implement our preconditioned recurrences by modifying the DeltaNet, GDN, and KDA models in `flash-linear-attention` by replacing the kernel that invokes the base recurrence with one that invokes our preconditioned recurrence shown in Eq. (9). Outside of this, the only other differences in our preconditioned models are the additional projection layers used to parameterize the input-dependent decay $\alpha_t^P$ and write term $\beta_t^P$ for the preconditioner recurrence. We do not tie these to the decay and write terms from the main recurrence as we find doing so hurts performance (see Appendix E). These projection layers are parameterized the same way as they are in the GDN main recurrence and thus introduce effectively no parametric overhead.

### 4.3. Experimental results

#### 4.3.1. SMALL-SCALE SYNTHETICS

We begin by investigating the performance of our preconditioned recurrences at a small scale on the multi-query associative recall task (MQAR) proposed in (Arora et al., 2024a). Each model comprises two sequence-mixing and two channel-mixing layers and is trained on various instances of the MQAR task, which are constructed by varying either the number of key-value pairs or the sequence length in the synthetic data. For additional details on the setup, refer to Appendix E.1.

In Figure 5, we plot the accuracy of DeltaNet (DN), PDN, GDN, and PGDN along two distinct cuts of the MQAR task space, both as a function of sequence length. In the first, we fix the number of key-value pairs, and in the second, we fix the ratio of the number of key-value pairs to sequence length. We find in both settings that the perfor-

mance of MQAR is either maintained or improved by the preconditioned recurrence.

*Figure 5.* Results on synthetic MQAR task.

#### 4.3.2. LARGE-SCALE LANGUAGE MODELING

We move beyond small-scale synthetics and scale our models up to 340M/1B parameters to demonstrate the efficacy of the preconditioned recurrences on language modeling.

**Setup.** Our experiments are designed to ensure a fair comparison between the base DeltaNet/GDN/KDA recurrences and their preconditioned forms. As such, the PDN/PGDN/P-KDA architectures change only the recurrence used for sequence mixing; the rest of the architectural components remain the same. We follow a similar training setup as prior work (Yang et al., 2024; Hu et al., 2025). In particular, we pretrain our models on the SlimPajama dataset (Soboleva et al., 2023), the 340M for 15B tokens and the 1B for 50B tokens. We employ the AdamW optimizer with a 4e-4 learning rate, cosine schedule, 0.01 weight decay, and 1.0 gradient clipping. We train with a context length of 2048. Furthermore, we use the `flash-linear-attention` and `flame` repositories to ensure reproducibility of our

training runs. All evaluations were performed using the `lm-evaluation-harness`. For additional details on the setup, refer to E.1.

**Commonsense reasoning.** In Table 1, we present the language modeling perplexity and zero-shot accuracy on commonsense reasoning benchmarks for models pretrained with 340M and 1B parameters. We find that preconditioned recurrences consistently outperform their unpreconditioned counterparts. We caution the reader against using these evaluations to make cross-architectural comparisons (e.g. GDN vs KDA) as the architectural configuration employed here is fixed and is adapted from the default GDN configuration in `flame`. In particular, prior work (Team et al., 2025) suggests that KDA outperforms GDN (presumably given proper architectural tuning). Nonetheless, we do find it interesting to note that in our experiments, PGDN is the best-performing model on commonsense reasoning at the 340M scale, suggesting a situational advantage to allocating per-dimension scaling to the write key $\tilde{k}_t$ as opposed to state $\mathbf{S}_t$. We leave further exploration into this axis of parametric allocation to future work.

*Figure 6.* 340M models trained on 2K context length evaluated on S-NIAH benchmark from RULER (Hsieh et al., 2024b).

**In-context retrieval.** Next, in Table 1 we further evaluate our models on real-world recall-intensive tasks used in (Arora et al., 2024b) that are geared towards probing in-context retrieval (ICR). Analogous to the commonsense reasoning results, we find that preconditioning improves performance on ICR across all base recurrences. We observe particularly strong improvements from preconditioning in the DN and GDN models. To further support this observation, we evaluate our models on the Single Needle-In-A-Haystack (S-NIAH) benchmark suite from RULER (Hsieh et al., 2024b), where a key-value pair acts as a needle in the haystack, and the model must recall the value when given the key. Note that this task is analogous to MQAR, but now only one key-value pair is present as opposed to multiple, making it a more targeted measure of long-context retrieval ability. The S-NIAH suite proposes 3 variants of increasing difficulty: S-NIAH-1, S-NIAH-2, and S-NIAH-3. We evaluate on each and find that in all task settings, there is an improvement in performance when using preconditioning (Figure 6).

### 4.3.3. ANALYZING RECURRENT DYNAMICS

To further understand why preconditioning has a positive impact on long-context retrieval abilities, we take a closer look at the eigenvalues of recurrent matrices in GDN and PGDN to understand how the dynamics along the time dimension change in the presence of a preconditioner.

*Figure 7.* Distribution of learned $k_t^T \tilde{k}_t$ which modulates the write eigenvalue in PGDN (averaged over layers).

Note that the GDN recurrence is given by $\alpha_t \mathbf{S}_{t-1} + (v_t - \alpha_t \mathbf{S}_{t-1} k_t) \beta_t k_t^\top = \alpha_t \mathbf{S}_{t-1} (\mathbf{I} - \beta_t k_t k_t^\top) + \beta_t v_t k_t^\top$. Since $\|k_t\| = 1$, the eigenvalues of $\alpha_t (\mathbf{I} - \beta_t k_t k_t^\top)$ are $\alpha_t$ with multiplicity $d_k - 1$ and $\alpha_t(1 - \beta_t)$ with multiplicity 1. In the PGDN recurrence, the eigenvalues of the transition matrix are given by $\alpha_t$ with multiplicity $d_k - 1$ and $\alpha_t(1 - \beta_t k_t^T \tilde{k}_t)$ with multiplicity 1 (see proof in Appendix E.3). We refer to the $\alpha_t(1 - \beta_t k_t^T \tilde{k}_t)$ eigenvalue as the write eigenvalue because in the PGDN recurrence, its value is modulated by the inner product between the read and write keys; in contrast, the GDN recurrence has no such expressivity. In Figure 7, we observe this added expressivity by plotting the distribution of the learned $k_t^T \tilde{k}_t$, showing that it deviates from 1, providing intuition for the improved long-context retrieval capabilities of PGDN discussed in Section 4.3.2.

## 5. The DGPS taxonomy: a new design space for linear recurrences

In this section, we briefly introduce a new taxonomy for designing linear recurrences motivated by a decomposition along the following four axes: decay term $\alpha_t$, preconditioner $\mathbf{P}_t$, gain/write $\beta_t$, and solve (offline vs online). So far, we have discussed two frameworks for thinking about recurrence design: TTR and DPLR. These two views lie on opposite ends of a spectrum: TTR is heavily backed by theory, but does not immediately expose a means of mapping recurrences to efficient implementations. DPLR is the opposite: it maps directly to chunkwise parallel forms but lacks the theory to make well-justified design choices. We now note a third one which attempts to bridge this gap: the online-convex programming (OCP) framework presented in (Liu et al., 2024). The OCP formulation is a complementary view that models recurrences as a solution to an

| Model | Decay | Gain | $\mathbf{P}_t$ (ATQ/ATK) | Solve |
|---|---|---|---|---|
| DeltaNet | – | $\beta_t$ | – | online |
| Gated DeltaNet | $\alpha_t$ | $\beta_t$ | – | online |
| Kimi Delta Attention | $\boldsymbol{\alpha_t}$ | $\beta_t$ | – | online |
| Linear Attention | – | – | – | offline |
| RetNet | $\alpha$ | – | – | offline |
| Mamba-2 | $\alpha_t$ | – | – | offline |
| GLA | $\boldsymbol{\alpha_t}$ | – | – | offline |
| MesaNet | $\alpha_t$ | $\beta_t$ | $\approx \mathbf{G}_t$ (ATQ) | offline |
| Comba$^*$ | $\alpha_t$ | $\beta_t/\alpha_t$ | $\boldsymbol{q}_t - d\,\boldsymbol{k}_t$ (ATQ) | online |
| LongHorn | – | $\beta_t/(1 + \beta_t\,\boldsymbol{k}^\top\boldsymbol{k})$ | – | online |
| NLMS (TTR) | – | $1/\|\boldsymbol{k}_t\|^2$ | – | online |
| EFLA | – | $(1 - e^{-\beta_t\lambda_t})/\lambda_t$ | – | online |
| PDN$^\dagger$ | – | $\beta_t$ | $\mathrm{diag}(\mathbf{G}_t)$ (ATK) | online |
| PGDN$^\dagger$ | $\alpha_t$ | $\beta_t$ | $\mathrm{diag}(\mathbf{G}_t)$ (ATK) | online |
| PKDA$^\dagger$ | $\boldsymbol{\alpha_t}$ | $\beta_t$ | $\mathrm{diag}(\mathbf{G}_t)$ (ATK) | online |
| PLA$^\dagger$ | – | – | $\mathrm{diag}(\mathbf{G}_t)$ (ATQ) | offline |
| P-Mamba-2$^\dagger$ | $\alpha_t$ | – | $\mathrm{diag}(\mathbf{G}_t)$ (ATQ) | offline |
| PGLA$^\dagger$ | $\boldsymbol{\alpha_t}$ | – | $\mathrm{diag}(\mathbf{G}_t)$ (ATQ) | offline |

*Table 2.* The **DGPS** taxonomy as shown by a comparison of decay, gain, preconditioning, and solve choices across various linear recurrences. We define the gain term as the scalar multiplied by the write key in the recurrence. Note that Comba (Hu et al., 2025) actually disentangles the gain term into a separate $\beta_t$ applied to the write key in the write term $\boldsymbol{v}\boldsymbol{k}^T$, and a $\frac{\tilde{\beta}_t}{\alpha_t}$ applied to the write key in the erase term $\boldsymbol{k}\boldsymbol{k}^T$. $\mathbf{G}_t$ denotes the inverse Gram. Models marked with $^\dagger$ denote the preconditioned recurrences we propose in this paper.

online convex program (OCP), where the state is updated by solving a proximal objective at each time step. For example, the Gated DeltaNet state update falls out of the solution to the following OCP: $\mathbf{S}_t \in \arg\min_{\mathbf{S}} \quad \|\mathbf{S} - \alpha_t\mathbf{S}_{t-1}\|_F^2 \ - \ 2\langle\mathbf{S}\boldsymbol{k}_t,\ \beta_t(\boldsymbol{v}_t - \alpha_t\mathbf{S}_{t-1}\boldsymbol{k}_t)\rangle$. Note that this formulation exposes the decay, gain, and solve axes, the latter of which is traversed by changing the data term $\langle\mathbf{S}\boldsymbol{k}_t,\ \beta_t(\boldsymbol{v}_t - \mathbf{S}_{t-1}\boldsymbol{k}_t)\rangle$. In fact, we can augment this formulation with a key-side preconditioner $\mathbf{P}$ by replacing $\|\mathbf{S} - \alpha_t\mathbf{S}_{t-1}\|_F^2$ with $\|\mathbf{S} - \alpha_t\mathbf{S}_{t-1}\|_{\mathbf{P}^{-1}}^2$. This preconditioned-OCP (POCP) framework now encapsulates the PDN, PGDN, and PKDA recurrences we explored in this work (see Appendix G for derivations). Unfortunately, a shortcoming of the POCP taxonomy is that it does not expose the query axis: in fact, it is entirely ignored in the formulation. And as we know from our discussion of the ATQ transform in Section 3.1, not only is query-side preconditioning theoretically-backed, but it also admits an efficient chunkwise parallel form. As such, we propose the **DGPS** (decay, gain, preconditioner, solve) taxonomy shown in Table 2, which parameterizes the design space as follows:

**Online** solve:

$$\mathbf{S}_t = \alpha_t\,\mathbf{S}_{t-1} + \beta_t\big(\boldsymbol{v}_t - \alpha_t\mathbf{S}_{t-1}\boldsymbol{k}_t\big)\tilde{\boldsymbol{k}}_t^\top,\ \boldsymbol{o}_t = \mathbf{S}_t\,\boldsymbol{q}_t,$$

**Offline** solve:

$$\mathbf{S}_t = \alpha_t\,\mathbf{S}_{t-1} + \beta_t\,\boldsymbol{v}_t\,\boldsymbol{k}_t^\top,\ \boldsymbol{o}_t = \mathbf{S}_t\,\tilde{\boldsymbol{q}}_t.$$

where $\alpha_t$ denotes the decay term, $\beta_t$ denotes the gain term and $\tilde{\boldsymbol{k}}_t$ and $\tilde{\boldsymbol{q}}_t$ denote the preconditioned keys and queries respectively. The benefit of DGPS is that the decay, gain, preconditioner, and solve axes are orthogonal levers that not

only immediately relate to the theory derived in TTR but also map clearly onto a chunkwise parallel form, enabling an efficient implementation. In contrast, the OCP framework does not expose the query preconditioner axis, but also conflates the gain and solve axes. In particular, changing the data term (i.e. the solve axis in OCP) in the Gated DeltaNet OCP to $\|\mathbf{S}\boldsymbol{k}_t - \boldsymbol{v}_t\|_2^2$ also changes the gain term from $\beta_t$ to $\frac{\beta_t}{1 + \beta_t\boldsymbol{k}_t^T\boldsymbol{k}_t}$ as shown in LongHorn (Liu et al., 2024). This demonstrates how axes are entangled in the OCP formulation, resulting in redundancies in the OCP design space. For an extended discussion of the DGPS taxonomy and how it relates to the existing space of frameworks for linear recurrence design, refer to Appendix B.

# 6. Conclusion

In this work, we incorporate the notion of preconditioning into the design of linear recurrences. Motivated by theory from online least squares, we derive a case in which linear attention and DeltaNet are equivalent under exact preconditioning. We then show that when considering a diagonal approximation to the exact preconditioner, we can realize efficient chunkwise parallel forms for both preconditioned linear attention and preconditioned DeltaNet. We translate this theory to practice by implementing preconditioned instances of DeltaNet, Gated DeltaNet, and Kimi Delta Attention. Empirically, we find that these preconditioned recurrences improve performance on synthetic recall and language benchmarks, uncovering the preconditioner axis as a lever for constructing novel linear recurrent models.

## Acknowledgements

Support for this work has been provided in part by the Singapore MIT Alliance M3 program and the Schmidt AI2050 program. This research was also sponsored by the Department of the Air Force Artificial Intelligence Accelerator and was accomplished under Cooperative Agreement Number FA8750-19-2-1000. The views and conclusions contained in this document are those of the authors and should not be interpreted as representing the official policies, either expressed or implied, of the Department of the Air Force or the U.S. Government. The U.S. Government is authorized to reproduce and distribute reprints for Government purposes notwithstanding any copyright notation herein.

## Impact Statement

This paper presents work whose goal is to make more efficient and expressive sequence models. While there are many potential societal consequences of our work, we feel none must be specifically highlighted here as this work does not possess any particular societal risk.

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

# A. Related work

## A.1. Linear attention and SSMs.

State space models (SSMs) are sequence models where the recurrent dynamics are governed by linear state space equations of the form $\boldsymbol{x}_t = \mathbf{A}\boldsymbol{x}_{t-1} + \mathbf{B}\boldsymbol{u}_t, \boldsymbol{y}_t = \mathbf{C}\boldsymbol{x}_t$ (Kalman, 1960). Early successes in SSMs emphasized careful parameterization and initialization of recurrent dynamics (Gu et al., 2022a;b). These early methods used non-time-varying $\mathbf{A}, \mathbf{B}, \mathbf{C}$, matrices, allowing them to be solved using the Fast Fourier Transform. Parallel scans can also be used to evaluate these recurrences in sublinear time (Smith et al., 2023; Hasani et al., 2023), and additionally enable time-dependent dynamics, used in time-varying SSM layers like Mamba (Gu & Dao, 2024), which leverages an input-dependent, diagonal decay on a vector-valued state. A parallel line of work iterating on linear attention (Katharopoulos et al., 2020) instead considers models that can be viewed as matrix-valued recurrences, where the state $\mathbf{S}_t$ can be interpreted as associative memory learning a mapping from the keys $\boldsymbol{k}_t$ to the values $\boldsymbol{v}_t$ (Schlag et al., 2021). Later work (Dao & Gu, 2024) shows that linear attention with decay can be considered a special case of SSMs with a matrix-valued state. Provided with sufficient structure on the recurrent matrix, these matrix-valued SSMs can admit an efficient *chunkwise parallel* form, similar to the parallel form of linear attention.

## A.2. Delta-rule sequence models.

DeltaNet, and its decayed variant Gated DeltaNet (Yang et al., 2024; 2025) leverage the delta rule update: $\mathbf{S}_t = \mathbf{S}_{t-1} + \beta_t \boldsymbol{\delta}_t \boldsymbol{k}_t^T$, where $\boldsymbol{\delta}_t$, is the error between the current readout of the key, $\hat{\boldsymbol{v}} = \mathbf{S}_{t-1}\boldsymbol{k}_t$ and the target value $\boldsymbol{v}_t$ given by $\boldsymbol{\delta}_t = \boldsymbol{v}_t - \hat{\boldsymbol{v}}_t$. This essentially lets the model correct the state so that the target value is associated with the key. Another interpretation is given by the effective recurrent matrix, $\mathbf{I} - \beta_t \boldsymbol{k}_t \boldsymbol{k}_t^\top$, which essentially modulates values of the state along the $\boldsymbol{k}_t$ axis. This gives the interpretation of DeltaNet updates as erasing what was previously stored in that state before writing it with the new value. Yang et al. (2024) shows that such recurrent transitions can be efficiently parallelized using the WY representation theorem. Notably, the recurrent matrices, $\mathbf{A}_t = \mathbf{I} - \beta_t \boldsymbol{k}_t \boldsymbol{k}_t^T$, do not commute with one another. A line of work has leveraged this non-commutativity to solve state-tracking tasks (Merrill et al., 2025; Grazzi et al., 2025; Siems et al., 2025). Further work has leveraged the WY representation to parallelize more expressive state transition matrices, extending it to more general forms of the diagonal-plus-low-rank (DPLR) class (Team et al., 2025; Peng et al., 2025a).

## A.3. Test-time training, mesa optimization, and online learning.

Test-time-regression (Wang et al., 2025) is a framework for analyzing recurrent models as approximately learning input-output mappings as the sequence is traversed. It primarily focuses on the setting of L2 regressions, constructing a taxonomy for recurrences that approximately solve this task, where keys and values act as the standard $(x, y)$ input tuples in linear regression, and the recurrent state takes the form of the linear regression weight. This framework frames linear attention as approximately solving this task via one full-batch (i.e. sequence-wide) gradient step (which was also noted in prior work Sun et al. (2024)). DeltaNet (Yang et al., 2024), in contrast, considered an online version of approximating this task, which performs single-step gradient descent on each key-value pair for the linear regression task. On the other hand, the TTT-layer (Sun et al., 2024) performs mini-batched gradient descent on non-linear regression (with an MLP hidden state). The Mesalayer (von Oswald et al., 2024) aims to solve the linear regression task *exactly* by computing the exact linear-regression weight at every time step, which is done in practice by Sherman-Morrison iteration on the key-Gram inverse matrix. Noting the slow sequential nature of Sherman-Morrison, MesaNet (von Oswald et al., 2025) solves the regression problem approximately using several conjugate gradients (CG), which enables a chunkwise parallel form that avoids computing sequential, non-linear recurrences in inverse space (as is done in Mesalayer). The downside of this approach is twofold: firstly, the number of FLOPs required to compute the MesaNet recurrence grows linearly in the number of CG iterations (which is set to 30). Secondly, due to the approximate nature of CG, exact convergence is not guaranteed, which can lead to instability, especially considering the backwards pass, which uses the Implicit Value Theorem to compute gradients, assumes *exact* convergence.

## A.4. Diagonal preconditioning in optimization.

One of the most standard optimization algorithms is gradient descent, which directly takes a step in the direction of the gradient, $\boldsymbol{\theta} \leftarrow \boldsymbol{\theta} - \eta\boldsymbol{g}$, and its stochastic variants, stochastic gradient descent (SGD) (Robbins, 1951). In practice, this performs poorly in highly ill-conditioned problems, where certain axes curve rapidly compared to others. For these problems, techniques such as momentum and preconditioning are used (Nesterov, 1983; Polyak, 1964). Preconditioning modifies the standard gradient step by multiplying the gradient by a preconditioner matrix: $\boldsymbol{\theta} \leftarrow \boldsymbol{\theta} - \eta\mathbf{P}\boldsymbol{g}$. In the case of quadratic

losses, a preconditioned gradient step leads directly to the minima, and in the case of $L_2$ regression with $N$ $(\boldsymbol{x}, \boldsymbol{y})$ pairs, $\mathbf{P}$ is given exactly by $(\sum_{i=1}^N \boldsymbol{x}_i \boldsymbol{x}_i^T)^{-1}$. In the event a full-rank preconditioner is too expensive, it is typical to use diagonal or factorized preconditioners (Kingma & Ba, 2017; Duchi et al., 2011; Martens & Grosse, 2020).

### A.5. Online convex programming and approximate continual learning

As discussed in Section 5, the preconditioning in this method maps to the norm used in the proximal term in the OCP framework, changing the term from $\|\mathbf{S} - \alpha_t \mathbf{S}_{t-1}\|_F^2$ to $\|\mathbf{S} - \alpha_t \mathbf{S}_{t-1}\|_{\mathbf{P}^{-1}}^2$. An alternative view of this term is as a *forgetting* term which constrains the state from deviating too much from its history to prevent forgetting previous key–value pairs. This ties closely to literature on continual learning, which deals with training model parameters over streaming tasks with limited access to former tasks. In the least-squares case, it is known that the exact preconditioning for recovering the least-squares solution is the inverse key Gram matrix: $\mathbf{P}^{-1} = \sum_{i=1}^T \boldsymbol{k}_i \boldsymbol{k}_i^T + \lambda \mathbf{I}$. For non-convex problems, it is common to use the Fisher information matrix, or a diagonal approximation of the Fisher information matrix (Kirkpatrick et al., 2017). Such diagonal-quadratic regularizers are common in continual learning work (Kirkpatrick et al., 2017; Nguyen et al., 2018; Loo et al., 2020). Noting this connection between continual learning and sequence modeling could motivate more effective sequence mixers in the future.

## B. The DGPS taxonomy continued

**Existing frameworks for linear recurrence design.** We begin by surveying other taxonomies for the recurrence design space that are tangentially related to the DGPS taxonomy discussed in Section 5, but are nonetheless useful frameworks for understanding the existing space of recurrent models. The DeltaNet (and originally GLA) paper (Yang et al., 2024) proposes the following functional form for a general class of associative recurrences with matrix-valued hidden states: $\mathbf{S}_t = \mathbf{S}_{t-1} \bullet \mathbf{M}_t + \boldsymbol{v}_t \boldsymbol{k}_t^T$ where $\bullet$ denotes an arbitrary associative operator. In practice, the use of an associative operator enables a parallel scan, (Gu et al., 2022a) which can be used to compute the hidden states $\mathbf{S}_t$ in a logarithmic number of steps. The drawback of this tree-based algorithm is that it has poor locality of reference and does not cast much of the computation as matrix multiplications, making it inefficient on modern GPUs. An example of a recurrence that falls under this taxonomy is Mamba (Gu & Dao, 2024), which uses the Hadamard product $\odot$ to define the associative algebra. Due to the lack of rank-1 structure in the Mamba $\mathbf{M}_t$ gate, it cannot be cast in an efficient chunkwise parallel form as doing so would require a contraction along the state dimension.

To address this, a new framework known as the state-space duality (SSD) was proposed in Mamba-2 (Dao & Gu, 2024) which considers an instantiation of Mamba with scalar decay $\alpha_t$, enabling a chunkwise parallel form as shown in Appendix F. The SSD provides a connection between SSMs and semi-separable structured matrix multiplications, which are efficient to compute. The SSM framework can be generalized to structured masked attention (SMA) which includes all models whose output $\mathbf{O}$ can be written as $\mathbf{P} = \mathbf{A} \odot \mathbf{M}, \mathbf{O} = \mathbf{PV}$ where $\mathbf{A}$ refers to the surrogate attention matrix (Parnichkun et al., 2025) and $\mathbf{M}$ refers to a lower-triangular causal mask. While softmax attention (Vaswani et al., 2017) technically falls under this framework, in practice it refers to models which have structure in $\mathbf{M}$ and can thus realize subquadratic algorithms for computing them. This includes long-convolutions such as Hyena (Poli et al., 2023) and recurrences with growing state size such as Log-Linear Attention (Guo et al., 2025).

**Unifying the TTR, DPLR, OCP and DGPS taxonomies.** We next discuss the frameworks mentioned in the main portion of this paper and elucidate where DGPS stands with respect to them: TTR, DPLR and OCP.

As discussed throughout the main text, the TTR framework provides a unified perspective on sequence layers by framing them as regression problems learning an associative memory map from the keys to the values. In particular, it considers the offline batch gradient descent (BGD), online stochastic gradient descent (SGD) and exact solution views of parametric regression, as well as extensions to nonparametric regression. The TTR framework is quite general, which comes at the cost of encompassing less efficient functional forms. Outside the linear recurrences we have discussed in the main text, it also includes models in the test-time training (Sun et al., 2024) space, such as Titans (Behrouz et al., 2024) and Atlas (Behrouz et al., 2025), which can be interpreted as an online SGD step using a nonlinear memory map. In the test-time training recurrences, this online SGD step (which induces the write term) is non-linear in the previous state, requiring an approximation to parallelize the otherwise non-linear recurrence. As such, we consider it out of the scope of the DGPS taxonomy, which encompasses linear recurrences only. We further note that the TTR framework even includes softmax attention, which can be interpreted as an instance of approximate kernel regression. But with this generality comes a rich basis from regression theory and optimization, which can be used to inform our design choices. As such, in our interpretation,

DGPS can be considered a practical instantiation of the TTR framework geared towards realizing its theory in the form of efficient implementations on modern hardware.

We next briefly discuss the DPLR taxonomy, which posits the form $\mathbf{S}_t = \mathbf{S}_{t-1}\big(\mathbf{D}_t - \boldsymbol{a}_t\boldsymbol{b}_t^\top\big) + \boldsymbol{v}_t\boldsymbol{k}_t^\top$, and how it relates to our proposed DGPS taxonomy. In particular, we note the following: DPGS $\not\subseteq$ DPLR and DPLR $\not\subseteq$ DGPS. The former follows from the lack of the query axis in the DPLR form and the latter follows from the unconstrained tying across $\boldsymbol{a}_t, \boldsymbol{b}_t, \boldsymbol{k}_t$. However, when we restrict ourselves to subspaces of these taxonomies, we observe the following: ATK-DGPS $\subseteq$ DPLR and fast-DPLR $\subseteq$ DGPS. The former refers to models with key-side preconditioning only (e.g. PDN, PGDN, PKDA) within the DGPS framework. The latter refers to models that leverage a constrained tying scheme within the general DPLR form, admitting a more efficient implementation. We defer a more detailed discussion on the algorithmic relationship between the DGPS and DPLR taxonomies to Appendix D.2.

Finally, we discuss the online-convex programming (OCP) framework briefly introduced in Section 5 which was originally proposed in the LongHorn (Liu et al., 2024) paper. The OCP framework proposes a unifying perspective on linear recurrence design via online learning: given a per-token convex loss $\ell_t(\mathbf{S})$ induced by $(\boldsymbol{k}_t, \boldsymbol{v}_t)$, the OCP update takes the form

$$\mathbf{S}_t \in \arg\min_{\mathbf{S}} \ ||\mathbf{S} - \alpha_t\mathbf{S}_{t-1}||_F + \beta_t\,\ell_t(\mathbf{S}),$$

where $\beta_t \geq 0$. This formulation exposes three design axes used throughout prior work (Yang et al., 2025): (i) the decay on $\mathbf{S}_{t-1}$ in the proximal term, (ii) the gain term $\beta_t$ controlling the magnitude of the write key, and (iii) the data term $\ell_t$ (e.g. squared loss or linearized variants), which induces different linear recurrences. The data term is analogous to the solve term in DGPS in that it corresponds to different approximations of the least squares loss. We note how particular forms of the OCP objective yield different recurrences as follows. In particular, the linear attention state update falls out of the solution to the following OCP:

$$\mathbf{S}_t \in \arg\min_{\mathbf{S}} \ \ ||\mathbf{S} - \mathbf{S}_{t-1}||_F^2 \ - \ 2\langle \mathbf{S}\boldsymbol{k}_t, \boldsymbol{v}_t\rangle,$$

Similarly, the DeltaNet update is the solution to

$$\mathbf{S}_t \in \arg\min_{\mathbf{S}} \ \ ||\mathbf{S} - \mathbf{S}_{t-1}||_F^2 \ - \ 2\left\langle \mathbf{S}\boldsymbol{k}_t,\ \beta_t(\boldsymbol{v}_t - \mathbf{S}_{t-1}\boldsymbol{k}_t)\right\rangle.$$

LongHorn uses the OCP framework to design a new recurrence whose OCP is given as follows:

$$\mathbf{S}_t \in \arg\min_{\mathbf{S}} \ \ ||\mathbf{S} - \mathbf{S}_{t-1}||_F^2 \ + \ \beta_t\,||\mathbf{S}\boldsymbol{k}_t - \boldsymbol{v}_t||_2^2.$$

Note that the solution to the LongHorn OCP is given by

$$\mathbf{S}_{t-1} + (\boldsymbol{v}_t - \mathbf{S}_{t-1}\boldsymbol{k}_t)\epsilon_t\boldsymbol{k}_t^\top$$

where $\epsilon_t = \frac{\beta_t}{1+\beta_t\boldsymbol{k}_t^T\boldsymbol{k}_t}$. With this, we observe the following property of the OCP framework: it conflates the gain and axes. In particular, changing the data term between linear attention and DeltaNet changes the functional form of the recurrence from one expressing an offline BGD solve to another expressing an online BGD solve, both of which admit fundamentally distinct chunkwise parallel forms. But then changing the data term again between DeltaNet and LongHorn changes only the gain term $\beta_t$ via a normalization factor that falls out of recursive least squares (as we showed in Appendix C.1). While in theory this is a nice property of the OCP formulation, in practice we find that it is necessary to disentangle the gain axis into a standalone gain term plus an additional transform on this gain term (in the case of LongHorn this additional transform would be normalizing by $1 + \beta_t\boldsymbol{k}_t^T\boldsymbol{k}_t$). We show this empirically in Appendix E.3 where we ablate various normalizing factors on the gain term (including the one proposed in LongHorn) and find that they worsen performance in our preconditioned recurrences.

On the flip side, a nice observation regarding OCP is that it can be trivially augmented with the preconditioner axis from DGPS by changing the proximal term norm from a Frobenius inner product to a general matrix-$\mathbf{P}^{-1}$ inner product, yielding the following preconditioned OCP (POCP):

$$\mathbf{S}_t \in \arg\min_{\mathbf{S}} \ \ ||\mathbf{S} - \mathbf{S}_{t-1}||_{\mathbf{P}^{-1}}^2 \ + \ \beta_t\,||\mathbf{S}\boldsymbol{k}_t - \boldsymbol{v}_t||_2^2.$$

We show in Appendix G that the PDN, PGDN, and PKDA recurrences can be derived from the POCP form. Furthermore, the POCP form has interpretations beyond an approximate preconditioner used for the key Gram in least squares: namely, in this

framework, $\mathbf{P}$ can be interpreted as an arbitrary positive semi-definite transform applied to the proximal space $\mathbf{S} - \mathbf{S}_{t-1}$ that corresponds to a matrix-inner product computed in non-Frobenius space. In particular, this suggests the use of non-identity matrices to modulate forgetting along the proximal axis. Nonetheless, a drawback of even the POCP form is that it, like the DPLR taxonomy, does not expose the query axis for transformation.

**DGPS outside the lens of least squares.** As we have now discussed DGPS and where it stands relative to other unifying perspectives on linear recurrence design, we next attempt to showcase its generality by discussing instances of linear recurrences that do not obviously fall out of the TTR framework, yet reside in DGPS. Note that to this point, we have implicitly tied together the notion of key-side preconditioning (i.e. ATK) with the base DeltaNet recurrence (i.e. online solve) and the notion of query-side preconditioning (i.e. ATQ) with the base linear attention recurrence (i.e. offline solve). But if we abstract away the connection between TTR and DGPS, then DGPS, like the DPLR form, is simply a set of four levers that map to an efficient chunkwise parallel form. In particular, there is nothing that precludes using ATQ in the online solve form and ATK in the offline solve form.

The utility of this abstracted lens is that we can leverage other theoretical bases to design the functional form of the recurrence and then implement it efficiently via the DGPS chunkwise parallel forms. One example of this is the Comba recurrence (Hu et al., 2025), which leverages a control-theoretic framework for deriving the functional form of their model, shown in Table 2. One interesting thing to note with the Comba recurrence is the presence of an output correction factor $d$, which transforms the query via $\boldsymbol{q}_t - d\boldsymbol{k}_t$ where $d$ is a learnable scalar. If we generalize the interpretation of ATQ from an approximate least-squares preconditioner to an arbitrary query-side transform, we see how Comba is an instance of a recurrence that falls into the DGPS space without entirely being derived from TTR. Indeed, Comba leverages the ATQ idea we formalize in this work when deriving the chunkwise parallel form for their recurrence. We can make the same generalization for ATK and consider the DGPS framework as a means of efficiently implementing key-side and query-side transforms in the cases where either linear attention or DeltaNet is the base recurrence. This DGPS abstraction away from TTR is also shown in the EFLA recurrence in Table 2, which derives the gain term using results from a continuous-time ODE perspective on linear recurrences.

Given the strong performance gains observed via the query-side transform in the online solve case in Comba, one promising avenue of future work within the DGPS space is exploring recurrences that perform both key and query-side preconditioning.

## C. Additional theory and discussion

### C.1. Deriving an equivalence between linear attention and DeltaNet under exact preconditioner

Let $\mathbf{G}_t := \sum_{i=1}^{t} \mathbf{k}_i \mathbf{k}_i^\top + \lambda \mathbf{I}$, $\mathbf{P}_t := \mathbf{G}_t^{-1}$ and $\mathbf{C}_t := \sum_{i=1}^{t} \mathbf{v}_i \mathbf{k}_i^\top$. The preconditioned linear attention state at step $t$ is given by $\mathbf{S}_t^{PLA} = \mathbf{C}_t \mathbf{P}_t$. At $t = 0$, $\mathbf{S}_0^{PLA} = \mathbf{S}_0^{PDN} = 0$.

**Theorem C.1.** $\mathbf{S}_t^{PLA} = \mathbf{S}_t^{PDN}$ *for all $t$ under exact preconditioning.*

*Proof.* We show the equivalence via induction. The base case at $t = 0$ is trivial, and both $\mathbf{S}_0^{PLA} = \mathbf{S}_0^{PDN} = 0$. Assume $\mathbf{S}_t^{PLA} = \mathbf{S}_t^{PDN}$.

We have:

$$
\begin{aligned}
\mathbf{S}_{t+1}^{PLA} &= \mathbf{C}_{t+1}\mathbf{P}_{t+1} \\
&= (\mathbf{C}_t + \mathbf{v}_{t+1}\mathbf{k}_{t+1}^\top)(\mathbf{G}_t + \mathbf{k}_{t+1}\mathbf{k}_{t+1}^\top)^{-1} \\
&= (\mathbf{C}_t + \mathbf{v}_{t+1}\mathbf{k}_{t+1}^\top)\left(\mathbf{G}_t^{-1} - \frac{\mathbf{G}_t^{-1}\mathbf{k}_{t+1}\mathbf{k}_{t+1}^\top\mathbf{G}_t^{-1}}{1 + \mathbf{k}_{t+1}^\top\mathbf{G}_t^{-1}\mathbf{k}_{t+1}}\right) \\
&= \mathbf{C}_t\mathbf{G}_t^{-1} - \frac{\mathbf{C}_t\mathbf{G}_t^{-1}\mathbf{k}_{t+1}\mathbf{k}_{t+1}^\top\mathbf{G}_t^{-1}}{1 + \mathbf{k}_{t+1}^\top\mathbf{G}_t^{-1}\mathbf{k}_{t+1}} + \mathbf{v}_{t+1}\mathbf{k}_{t+1}^\top\mathbf{G}_t^{-1} - \frac{\mathbf{v}_{t+1}\mathbf{k}_{t+1}^\top\mathbf{G}_t^{-1}\mathbf{k}_{t+1}\mathbf{k}_{t+1}^\top\mathbf{G}_t^{-1}}{1 + \mathbf{k}_{t+1}^\top\mathbf{G}_t^{-1}\mathbf{k}_{t+1}} \\
&= \mathbf{C}_t\mathbf{G}_t^{-1}\left(\mathbf{I} - \mathbf{k}_{t+1}\left(\frac{\mathbf{G}_t^{-1}\mathbf{k}_{t+1}}{1 + \mathbf{k}_{t+1}^\top\mathbf{G}_t^{-1}\mathbf{k}_{t+1}}\right)^\top\right) + \mathbf{v}_{t+1}\mathbf{k}_{t+1}^\top\mathbf{G}_t^{-1}\left(\mathbf{I} - \frac{\mathbf{k}_{t+1}\mathbf{k}_{t+1}^\top\mathbf{G}_t^{-1}}{1 + \mathbf{k}_{t+1}^\top\mathbf{G}_t^{-1}\mathbf{k}_{t+1}}\right) \\
&= \mathbf{C}_t\mathbf{G}_t^{-1}\left(\mathbf{I} - \mathbf{k}_{t+1}\left(\frac{\mathbf{G}_t^{-1}\mathbf{k}_{t+1}}{1 + \mathbf{k}_{t+1}^\top\mathbf{G}_t^{-1}\mathbf{k}_{t+1}}\right)^\top\right) + \mathbf{v}_{t+1}\mathbf{k}_{t+1}^\top\mathbf{G}_t^{-1}\left(\frac{\mathbf{I} + \mathbf{k}_{t+1}^\top\mathbf{G}_t^{-1}\mathbf{k}_{t+1} - \mathbf{k}_{t+1}\mathbf{k}_{t+1}^\top\mathbf{G}_t^{-1}}{1 + \mathbf{k}_{t+1}^\top\mathbf{G}_t^{-1}\mathbf{k}_{t+1}}\right) \\
&= \mathbf{C}_t\underbrace{\mathbf{G}_t^{-1}}_{\mathbf{P}_t}\left(\mathbf{I} - \mathbf{k}_{t+1}\left(\frac{\mathbf{G}_t^{-1}\mathbf{k}_{t+1}}{1 + \mathbf{k}_{t+1}^\top\mathbf{G}_t^{-1}\mathbf{k}_{t+1}}\right)^\top\right) + \mathbf{v}_{t+1}\mathbf{k}_{t+1}^\top\mathbf{G}_t^{-1}\left(\frac{1}{1 + \mathbf{k}_{t+1}^\top\mathbf{G}_t^{-1}\mathbf{k}_{t+1}}\right) \\
&= \underbrace{\mathbf{C}_t\mathbf{P}_t}_{S_t^{PLA}=S_t^{PDN}}\left(\mathbf{I} - \mathbf{k}_{t+1}\left(\frac{\mathbf{G}_t^{-1}\mathbf{k}_{t+1}}{1 + \mathbf{k}_{t+1}^\top\mathbf{G}_t^{-1}\mathbf{k}_{t+1}}\right)^\top\right) + \mathbf{v}_{t+1}\underbrace{\left(\frac{\mathbf{G}_t^{-1}\mathbf{k}_{t+1}}{1 + \mathbf{k}_{t+1}^\top\mathbf{G}_t^{-1}\mathbf{k}_{t+1}}\right)^\top}_{\tilde{\mathbf{k}}_{t+1}^\top} \\
&= \mathbf{S}_t^{PDN} - \mathbf{S}_t^{PDN}\mathbf{k}_{t+1}\tilde{\mathbf{k}}_{t+1}^\top + \mathbf{v}_{t+1}\tilde{\mathbf{k}}_{t+1}^\top \\
&= \mathbf{S}_t^{PDN} + (\mathbf{v}_{t+1} - \mathbf{S}_t^{PDN}\mathbf{k}_{t+1})\tilde{\mathbf{k}}_{t+1}^\top \\
&= \mathbf{S}_{t+1}^{PDN}
\end{aligned}
$$

Where we expand $\mathbf{G}_{t+1}^{-1}$ via. Sherman-Morrison.

$\square$

We note that we can also write $\tilde{\mathbf{k}}_{t+1}$ as $\mathbf{P}_{t+1}\mathbf{k}_{t+1}$:

$$
\tilde{\mathbf{k}}_{t+1} = \frac{\mathbf{G}_t^{-1}\mathbf{k}_{t+1}}{1 + \mathbf{k}_{t+1}^\top\mathbf{G}_t^{-1}\mathbf{k}_{t+1}} = \left(\mathbf{G}_t^{-1} - \frac{\mathbf{G}_t^{-1}\mathbf{k}_{t+1}\mathbf{k}_{t+1}^\top\mathbf{G}_t^{-1}}{1 + \mathbf{k}_{t+1}^\top\mathbf{G}_t^{-1}\mathbf{k}_{t+1}}\right)\mathbf{k}_{t+1} = \mathbf{P}_{t+1}\mathbf{k}_{t+1}.
$$

Again, using Sherman-Morrison.

## C.2. Linear attention and DeltaNet are not equal under approximate preconditioning

**Theorem C.2.** $\mathbf{S}_T^{PLA} \neq \mathbf{S}_T^{PDN}$ *for approximate preconditioning.*

*Proof.* Let $\mathbf{S}_T^{APLA}$ and $\mathbf{S}_T^{APDN}$ the approximately preconditioned states. It suffices to show this for the diagonal approximate preconditioning case. To show this, we consider the case where $t = 1$, and assume that $\mathbf{S}_0^{APLA} = \mathbf{S}_0^{APDN} = 0$.

Let $\lambda = 1$, $\boldsymbol{v}_1 = [1, 1]^\top$ and $\boldsymbol{k}_1 = [1, 1]^\top$. We have $\mathbf{G}_0 = \mathbf{I}$ and $\mathbf{G}_1 = \mathbf{I} + \mathbf{1}\mathbf{1}^\top$. Let $\hat{\mathbf{G}}$ be the diagonal approximation, so $\hat{\mathbf{G}}_1 = 2\mathbf{I}$.

$$\mathbf{S}_1^{APLA} = \mathbf{C}_1 \hat{\mathbf{P}}_1 = (\boldsymbol{v}_1 \boldsymbol{k}_1^\top)(\hat{\mathbf{G}}_1)^{-1}$$
$$= (\mathbf{1}\mathbf{1}^\top)(\frac{1}{2}I) = \frac{1}{2}\mathbf{1}\mathbf{1}^\top$$

Now expanding $\mathbf{S}_1^{APDN}$:

$$\tilde{\boldsymbol{k}}_1 = \frac{\hat{\mathbf{G}}_0^{-1} \boldsymbol{k}_1}{1 + \boldsymbol{k}_1^\top \hat{\mathbf{G}}_0^{-1} \boldsymbol{k}_1} = \frac{\mathbf{I}\mathbf{1}}{1 + \mathbf{1}^\top \mathbf{I}\mathbf{1}} = \frac{1}{3}\mathbf{1}$$

$$\mathbf{S}_1^{APDN} = \mathbf{S}_0^{APDN} + (\boldsymbol{v}_{t+1} - \mathbf{S}_0^{APDN} \boldsymbol{k}_1)\tilde{\boldsymbol{k}}_1^\top$$
$$= 0 + (\mathbf{1} - 0)\frac{1}{3}\mathbf{1}^\top = \frac{1}{3}\mathbf{1}\mathbf{1}^\top \neq \mathbf{S}_1^{APLA}$$

$\square$

## C.3. Stability of write-key formulation

As discussed in Appendix C.1, we have a choice of writing $\tilde{\boldsymbol{k}}_{t+1}$ as $\frac{\mathbf{G}_t^{-1} \boldsymbol{k}_{t+1}}{1 + \boldsymbol{k}_{t+1}^\top \mathbf{G}_t^{-1} \boldsymbol{k}_{t+1}}$ (1) or $\mathbf{G}_{t+1}^{-1} \boldsymbol{k}_{t+1}$ (2). In this diagonal approximate case, however, there is a distinction, and it is necessary to use method (1). The state transition matrix is given by $\mathbf{I} - \beta_{t+1} \boldsymbol{k}_{t+1} \tilde{\boldsymbol{k}}_{t+1}^\top$. For the eigenvalues of this matrix to be norm-bounded by 1, it is necessary that $0 \leq \boldsymbol{k}_{t+1}^\top \tilde{\boldsymbol{k}}_{t+1} \leq 2$. When using (1), this is guaranteed, even in the diagonal approximate case:

$$\boldsymbol{k}_{t+1}^\top \tilde{\boldsymbol{k}}_{t+1} = \boldsymbol{k}_{t+1}^\top \left( \frac{\hat{\mathbf{G}}^{-1} \boldsymbol{k}_{t+1}}{1 + \boldsymbol{k}_{t+1}^\top \hat{\mathbf{G}}_t^{-1} \boldsymbol{k}_{t+1}} \right)$$
$$= \frac{\boldsymbol{k}_{t+1}^\top \hat{\mathbf{G}}_t^{-1} \boldsymbol{k}_{t+1}}{1 + \boldsymbol{k}_{t+1}^\top \hat{\mathbf{G}}_t^{-1} \boldsymbol{k}_{t+1}}$$

which is bounded between 0 and 1 because $\hat{\mathbf{G}}_t$ is positive definite. However, it is not the case that $\boldsymbol{k}_{t+1}^\top \hat{\mathbf{G}}_{t+1}^{-1} \boldsymbol{k}_{t+1}$ is bounded, as there is a normalizing term in the denominator.

## C.4. MesaNet parallelization

Exact inverse Gram updates follow the Sherman-Morrison recursion

$$\mathbf{P}_t = \mathbf{P}_{t-1} - \frac{\mathbf{P}_{t-1} \boldsymbol{k}_t \boldsymbol{k}_t^\top \mathbf{P}_{t-1}}{1 + \boldsymbol{k}_t^\top \mathbf{P}_{t-1} \boldsymbol{k}_t},$$

which depends on $\mathbf{P}_{t-1}$ in a way that cannot be summarized into a chunk-local affine transition. As a result, one cannot precompute per-chunk update statistics for $\mathbf{P}$ without performing the full within-chunk sequence of inverse updates, precluding a chunkwise parallel form from being constructed. Thus, Mesalayer (von Oswald et al., 2024), which employs the naive, sequential form of the recurrence, is quite inefficient. We note that it is possible to use an intra-chunk sequential, inter-chunk parallel form, by computing the Gram matrix at chunk boundaries in parallel, and running the recurrence sequentially within a chunk. Future work could explore this option.

In contrast, MesaNet (von Oswald et al., 2025), solves these inverses in a parallel form by iteratively performing conjugate gradient steps. The key insight for this is that conjugate gradient directions can be computed by repeatedly applying linear

attention with a fixed state, but varying queries, which can be implemented efficiently. The limitation of this is that to do the backwards pass, it is necessary to use the Implicit Value Theorem to avoid storing the conjugate gradient unrolling steps used in the forward pass. In the case of non-optimality of the forward pass (i.e. not full convergence), this process can lead to incorrect gradients, offering potential intuition as to the training instabilities we observed when training MesaNet ourselves (see Appendix E.5 for more details).

## C.5. DeltaNet better approximates least squares than linear attention

For this analysis, we set the ridge to $\lambda = 0$, i.e. $\mathbf{G}_t := \sum_{i=1}^{t} \mathbf{k}_i \mathbf{k}_i^\top$ and $\mathbf{P}_t = \mathbf{G}_t^{-1}$.

*Proof.* Since $\mathbf{k}_t, \mathbf{q}_t \sim \mathrm{Unif}(\mathbb{S}^{d-1})$, we have $\|\mathbf{k}_t\|_2 = \|\mathbf{q}_t\|_2 = 1$ as $\mathbb{E}[\mathbf{k}_t \mathbf{k}_t^\top] = \mathbb{E}[\mathbf{q}_t \mathbf{q}_t^\top] = \frac{1}{d}\mathbf{I}$, and $(\mathbf{k}_t \mathbf{k}_t^\top)^2 = \mathbf{k}_t \mathbf{k}_t^\top$.

Recall from the normal equations (Eq. (2) in the main text) that the exact least-squares map at time $t$ is $\mathbf{S}_t^\star = \mathbf{C}_t \mathbf{P}_t$ with $\mathbf{P}_t = \mathbf{G}_t^{-1}$, and hence $\mathbf{o}_t^\star = (\mathbf{C}_t \mathbf{P}_t)\mathbf{q}_t$ (equivalently $\mathbf{o}_t^\star = \mathbf{C}_t(\mathbf{P}_t \mathbf{q}_t)$ by ATK/ATQ).

**Noiseless least-squares regime.** As in the noiseless regression setting, assume there exists a fixed map $\mathbf{S}^\star$ such that $\mathbf{v}_t = \mathbf{S}^\star \mathbf{k}_t$ for all $t$. Then

$$\mathbf{C}_t = \sum_{i=1}^{t} \mathbf{v}_i \mathbf{k}_i^\top = \mathbf{S}^\star \sum_{i=1}^{t} \mathbf{k}_i \mathbf{k}_i^\top = \mathbf{S}^\star \mathbf{G}_t,$$

so (for $t \geq d$, when $\mathbf{G}_t$ is invertible) we have $\mathbf{S}_t^\star = \mathbf{C}_t \mathbf{P}_t = \mathbf{S}^\star$ and therefore $\mathbf{o}_t^\star = \mathbf{S}^\star \mathbf{q}_t$. Moreover, linear attention outputs $\mathbf{o}_t^{\mathrm{LA}} = \mathbf{C}_t \mathbf{q}_t = \mathbf{S}^\star \mathbf{G}_t \mathbf{q}_t$.

**Averaging over isotropic queries.** For any (possibly random) matrix $\mathbf{A}$ independent of $\mathbf{q}_t$,

$$\mathbb{E}_{\mathbf{q}_t} \|\mathbf{A}\mathbf{q}_t\|_2^2 = \mathrm{tr}\Big(\mathbf{A}\, \mathbb{E}[\mathbf{q}_t \mathbf{q}_t^\top]\, \mathbf{A}^\top\Big) = \frac{1}{d}\|\mathbf{A}\|_F^2. \tag{C.5.1}$$

**Linear attention deviation from exact LS.** We have

$$\mathbf{o}_t^{\mathrm{LA}} - \mathbf{o}_t^\star = \mathbf{S}^\star(\mathbf{G}_t - \mathbf{I})\mathbf{q}_t.$$

Conditioning on $\{\mathbf{k}_i\}_{i \leq t}$ and using Eq. (C.5.1),

$$\mathbb{E}_{\mathbf{q}_t}\|\mathbf{o}_t^{\mathrm{LA}} - \mathbf{o}_t^\star\|_2^2 = \frac{1}{d}\|\mathbf{S}^\star(\mathbf{G}_t - \mathbf{I})\|_F^2 = \frac{1}{d}\,\mathrm{tr}\big(\mathbf{S}^{\star\top}\mathbf{S}^\star(\mathbf{G}_t - \mathbf{I})^2\big).$$

By rotational invariance of $\mathbf{G}_t$ under i.i.d. isotropic keys, $\mathbb{E}[(\mathbf{G}_t - \mathbf{I})^2] = \gamma_t \mathbf{I}$ for some scalar $\gamma_t = \frac{1}{d}\mathbb{E}\|\mathbf{G}_t - \mathbf{I}\|_F^2$, hence

$$\mathbb{E}\|\mathbf{o}_t^{\mathrm{LA}} - \mathbf{o}_t^\star\|_2^2 = \frac{\|\mathbf{S}^\star\|_F^2}{d}\,\gamma_t.$$

It remains to compute $\gamma_t$. Using $\mathrm{tr}(\mathbf{G}_t) = t$ and

$$\mathrm{tr}(\mathbf{G}_t^2) = t + \sum_{i \neq j}(\mathbf{k}_i^\top \mathbf{k}_j)^2, \qquad \mathbb{E}(\mathbf{k}_i^\top \mathbf{k}_j)^2 = \frac{1}{d}\ (i \neq j),$$

we obtain

$$\mathbb{E}\|\mathbf{G}_t - \mathbf{I}\|_F^2 = \Big(t + \frac{t(t-1)}{d}\Big) - 2t + d = d - t + \frac{t(t-1)}{d},$$

so

$$\mathbb{E}\|\mathbf{o}_t^{\mathrm{LA}} - \mathbf{o}_t^\star\|_2^2 = \frac{\|\mathbf{S}^\star\|_F^2}{d}\Big(1 - \frac{t}{d} + \frac{t(t-1)}{d^2}\Big). \tag{C.5.2}$$

**DeltaNet deviation from exact LS.** Let $\mathbf{S}_t$ denote the DeltaNet state with $\mathbf{S}_0 = \mathbf{0}$ and update

$$\mathbf{S}_t = \mathbf{S}_{t-1} + \beta_t(\boldsymbol{v}_t - \mathbf{S}_{t-1}\boldsymbol{k}_t)\boldsymbol{k}_t^\top, \qquad \beta_t \in [0, 1].$$

Define the operator error $\boldsymbol{\Delta}_t := \mathbf{S}^\star - \mathbf{S}_t$. Using $\boldsymbol{v}_t = \mathbf{S}^\star \boldsymbol{k}_t$,

$$\boldsymbol{\Delta}_t = \boldsymbol{\Delta}_{t-1} - \beta_t(\boldsymbol{\Delta}_{t-1}\boldsymbol{k}_t)\boldsymbol{k}_t^\top = \boldsymbol{\Delta}_{t-1}(\mathbf{I} - \beta_t\,\boldsymbol{k}_t\boldsymbol{k}_t^\top).$$

Since $(\boldsymbol{k}_t\boldsymbol{k}_t^\top)^2 = \boldsymbol{k}_t\boldsymbol{k}_t^\top$, we have

$$\|\boldsymbol{\Delta}_t\|_F^2 = \|\boldsymbol{\Delta}_{t-1}\|_F^2 - \beta_t(2 - \beta_t)\|\boldsymbol{\Delta}_{t-1}\boldsymbol{k}_t\|_2^2 \le \|\boldsymbol{\Delta}_{t-1}\|_F^2,$$

and therefore $\|\boldsymbol{\Delta}_t\|_F^2 \le \|\boldsymbol{\Delta}_0\|_F^2 = \|\mathbf{S}^\star\|_F^2$. Finally, $\boldsymbol{o}_t^{\mathrm{DN}} - \boldsymbol{o}_t^\star = (\mathbf{S}_t - \mathbf{S}^\star)\boldsymbol{q}_t = -\boldsymbol{\Delta}_t\boldsymbol{q}_t$ and thus by Eq. (C.5.1),

$$\mathbb{E}\|\boldsymbol{o}_t^{\mathrm{DN}} - \boldsymbol{o}_t^\star\|_2^2 = \frac{1}{d}\mathbb{E}\|\boldsymbol{\Delta}_t\|_F^2 \le \frac{\|\mathbf{S}^\star\|_F^2}{d}. \tag{C.5.3}$$

**Conclude existence of $t_0$.** By Eq. (C.5.2), the linear-attention error satisfies $\mathbb{E}\|\boldsymbol{o}_t^{\mathrm{LA}} - \boldsymbol{o}_t^\star\|_2^2 = \frac{\|\mathbf{S}^\star\|_F^2}{d}\left(1 - \frac{t}{d} + \frac{t(t-1)}{d^2}\right)$, and the bracketed factor is $\ge 1$ for all $t \ge d + 1$. Combining with Eq. (C.5.3), we obtain that for all $t \ge d + 1$,

$$\mathbb{E}\|\boldsymbol{o}_t^{\mathrm{DN}} - \boldsymbol{o}_t^\star\|_2^2 \le \frac{\|\mathbf{S}^\star\|_F^2}{d} \le \mathbb{E}\|\boldsymbol{o}_t^{\mathrm{LA}} - \boldsymbol{o}_t^\star\|_2^2.$$

Thus the claim holds with (for example) $t_0 = d + 1$. $\qquad\square$

# D. Algorithm and implementation details

## D.1. Pseudocode comparing the GDN and PGDN chunkwise parallel kernels

*Figure 8.* Pseudocode for forward pass of GDN vs. PGDN chunkwise parallel kernels. The PKDA forward pass is analogously adapted from the KDA forward pass.

**(a) GDN recurrence forward pass**

```
1 # Recurrence:
2 #   S_t = alpha * S_{t-1}
3 #       + beta * k @ (v - alpha * S_{t-1}^T @ k)^T
4 # where alpha = exp(g)
5 #
6 # Notation:
7 #   h = S (hidden state)
8 #   g = log(alpha), so exp(g) = alpha (decay gate)
9
10 def gated_delta_rule_forward(q, k, v, g, beta, initial_state):
11     # Step 1: Cumsum of log-space gate
12     g_cumsum = chunk_local_cumsum(g)
13
14     # Step 2: Symmetric KKT matrix (k @ k^T)
15     A = chunk_scaled_dot_kkt_fwd(k, g_cumsum, beta)
16     A = solve_tril(A)
17
18     # Step 3: WY representation
19     w, u = recompute_w_u_fwd(k, v, beta, A, g_cumsum)
20
21     # Step 4: Hidden state update
22     h, v_new, final_state = chunk_gated_delta_rule_fwd_h(
23         k, w, u, g_cumsum, initial_state
24     )
25
26     # Step 5: Output
27     o = chunk_fwd_o(q, k, v_new, h, g_cumsum)
28
29     return o, final_state
```

**(b) PGDN recurrence forward pass**

```
1  # Recurrence:
2  #   ATK:
3  #      A_t = alpha_atk * A_{t-1} + beta_atk * k^2    (diagonal)
4  #   Delta Rule:
5  #      S_t = alpha * S_{t-1}
6  #            + beta * k_precond @ (v - alpha * S_{t-1}^T @ k)^T
7  #
8  # where alpha_atk = exp(g_atk), alpha = exp(g)
9  #
10 # Notation:
11 #   h = S (hidden state)
12 #   g = log(alpha), g_atk = log(alpha_atk)
13 #   ac = accumulated ATK state at chunk boundaries (inter-chunk)
14 #   a_atk = per-chunk ATK state (intra-chunk)
15 #   k_precond = k * B(A), where B is the squash function
16 #
17 # Key:
18 #   k for reading, k_precond for writing
19
20 def precond_gated_delta_rule_forward(
21     q, k, v, g_atk, g, beta_atk, beta, initial_state, log_A_scale
22 ):
23     # Step 1 [NEW]: ATK diagonal recurrence
24     # Outputs k_precond = k * B(A)
25     k_precond, ac, a_atk, at = fused_atk_fwd(
26         k, beta_atk, g_atk, log_A_scale
27     )
28
29     # Step 2: Cumsum of log-space gate (uses g, not g_atk)
30     g_cumsum = chunk_local_cumsum(g)
31
32     # Step 3 [MODIFIED]: Asymmetric KKT
33     # (k @ k_precond^T instead of k @ k^T)
34     A = chunk_precond_kkt_fwd(k, k_precond, g_cumsum, beta)
35     A = solve_tril(A)
36
37     # Step 4: WY representation (uses k for reading)
38     w, u = recompute_w_u_fwd(k, v, beta, A, g_cumsum)
39
40     # Step 5 [MODIFIED]: Hidden state update
41     # (uses k_precond for writing)
42     h, v_new, final_state = chunk_gated_delta_rule_fwd_h(
43         k_precond, w, u, g_cumsum, initial_state
44     )
45
46     # Step 6 [MODIFIED]: Output
47     # (uses k_precond for local attention)
48     o = chunk_fwd_o(q, k_precond, v_new, h, g_cumsum)
49
50     return o, final_state, k_precond, ac, a_atk, at
```

*Figure 9.* Pseudocode for backward pass of GDN vs. PGDN chunkwise parallel kernels. The PKDA backward pass is analogously adapted from the KDA backward pass.

**(a) GDN recurrence backward pass**

```
1  def gated_delta_rule_backward(
2      q, k, v, g_cumsum, beta, A, initial_state, do, dht
3  ):
4      # Recompute forward intermediates
5      w, u = recompute_w_u_fwd(k, v, beta, A, g_cumsum)
6      h, v_new, _ = chunk_gated_delta_rule_fwd_h(
7          k, w, u, g_cumsum, initial_state
8      )
9
10     # Step 1-3: Backward through output and state
11     dv = chunk_bwd_dv_local(q, k, g_cumsum, do)
12     dh, dh0, dv = chunk_gated_delta_rule_bwd_dhu(
13         q, k, w, g_cumsum, initial_state, dht, do, dv
14     )
15     dq, dk, dw, dg = chunk_bwd_dqkwg(
16         q, k, v_new, w, g_cumsum, h, dv, do, dh
17     )
18
19     # Step 4: Backward through WY (symmetric)
20     dk2, dv, dbeta, dg2 = prepare_wy_repr_bwd(
21         k, v, beta, g_cumsum, A, dw, dv
22     )
23
24     # Step 5: Combine gradients
25     dk = dk + dk2
26     dg = reverse_cumsum(dg + dg2)
27
28     return dq, dk, dv, dbeta, dg, dh0
```

**(b) PGDN recurrence backward pass**

```
1 def precond_gated_delta_rule_backward(
2     q, k, v, g_atk, g_cumsum, beta_atk, beta, A,
3     ac, a_atk, k_precond, initial_state, do, dht, log_A_scale
4 ):
5     # Recompute forward intermediates
6     w, u = recompute_w_u_fwd(k, v, beta, A, g_cumsum)
7     h, v_new, _ = chunk_gated_delta_rule_fwd_h(
8         k_precond, w, u, g_cumsum, initial_state
9     )
10
11    # Step 1-3 [MODIFIED]:
12    # Backward through output and state (uses k_precond instead of k)
13    dv = chunk_bwd_dv_local(q, k_precond, g_cumsum, do)
14    dh, dh0, dv = chunk_gated_delta_rule_bwd_dhu(
15        q, k_precond, w, g_cumsum, initial_state, dht, do, dv
16    )
17    dq, dk_precond, dw, dg = chunk_bwd_dqkwg(
18        q, k_precond, v_new, w, g_cumsum, h, dv, do, dh
19    )
20
21    # Step 4 [MODIFIED]: Asymmetric WY backward
22    # (returns separate dk_wy and dk_precond_wy)
23    dk_wy, dk_precond_wy, dv, dbeta, dg2 = prepare_precond_wy_repr_bwd(
24        k, k_precond, v, beta, g_cumsum, A, dw, dv
25    )
26    dk_precond = dk_precond + dk_precond_wy
27
28    # Step 5 [NEW]: ATK backward
29    # (propagates dk_precond through k_precond = k * B(A))
30    dk_atk, dbeta_atk, dg_atk, d_log_A_scale = atk_backward(
31        k, g_atk, beta_atk, ac, a_atk, dk_precond, log_A_scale
32    )
33
34    # Step 6: Combine gradients
35    dk = dk_wy + dk_atk
36    dg = reverse_cumsum(dg + dg2)
37
38    return dq, dk, dv, dbeta_atk, dbeta, dg_atk, dg, dh0, d_log_A_scale
```

## D.2. The DPLR taxonomy from an algorithmic standpoint

*Table 3.* Tying forms of various linear recurrences within the DPLR space parameterized by $\mathbf{S}_t = \mathbf{S}_{t-1}\big(\mathbf{D}_t - \boldsymbol{a}_t \boldsymbol{b}_t^\top\big) + \boldsymbol{v}_t \boldsymbol{k}_t^\top$.

| Model | Tying |
|-------|-------|
| DN | $\boldsymbol{a}_t = \boldsymbol{b}_t = \boldsymbol{k}_t$ |
| GDN | $\boldsymbol{a}_t = \boldsymbol{b}_t = \boldsymbol{k}_t$ |
| KDA | $\boldsymbol{a}_t = \boldsymbol{b}_t = \boldsymbol{k}_t$ |
| Comba | $\boldsymbol{a}_t = \boldsymbol{b}_t = \boldsymbol{k}_t$ |
| RWKV-7 | $\boldsymbol{a}_t = \boldsymbol{b}_t, \; \boldsymbol{k}_t$ |
| PDN | $\boldsymbol{a}_t, \boldsymbol{b}_t = \boldsymbol{k}_t$ |
| PGDN | $\boldsymbol{a}_t, \boldsymbol{b}_t = \boldsymbol{k}_t$ |
| PKDA | $\boldsymbol{a}_t, \boldsymbol{b}_t = \boldsymbol{k}_t$ |

As discussed in Section 4.1, the DPLR taxonomy proposes the following functional form for recurrence design: $\mathbf{S}_t = \mathbf{S}_{t-1}\big(\mathbf{D}_t - \boldsymbol{a}_t \boldsymbol{b}_t^\top\big) + \boldsymbol{v}_t \boldsymbol{k}_t^\top$. Here, we discuss how existing models and our preconditioned variants reside in the DPLR framework, particularly as it pertains to constrained forms of tying among the $\boldsymbol{a}_t, \boldsymbol{b}_t, \boldsymbol{k}_t$ vectors as shown in Table 3.

**Efficiency of constrained DPLR via interaction tensors.** We write the DPLR-style update in the right-multiplying form used throughout this paper,

$$\mathbf{S}_t \;=\; \mathbf{S}_{t-1}\big(\mathbf{D}_t - \boldsymbol{a}_t \boldsymbol{b}_t^\top\big) \;+\; \boldsymbol{v}_t \boldsymbol{k}_t^\top. \tag{D.2.1}$$

In chunkwise parallel implementations (see Section 6.2 of the KDA paper) of the general DPLR form, the dominant intra-chunk cost is governed by the number of distinct lower-triangular interaction tensors (BT×BT) that must be formed. Denoting the per-chunk interaction matrices by

$$\mathbf{A}_{xy}[i,j] \;\propto\; \langle \boldsymbol{x}_i, \boldsymbol{y}_j \rangle \quad (i,j \text{ within the same chunk}),$$

the required set of interactions depends on how $\boldsymbol{a}_t, \boldsymbol{b}_t, \boldsymbol{k}_t$ are tied.

**(i) General DPLR (no tying).** In the general case, the chunkwise DPLR computation requires four distinct interaction tensors (Listing 8a in the KDA paper):

$$\mathbf{A}_{qk}, \quad \mathbf{A}_{qb}, \quad \mathbf{A}_{ak}, \quad \mathbf{A}_{ab}.$$

These additional interaction paths are precisely what the KDA paper highlights as a major source of inefficiency, compounded in practice by the need for numerically stable handling of reciprocal cumulative decays in the chunkwise formulation.

**(ii) KDA tying (single-key case).** KDA corresponds to a constrained DPLR family in which the relevant vectors are all derived from the same key stream such that $a_t = b_t = k_t$ (up to scaling). This means that the required interactions collapse to two (Listing 8b in the KDA report):

$$\mathbf{A}_{qk}, \quad \mathbf{A}_{kk}.$$

KDA further notes that this constraint also removes several matrix multiplications in the inter-chunk/output paths relative to general DPLR (Listing 8a vs. Listing 8b), yielding a substantially more efficient kernel.
Note that this tying scheme is also used in DeltaNet and GDN.

**(iii) RWKV-7 tying ($a = b$, $k$ free).** If one ties only $a_t = b_t$ while keeping the additive-update key $k_t$ independent, then

$$\mathbf{A}_{ab} = \mathbf{A}_{aa}, \qquad \mathbf{A}_{qb} = \mathbf{A}_{qa},$$

but the interactions involving $k$ remain distinct. As a result, the required interaction tensors remain four in general:

$$\mathbf{A}_{qk}, \quad \mathbf{A}_{qa}, \quad \mathbf{A}_{ak}, \quad \mathbf{A}_{aa}.$$

Thus, tying $a = b$ alone does not yield the two-interaction structure that underlies the efficiency of KDA. We note that this tying scheme is used in the RWKV-7 (Peng et al., 2025a) model, which necessitates the use of the general DPLR kernel in `flash-linear-attention`.

**(iv) Our tying ($b = k$, $a$ free).** Because Eq. (D.2.1) updates along the right direction $k_t$ via $v_t k_t^\top$, aligning the right factor of the low-rank term with the same write direction by setting

$$b_t \equiv k_t, \qquad a_t \text{ free}, \tag{D.2.2}$$

this collapses the interaction set as follows:

$$\mathbf{A}_{qb} = \mathbf{A}_{qk}, \qquad \mathbf{A}_{ab} = \mathbf{A}_{ak}.$$

Consequently, the chunkwise computation requires only two interaction tensors,

$$\mathbf{A}_{qk}, \quad \mathbf{A}_{ak},$$

matching the same two-interaction scaling as KDA. Relative to the RWKV-7 tying scheme $a = b$, the constraint Eq. (D.2.2) is strictly more efficient at the intra-chunk level because it eliminates two interaction tensors outright, whereas $a = b$ does not reduce the general DPLR interaction count.

**Distinguishing between the write and read key.** From an interpretability standpoint, in our preconditioned recurrences (i.e. PDN, PGDN, PKDA), we can intuit $a_t$ as the read key and $b_t = k_t$ as the write key, each serving a distinct purpose in modulating the recurrent dynamics as discussed in Section 4.3.3

**Transpose convention.** The KDA paper presents the DPLR recurrence using the left-multiplying form $\mathbf{S}_t = (\mathbf{D}_t - a_t b_t^\top)\mathbf{S}_{t-1} + k_t v_t^\top$. Under this transposed convention, the roles of $a$ and $b$ swap with respect to the write direction, so our tying $b \equiv k$ in Eq. (D.2.2) corresponds to the statement $a \equiv k$ with $b$ free using the notation in KDA.

# E. Experimental setup and additional results

## E.1. Experimental details

**Training hyperparameters.** We follow the same training setup as prior work, (Yang et al., 2024; Hu et al., 2025; Yang et al., 2025) where we use 8 H100 GPUs for 340M and 1B language modeling experiments. We train all of our models using the `flame` repository with the following hyperparameters:

- AdamW for optimization

- Peak learning rate of $4 \times 10^{-4}$, cosine learning rate schedule, initial and final learning rates set at $4 \times 10^{-5}$

- Warmup phase of 0.5B tokens

- Batch size of 0.5M tokens

- Weight decay of 0.01 and gradient clipping of 1.0

The 340M models are trained using 15B tokens, while the 1B models are trained with 50 billion tokens.

**Architectural hyperparameters.** For the 340M DeltaNet, GDN, and KDA models, the following architectural hyperparameters are used to instantiate the model. All hyperparameters that are not specified follow the default ones as implemented in `flash-linear-attention`. The model dimension is set to 1024, the number of heads is set to 8, and the key and value dimensions are set to 128 (i.e. no expansion factor). A total of 24 layers are used in the network, each of which consists of both a sequence mixing layer and a channel mixing layer (SwiGLU). For DeltaNet and Gated DeltaNet, we do not enable the use of an output gate after the recurrence. However, in KDA, we are forced, since there is no option to disable the output gate in the KDA kernel in `flash-linear-attention`, as its computation is fused into the recurrence. Furthermore, since all kernels in `flash-linear-attention` are optimized for strides of 16, using a smaller head dimension of 120 in KDA to construct an iso-parameter comparison across DeltaNet, GDN and KDA results significantly reduces throughput. As we wanted the architectural configurations we analyzed in our training throughput analyses to align with the models we trained, the model referred to as 340M KDA actually has 355M parameters.

For the 1B GDN and KDA models, we increase the number of heads to 14. In the GDN 1B, we keep the head dimension as 128, resulting in a model dimension of 1792. To enable an iso-parametric comparison at the larger scale, in the KDA 1B model, we scale the model dimension down to 1680, resulting in a head dimension of 120 to account for the use of output gates in KDA. We do not train DeltaNet at the 1B scale, given the significantly worse performance we observe at the 340M scale relative to GDN and KDA due to the lack of a decay term in the recurrence.

We also note that for all models, we L2 normalize the keys and queries such that $||\boldsymbol{k}_t|| = 1$ and $||\boldsymbol{q}_t|| = 1$. This point will be relevant regarding our discussion on the eigenvalues of the recurrence in Appendix E.3.

**Preconditioner hyperparameters.** The PDN, PGDN, and PKDA models share all the same architectural components with their unpreconditioned baselines, with marginal parametric overhead coming from parameters used in the preconditioner recurrence. The additional parameters include $\alpha_t^P$ and $\beta_t^P$ projection layers used to parameterize the decay and gain terms, respectively, in the preconditioner recurrence (each of which adds $DH$ parameters to the layer) and a per-head learnable scalar to center the learned preconditioner distribution (which adds $H$ parameters to the layer). Note that $D$ denotes the model dimension and $H$ denotes the number of heads. The last relevant parameter of the preconditioner recurrence is the value of $x$, which we set to be 1.5, meaning that the elements of the learned preconditioner $\mathbf{B}_t$ lie in the interval $[\frac{2}{3}, \frac{3}{2}]$. For more details on the parameterization of the preconditioner recurrence, refer to Appendix E.3.

**Evaluation suite.** For our small-scale synthetic experiments, to probe the associative recall capabilities of our model, we evaluate on the MQAR benchmark and vary the number of key-value pairs as well as the sequence length to modulate the difficulty of the task. For all other hyperparameter settings, we use the defaults from (Arora et al., 2024a).

Following the evaluation scheme used in (Yang et al., 2025; Hu et al., 2025), for our large-scale language models, we test on a range of commonsense-reasoning benchmarks, including PIQA (Bisk et al., 2020), HellaSwag (Zellers et al., 2019), WinoGrande (Sakaguchi et al., 2020), ARC-Easy and ARC-Challenge (Clark et al., 2018), WikiText (Merity et al., 2017), LAMBADA (Paperno et al., 2016), BoolQ (Clark et al., 2019) and SciQ (Welbl et al., 2017).

Additionally, we evaluate our models on both synthetic and real-world in-context retrieval (ICR) tasks. For synthetic long-context retrieval, we evaluate on the Single Needle-In-A-Haystack (S-NIAH) subset of the RULER (Hsieh et al., 2024a) benchmark. In particular, we evaluate our models on instances of this task corresponding to increasing levels of difficulty: S-NIAH-1, S-NIAH-2, and S-NIAH-3. For real-world retrieval, following (Arora et al., 2024c), we evaluate on the SWDE

(Lockard et al., 2019), FDA (Arora et al., 2023), SQuAD (Rajpurkar et al., 2018), TQA (Joshi et al., 2017) and DROP (Dua et al., 2019) benchmarks. Note that for TQA and DROP, we used the task versions in `lm-evaluation-harness` corresponding to the ones presented in (Arora et al., 2024b). We omit the NQ (Kwiatkowski et al., 2019) benchmark from the set of evaluations since our models were trained at a context length of 2K and most of the examples in the NQ suite exceed this length.

## E.2. Results continued

### E.2.1. ADDITIONAL MQAR RESULTS

*Figure 10.* Additional results on the synthetic MQAR task. Here, we keep the sequence length fixed at 1024 and vary the number of KV pairs. As we observe in Section 4.3.2 with the other MQAR tasks configurations, we find that preconditioned recurrences either maintain or improve performance.

### E.2.2. FULL LANGUAGE RESULTS

*Table 4.* Full table of language modeling evaluation results for 340M and 1B models trained on the SlimPajama dataset. This includes results on 340M DN, GDN, and KDA models where we allow $\beta_t$ to vary in $[0, 2]$ (instead of the default $[0, 1]$), which enables these recurrences to learn negative eigenvalues (see (Grazzi et al., 2025)). These models are denoted by $[-1, 1]$. All tasks are evaluated using `lm-evaluation-harness`.

| Model | | Commonsense Reasoning ↑ | | | | | | | | | In-context retrieval ↑ | | | | | |
|---|---|---|---|---|---|---|---|---|---|---|---|---|---|---|---|---|
| | LAMB ppl↓ | Wiki ppl↓ | ARC$_e$ acc | ARC$_c$ acc$_n$ | HellaS acc$_n$ | Lamb. acc | PIQA acc | WinoG acc | BoolQ acc | SciQ acc | Avg acc | FDA acc | SWDE acc | SQuAD acc | TQA acc | DROP acc | Avg acc |
| *340M params with 15B training tokens and 0.5M batchsize tokens* | | | | | | | | | | | | | | | | |
| DN $[-1, 1]$ | 48.30 | 28.96 | 44.53 | 23.89 | 33.99 | 29.07 | 65.02 | 52.01 | 55.10 | 75.10 | 47.34 | 9.53 | 30.15 | 28.95 | 29.60 | 15.40 | 22.73 |
| DN | 45.76 | 29.04 | 44.02 | 23.55 | 34.08 | 29.58 | 65.07 | 50.91 | 56.80 | 75.30 | 47.41 | 8.53 | 27.09 | 28.32 | 31.20 | 17.80 | 22.59 |
| **PDN** | 43.05 | 29.05 | 44.70 | 23.52 | 33.83 | 30.48 | 65.14 | 52.41 | 58.50 | 75.10 | 47.96 | 14.16 | 29.17 | 28.89 | 30.60 | 16.40 | 23.84 |
| GDN $[-1, 1]$ | 30.11 | 27.00 | 45.88 | 23.81 | 35.56 | 35.16 | 65.07 | 51.30 | 58.70 | 77.50 | 49.12 | 16.97 | 31.41 | 31.03 | 35.00 | 16.70 | 26.22 |
| GDN | 31.39 | 27.08 | 44.49 | 24.32 | 35.96 | 34.50 | 65.83 | 51.30 | 56.30 | 78.20 | 48.86 | 13.61 | 29.43 | 29.83 | 33.80 | 17.20 | 24.77 |
| **PGDN** | 28.43 | 26.86 | 47.81 | 24.21 | 36.19 | 35.55 | 64.83 | 52.97 | 60.10 | 78.00 | 49.96 | 14.25 | 32.24 | 31.76 | 35.20 | 17.40 | 26.17 |
| KDA $[-1, 1]$ | 28.57 | 26.28 | 46.80 | 23.89 | 36.09 | 36.08 | 64.64 | 50.75 | 55.50 | 78.90 | 49.08 | 16.42 | 32.49 | 30.73 | 35.00 | 15.30 | 25.99 |
| KDA | 31.37 | 26.18 | 45.45 | 22.70 | 36.06 | 34.04 | 66.00 | 52.25 | 57.00 | 79.50 | 49.12 | 13.88 | 34.11 | 29.69 | 34.80 | 16.90 | 25.88 |
| **PKDA** | 25.33 | 25.98 | 46.97 | 22.95 | 37.01 | 37.22 | 65.68 | 51.46 | 59.10 | 78.70 | 49.89 | 14.25 | 34.65 | 31.39 | 35.50 | 18.00 | 26.76 |
| *1B params with 50B training tokens and 0.5M batchsize tokens* | | | | | | | | | | | | | | | | |
| GDN | 13.46 | 18.05 | 56.10 | 27.30 | 48.91 | 46.42 | 70.08 | 54.22 | 55.80 | 84.70 | 55.44 | 19.69 | 49.86 | 37.70 | 41.60 | 20.50 | 33.87 |
| **PGDN** | 13.09 | 18.01 | 56.66 | 26.96 | 49.66 | 46.49 | 70.73 | 54.54 | 59.90 | 86.10 | 56.38 | 18.33 | 50.32 | 41.19 | 44.30 | 20.80 | 34.99 |
| KDA | 14.52 | 17.95 | 54.55 | 26.88 | 48.95 | 45.16 | 70.29 | 54.22 | 57.60 | 85.80 | 55.43 | 23.23 | 50.14 | 37.00 | 42.70 | 19.70 | 34.55 |
| **PKDA** | 11.86 | 17.82 | 57.24 | 26.96 | 49.05 | 48.52 | 70.62 | 55.49 | 58.80 | 86.40 | 56.64 | 21.69 | 48.79 | 38.61 | 44.60 | 20.70 | 34.88 |

*Table 5.* RULER (S-NIAH) evaluation results for 340M and 1B models across varying context lengths of 512, 1K, 2K, 4K, and 8K. This includes results on 340M DN, GDN, and KDA models where we allow $\beta_t$ to vary in $[0, 2]$ (instead of the default $[0, 1]$), which enables these recurrences to learn negative eigenvalues (see (Grazzi et al., 2025)). These models are denoted by $[-1, 1]$. All tasks are evaluated using `lm-evaluation-harness`.

| Model | RULER ↑ | | | | | | | | | | | | | | |
| --- | --- | --- | --- | --- | --- | --- | --- | --- | --- | --- | --- | --- | --- | --- | --- |
| | S-NIAH-1 | | | | | S-NIAH-2 | | | | | S-NIAH-3 | | | | |
| | 512 acc | 1K acc | 2K acc | 4K acc | 8K acc | 512 acc | 1K acc | 2K acc | 4K acc | 8K acc | 512 acc | 1K acc | 2K acc | 4K acc | 8K acc |
| *340M params with 15B training tokens and 0.5M batchsize tokens* | | | | | | | | | | | | | | | |
| DN | 100 | 100 | 100 | 100 | 100 | 100 | 91.8 | 95.4 | 29.2 | 8.4 | 79.2 | 18.0 | 33.4 | 19.0 | 3.0 |
| **PDN** | 100 | 100 | 100 | 99.8 | 99.8 | 100 | 100 | 99.2 | 28.4 | 14.2 | 90.6 | 82.8 | 59.4 | 22.0 | 2.0 |
| DN $[-1, 1]$ | 100 | 100 | 100 | 100 | 100 | 100 | 99.8 | 98.8 | 31.2 | 14.4 | 94.4 | 76.8 | 51.2 | 12.0 | 8.0 |
| GDN | 100 | 100 | 100 | 99.6 | 95.2 | 100 | 100 | 97.6 | 29.4 | 1.4 | 96.2 | 10.4 | 35.4 | 20.0 | 3.0 |
| **PGDN** | 100 | 100 | 99.8 | 100 | 98.8 | 100 | 100 | 93.8 | 52.8 | 31.4 | 99.2 | 89.0 | 66.8 | 63.8 | 12.8 |
| GDN $[-1, 1]$ | 100 | 100 | 100 | 100 | 99.8 | 100 | 100 | 88.4 | 31.6 | 15.2 | 99.6 | 78.8 | 62.2 | 24.8 | 13.6 |
| KDA | 100 | 100 | 100 | 99.8 | 96.8 | 100 | 100 | 98.8 | 37.6 | 11.4 | 93.6 | 77.0 | 58.4 | 24.6 | 3.2 |
| **PKDA** | 100 | 100 | 100 | 100 | 98.4 | 100 | 100 | 98.2 | 44.2 | 12.0 | 99.8 | 83.2 | 53.8 | 19.0 | 7.0 |
| KDA $[-1, 1]$ | 100 | 100 | 100 | 100 | 99.6 | 100 | 100 | 99.4 | 36.0 | 3.4 | 99.2 | 94.6 | 4.6 | 1.4 | 8.0 |
| *1B params with 50B training tokens and 0.5M batchsize tokens* | | | | | | | | | | | | | | | |
| GDN | 100 | 100 | 100 | 100 | 100 | 100 | 100 | 100 | 67.4 | 24.8 | 97.6 | 58.6 | 60.4 | 64.2 | 9.4 |
| **PGDN** | 100 | 100 | 100 | 100 | 100 | 100 | 100 | 99.0 | 90.2 | 11.3 | 98.2 | 90.8 | 76.8 | 63.8 | 12.8 |
| KDA | 100 | 100 | 100 | 100 | 100 | 100 | 100 | 100 | 80.4 | 22.8 | 95.2 | 98.0 | 80.2 | 40.8 | 5.4 |
| **PKDA** | 100 | 100 | 100 | 100 | 100 | 100 | 100 | 100 | 79.0 | 25.8 | 97.4 | 92.4 | 78.2 | 47.6 | 8.8 |

## E.3. Ablating on the functional form of the preconditioner recurrence

*Table 6.* Training loss from various ablations on the functional form of the preconditioner computed via a time-weighted EMA. All results are on 340M parameter models trained for 15B tokens. The base recurrence used in all the recurrences is GDN, apart from the ATQ case, which uses Mamba-2 with a gain term added to the write. See below for further discussion.

| | PGDN | GDN | P-Mamba-2 (ATQ) | PGDN (unstable ATK) | PGDN (tied) | GDN[-1, 1] |
| --- | --- | --- | --- | --- | --- | --- |
| **Train loss** | 2.511 | 2.528 | 2.543 | 2.536 | 2.518 | 5.520 |

**ATQ vs ATK.** The first ablation we highlight compares the ATQ and ATK approaches for preconditioning the recurrence. We note that while in the main text we formulated the diagonal preconditioner recurrences without a decay or write term to reduce notational clutter, in practice the diagonal preconditioner recurrence we consider is given by

$$\mathbf{A}_t = \alpha_t \mathbf{A}_{t-1} + \beta_t (\boldsymbol{k}_t \odot \boldsymbol{k}_t).$$

Recall that in the case of an approximate preconditioner, ATQ and ATK are no longer equivalent, and so a natural question arises as to which one is the better approach when using a diagonal approximation to the key Gram. To be concrete, the two recurrences we are comparing here are

$$\mathbf{S}_t = \alpha_t \mathbf{S}_{t-1} + \beta_t \boldsymbol{v}_t \boldsymbol{k}_t^\top, \qquad \boldsymbol{o}_t = \mathbf{S}_t \tilde{\boldsymbol{q}}_t, \qquad \tilde{\boldsymbol{q}}_t = \mathbf{A}_t^{-1} \boldsymbol{q}_t,$$

which corresponds to preconditioned Mamba-2, which is realized via the ATQ transform, and

$$\mathbf{S}_t = \alpha_t \mathbf{S}_{t-1} + \beta_t (\boldsymbol{v}_t - \alpha_t \mathbf{S}_{t-1} \boldsymbol{k}_t) \tilde{\boldsymbol{k}}_t^\top, \qquad \boldsymbol{o}_t = \mathbf{S}_t \boldsymbol{q}_t, \qquad \tilde{\boldsymbol{k}}_t = \frac{\mathbf{A}_{t-1}^{-1} \boldsymbol{k}_t}{1 + \boldsymbol{k}_t^\top \mathbf{A}_{t-1}^{-1} \boldsymbol{k}_t}.$$

which corresponds to preconditioned GDN, which is realized via the ATK transform. As shown in Table 6, the ATK (unstable) approach outperforms the ATQ approach, aligning with our theoretical result in Appendix C.5.

**Stable parameterization via $\mathbf{B}_t$** Although PGDN (unstable ATK) has lower training loss than P-Mamba-2, we note that the baseline GDN model still outperforms it (see Table 6), even though we were adding second-order information to the

recurrence. To pinpoint the source of instability, we investigated the distribution of the learned $\mathbf{A}_t$ and, as shown in the main text, found that they were log-normally distributed with heavy tails. This causes certain dimensions of the state to be significantly dampened, which negatively impacts training dynamics. To resolve this, we constrained the range of $\mathbf{A}_t$ to the interval $[\frac{1}{x}, x]$, where $x$ is a hyperparameter, using the following nonlinear, element-wise transformation:

$$\log(\mathbf{A}_t) - \mu, \boldsymbol{s}_t = \boldsymbol{r}_t \oslash (\mathbf{1} + |\boldsymbol{r}_t|), \mathbf{B}_t = \exp\big(-\log(x)\boldsymbol{s}_t\big)$$

where $\oslash$ denotes element-wise division and $\mu > 0$ is a learnable scalar, which we parameterize via $e^{\text{logAScale}}$ where logAScale is unconstrained.

*Figure 11.* Plot of learned centers $\mu$ across various layers and heads in the PGDN model.

The intuition behind the first term $\log(\mathbf{A}_t) - \mu$ is that since we know our distribution of $\mathbf{A}_t$ looks roughly log-normal, we should transform it into log-space and let it learn its mean $\mu$ there. This accomplishes two things: one, the model can now learn $\mathbf{A}_t$ in a space that looks roughly normal (which is likely easier to train) and two, it enables $\mathbf{A}_t$ to learn a distribution that is not necessarily centered at 1, allowing it to place itself around a potentially different point (as shown in Figure 11) where the preconditioner neither amplifies nor dampens the state.

The next term in the nonlinear transform is given by the squash function $f(r) = \frac{r}{1+|r|}$ which maps an input $r$ onto the interval $[-1, 1]$. The first thing we note about this function is that it moves quite fast as compared to the canonical tanh squash (see Figure 12). This is a useful property as it gives the function room to distribute the amplifying and dampening dynamics from $\mathbf{A}_t$ onto the interval $[-1, 1]$. This prevents $\mathbf{A}_t$ from saturating the squash function, promoting better training.

*Figure 12.* **Left.** Plot of the fast-moving inverted squash function $f(r) = \frac{r}{1+|r|}$ which we use in our preconditioned recurrences, as well as the slower-moving inverted tanh function, which we found performed worse empirically. **Right.** Plot of average saturation of each squash function across the sequence.

The final term in the transform on $\mathbf{A}_t$ is given by $\mathbf{B}_t = \exp\big(-\log(x)\boldsymbol{s}_t\big)$ which performs an inverse transform on the squashed $\mathbf{A}_t$. The benefit of this parameterization is that it enables us to learn the inverse of $\mathbf{A}_t$ implicitly, which avoids the numerical training instabilities associated with explicitly training in inverse space (von Oswald et al., 2024).

We briefly note two more important considerations that were taken into account when converging upon the final functional form of the preconditioner: (i) the range of the interval to map onto and (ii) the value of $x$.

*Figure 13.* Plot showing the learned distribution of $\mathbf{B}_t$ across a few layers in the PGDN model.

Regarding the former, aside from the interval we chose $[\frac{1}{x}, x]$, the other two immediately obvious candidates are $[1, x]$ and $[\frac{1}{x}, 1]$. The difference between these intervals and the one we chose is that they do not center around 1, meaning that they are biased towards learning amplifying and dampening dynamics only, respectively. Empirically, we found both to perform worse than the symmetric interval $[\frac{1}{x}, x]$, which is supported by the learned distribution of $\mathbf{B}_t$ in Figure 13, which shows both amplifying and dampening dynamics being learned by the preconditioner.

**The choice of $x$ and its impact on recurrent dynamics.** The second axis that we found to impact performance was the choice of the hyperparameter $x$. To understand why, we prove the following results regarding the role $x$ plays in bounding the eigenvalues (Tumma et al., 2024) in the recurrence.

**Lemma E.1** (Eigenvalues of the GDN/PGDN transition). *Consider the modified GDN update with distinct read and write keys*

$$\mathbf{S}_t = \alpha_t \mathbf{S}_{t-1} + \beta_t\big(\boldsymbol{v}_t - \alpha_t \mathbf{S}_{t-1}\boldsymbol{k}_t\big)\tilde{\boldsymbol{k}}_t^\top = \alpha_t \mathbf{S}_{t-1}\big(\mathbf{I} - \beta_t \boldsymbol{k}_t\tilde{\boldsymbol{k}}_t^\top\big) + \beta_t \boldsymbol{v}_t\tilde{\boldsymbol{k}}_t^\top,$$

*where $\alpha_t, \beta_t \in \mathbb{R}$ and $\boldsymbol{k}_t, \tilde{\boldsymbol{k}}_t \in \mathbb{R}^{d_k}$. Then the eigenvalues of the matrix factor $\alpha_t\big(\mathbf{I} - \beta_t \boldsymbol{k}_t\tilde{\boldsymbol{k}}_t^\top\big)$ are*

$$\alpha_t \quad (\text{mult. } d_k - 1), \qquad \alpha_t\big(1 - \beta_t\,\tilde{\boldsymbol{k}}_t^\top \boldsymbol{k}_t\big) \quad (\text{mult. } 1).$$

*In particular, if $\tilde{\boldsymbol{k}}_t = \boldsymbol{k}_t$ and $\|\boldsymbol{k}_t\|_2 = 1$ (the standard GDN case), then the eigenvalues are*

$$\alpha_t \quad (\text{mult. } d_k - 1), \qquad \alpha_t(1 - \beta_t) \quad (\text{mult. } 1).$$

Recall the GDN recurrence (from the main text) can be written in the form

$$\mathbf{S}_t = \alpha_t \mathbf{S}_{t-1} + \beta_t\big(\boldsymbol{v}_t - \alpha_t \mathbf{S}_{t-1}\boldsymbol{k}_t\big)\boldsymbol{k}_t^\top = \alpha_t \mathbf{S}_{t-1}\big(\mathbf{I} - \beta_t \boldsymbol{k}_t\boldsymbol{k}_t^\top\big) + \beta_t \boldsymbol{v}_t\boldsymbol{k}_t^\top.$$

In PGDN, we replace the write key by a vector $\tilde{\boldsymbol{k}}_t$, yielding

$$\mathbf{S}_t = \alpha_t \mathbf{S}_{t-1} + \beta_t\big(\boldsymbol{v}_t - \alpha_t \mathbf{S}_{t-1}\boldsymbol{k}_t\big)\tilde{\boldsymbol{k}}_t^\top = \alpha_t \mathbf{S}_{t-1}\big(\mathbf{I} - \beta_t \boldsymbol{k}_t\tilde{\boldsymbol{k}}_t^\top\big) + \beta_t \boldsymbol{v}_t\tilde{\boldsymbol{k}}_t^\top.$$

Assuming $\|\boldsymbol{k}_t\|_2 = 1$, the transition factor

$$\mathbf{I} - \beta_t \boldsymbol{k}_t\tilde{\boldsymbol{k}}_t^\top$$

has eigenvalue 1 with multiplicity $d_k - 1$ and one additional eigenvalue

$$1 - \beta_t\,\tilde{\boldsymbol{k}}_t^\top \boldsymbol{k}_t.$$

Therefore, the corresponding eigenvalues for the PGDN step are

$$\alpha_t \quad (\text{mult. } d_k - 1), \qquad \alpha_t\big(1 - \beta_t\,\tilde{\boldsymbol{k}}_t^\top \boldsymbol{k}_t\big) \quad (\text{mult. } 1).$$

In particular, for GDN where $\tilde{\boldsymbol{k}}_t = \boldsymbol{k}_t$ and $\|\boldsymbol{k}_t\|_2 = 1$, we obtain

$$\alpha_t \quad (\text{mult. } d_k - 1), \qquad \alpha_t(1 - \beta_t) \quad (\text{mult. } 1).$$

**Lemma E.2** (Sufficient condition for unit-disk eigenvalues in the PGDN recurrence). *Denote the recurrent transition matrix in the PGDN update by*

$$\mathbf{T}_t = \alpha_t\big(\mathbf{I} - \beta_t\,\boldsymbol{k}_t\tilde{\boldsymbol{k}}_t^\top\big), \qquad \alpha_t \in [0, 1], \ \beta_t \in [0, 1], \ \|\boldsymbol{k}_t\|_2 = 1,$$

*with $\tilde{k}_t = \mathbf{B}_t \boldsymbol{k}_t$ and $\mathbf{B}_t = \mathrm{diag}(b_i)$ satisfying $b_i \in [1/x, x]$ for some $x > 1$. If*

$$\beta_t x \leq 2,$$

*then every eigenvalue of $\mathbf{T}_t$ satisfies $|\lambda| \leq 1$ for every admissible $\mathbf{B}_t$. In particular, if $\beta_t \in [0, 1]$ and $x \leq 2$, the condition holds automatically.*

*Proof.* By Lemma E.1, the eigenvalues of $\mathbf{T}_t$ are $\alpha_t$ (mult. $d_k - 1$) and

$$\lambda(t) = \alpha_t \big(1 - \beta_t \, \tilde{\boldsymbol{k}}_t^\top \boldsymbol{k}_t\big).$$

Since $\|\boldsymbol{k}_t\|_2 = 1$ and $\tilde{\boldsymbol{k}}_t = \mathbf{B}_t \boldsymbol{k}_t$ with $b_i \in [1/x, x]$, we have $\tilde{\boldsymbol{k}}_t^\top \boldsymbol{k}_t = \boldsymbol{k}_t^\top \mathbf{B}_t \boldsymbol{k}_t \in [1/x, x]$. Hence

$$1 - \beta_t \, \tilde{\boldsymbol{k}}_t^\top \boldsymbol{k}_t \in \big[\, 1 - \beta_t x, \; 1 - \beta_t/x \,\big] \subseteq [-1, 1]$$

whenever $\beta_t x \leq 2$. Multiplying by $\alpha_t \in [0, 1]$ preserves the bound $|\lambda(t)| \leq 1$, and also $|\alpha_t| \leq 1$. Therefore, all eigenvalues of $\mathbf{T}_t$ lie in the unit disk. $\square$

Given these results, we conclude that $x$ must lie in the interval $[1, 2]$ where $x = 1$ reduces the PGDN to GDN and $x = 2$ takes PGDN to the edge of stability. As such, empirically, we search across a few values of $x$ in this interval and found $x = 1.5$ to perform the best, which interpolates exactly between these two regimes.

An additional benefit of this result is that if we keep $x$ within the stable interval $[1, 2]$, we are no longer constrained to the Sherman-Morrison form of the $\mathbf{A}_t$ preconditioner. Recall that the Sherman-Morrison form states that we must precondition the key with $\frac{\mathbf{A}_{t-1}^{-1} \boldsymbol{k}_t}{1 + \boldsymbol{k}_t^\top \mathbf{A}_{t-1}^{-1} \boldsymbol{k}_t}$ to realize a stable recurrent form (as discussed in Appendix C.4). As discussed in (von Oswald et al., 2024), this particular recurrence is difficult to train. Fortunately, since our stabilized preconditioner $\mathbf{B}_t$ innately has eigenstability from the $x \in [1, 2]$ constraint, we are no longer required to use the precise combination of the normalization term and the previous time step $\mathbf{A}_{t-1}$ to precondition the keys. Dropping the normalization term and using the current time step $f(\mathbf{A}_t) = \mathbf{B}_t$ where $f$ denotes the inverse squash function, yields the final form of the preconditioned recurrence. To substantiate this decision, we conduct ablations on various normalization factors in Table 7, showing that all of them worsen performance.

*Table 7.* Results from an ablation on the normalization factor applied to the preconditioned key show that no normalization performs the best when using $\mathbf{B}_t$ as the preconditioner.

|  | **no norm** | $\mathbf{1/(\boldsymbol{k}^\top \mathbf{B}_t \boldsymbol{k})}$ | $\mathbf{1/(1 + \boldsymbol{k}^\top \mathbf{B}_t \boldsymbol{k})}$ | $\boldsymbol{\beta_t}/(\alpha_t + \beta_t \, \boldsymbol{k}^\top \mathbf{B}_t \boldsymbol{k})$ |
|---|---|---|---|---|
| **Value** | 2.511 | 2.517 | 2.521 | 2.524 |

*Figure 14.* Impact of various normalization factors on learned decay and write terms in main PGDN recurrence. Here, $m_t = \boldsymbol{k}^T \mathbf{B}_t \boldsymbol{k}$.

**Tying $\alpha_t$ and $\beta_t$.** The final ablation axis we discuss is tying together the decay $\alpha_t$ and gain $\beta_t$ between the preconditioner recurrence and main recurrence, as is done in MesaNet (von Oswald et al., 2025). We find that untying improves training loss (as shown in Table 6), likely due to the distinct training dynamics between the approximate preconditioner recurrence and main recurrence.

### E.4. Augmenting baseline recurrences with negative eigenvalues does not outperform preconditioning

An additional nice property of the preconditioner parameterization we use is that the model, in theory, can learn negative eigenvalues if the following condition is met:

**Lemma E.3** (Sufficient condition for a negative eigenvalue). *Under the assumptions of Lemma E.1, if*

$$\beta_t x \;>\; 1,$$

*then the recurrent transition matrix admits a negative eigenvalue (for example, by taking $\mathbf{B}_t = x\mathbf{I}$).*

*Proof.* Take $\mathbf{B}_t = x\mathbf{I}$. Then (using $\|\mathbf{k}_t\|_2 = 1$ from the assumptions)

$$\mathbf{k}_t^\top \mathbf{B}_t \mathbf{k}_t \;=\; x.$$

By Lemma E.1, the nontrivial eigenvalue is

$$\lambda_*(t) \;=\; \alpha_t\big(1 - \beta_t\, \mathbf{k}_t^\top \mathbf{B}_t \mathbf{k}_t\big) \;=\; \alpha_t(1 - \beta_t x).$$

Since $\alpha_t > 0$, if $\beta_t x > 1$ then $\lambda_*(t) < 0$. $\qquad\square$

In practice, however, we find that the PGDN model does not learn negative eigenvalues, as shown in Figure 16. This is potentially a drawback, as we know from (Grazzi et al., 2025) that recurrences equipped with negative eigenvalues perform better on state tracking and in some instances, long-context retrieval tasks. In fact, all the baseline delta-rule recurrences can be modified trivially by allowing $\beta_t \in [0, 2]$ instead of $[0, 1]$, which enables the model to learn eigenvalues on the range $[-1, 1]$. To show that preconditioning adds something beyond just changing the eigenspectrum of the model, we train 340M parameter DeltaNet, GDN, and KDA models that can learn negative eigenvalues and evaluate them on the same benchmarks as our baselines and preconditioned recurrence. We find that while augmenting the baseline recurrences with negative eigenvalues improves performance relative to the baselines on the in-context retrieval language benchmarks and synthetic S-NIAH-3 benchmarks (see Table 5), it still falls short of our preconditioned recurrences (see Table 4 for full results). An interesting direction for future work would be to augment our preconditioned recurrences with negative eigenvalues and see how performance changes. Note that using the same trick of allowing $\beta_t \in [0, 2]$ does not work in our preconditioned recurrence case, as we are no longer guaranteed eigenvalues less than 1.

### E.5. Encountering training instabilities in MesaNet

We briefly note training instabilities we encounter when attempting to train a MesaNet 340M parameter model to benchmark our models against. In particular, using the same training pipeline we used for the rest of our models (described in Appendix E.1) as well as the implementation of MesaNet in `flash-linear-attention`, we were unable to train the model to completion as we encountered NaNs roughly 5K steps into training (out of 30K). We note that this training instability is also observed in Peng et al. (2025b). We additionally tried using the hyperparameters specified in the MesaNet paper to train the model, including the trick of upper-bounding the decay term discussed in Appendix F.5 of the paper (von Oswald et al., 2025). As the original code for MesaNet is not open-source, we are unable to validate their model against ours.

In any case, one area we note as an advantage of our diagonal preconditioner relative to the exact (up to conjugate gradients approximation error) least-squares preconditioner learned in MesaNet is that diagonal matrices are commutative, which promotes state-tracking abilities as discussed in Merrill et al. (2025).

## E.6. Additional plots of model internals

*Figure 15.* Distribution of learned $\tilde{\boldsymbol{k}}_t^T \boldsymbol{k}_t$ in PGDN showing that it deviates from 1, indicative of increased recurrent expressivity relative to GDN.

*Figure 16.* Learned $\alpha_t, \beta_t$ and eigenvalues in the GDN and PGDN recurrences.

# F. Deriving the chunkwise parallel forms

## F.1. Chunkwise parallel forms for the preconditioner recurrences

In this section, we derive the chunkwise parallel form for the recursion in Eq. (7), which is shown in Eq. (8). We consider a more general recurrence which includes a learnable decay, $\alpha_t$ and gain term, $\beta_t$:

$$\mathbf{A}_t = \alpha_t \mathbf{A}_{t-1} + \beta_t \boldsymbol{k}_t \odot \boldsymbol{k}_t.$$

Split the sequence into chunks of length $C$. For chunk $[i]$, let $\mathbf{K}_{[i]} \in \mathbb{R}^{C \times d_k}$ stack $\{\boldsymbol{k}_{(i-1)C+j}\}_{j=1}^C$ as rows and let $\beta_{[i]} \in \mathbb{R}^C$ stack $\{\beta_{(i-1)C+j}\}_{j=1}^C$. Define $\mathbf{K}_{[i]}^{(2)} := \mathbf{K}_{[i]} \odot \mathbf{K}_{[i]}$. We let $\mathbf{1}$ denote an all-ones vector of appropriate dimension.

Let $\mathbf{A}_{(i-1)C}$ be a known chunk boundary state at the start of chunk $i$. Then the end-of-chunk state is

$$\mathbf{A}_{iC} = \Big( \prod_{t=(i-1)C+1}^{iC} \alpha_t \Big) \mathbf{A}_{(i-1)C} \; + \sum_{t=(i-1)C+1}^{iC} \Big( \prod_{l=t+1}^{iC} \alpha_l \Big) \beta_t \, \boldsymbol{k}_t^{\odot 2}. \tag{F.1.1}$$

Let $\overrightarrow{\boldsymbol{\Phi}}_{[i]}$ be the $C \times C$ diagonal matrix with $j$-th diagonal entry

$$(\overrightarrow{\boldsymbol{\Phi}}_{[i]})_{jj} := \prod_{l=(i-1)C+j+1}^{iC} \alpha_l, \qquad j = 1, \dots, C,$$

(with the empty product equal to 1). Then Eq. (F.1.1) can be written as

$$\mathbf{A}_{iC} = \Big( \prod_{t=(i-1)C+1}^{iC} \alpha_t \Big) \mathbf{A}_{(i-1)C} \; + \; \mathbf{1}^\top \overrightarrow{\boldsymbol{\Phi}}_{[i]} \, \mathrm{diag}(\beta_{[i]}) \mathbf{K}_{[i]}^{(2)}.$$

### F.1.1. ATQ PRECONDITIONER RECURRENCE

For this recurrence we require $(\mathbf{A}_t + \mathbf{\Lambda})^{-1}\boldsymbol{q}_t$, where the inverse is performed elementwise (because we are modeling a diagonal matrix). From Eq. (F.1.1), we can form the within-chunk states as follows.

Let $\mathbf{A}_{[i]} \in \mathbb{R}^{C \times d_k}$ be the chunk-states with row $j$ given by $\mathbf{A}_{(i-1)C+j} \in \mathbb{R}^{d_k}$. Define the decayed causal mask $\mathbf{M}_{[i]} \in \mathbb{R}^{C \times C}$ by

$$\mathbf{M}_{[i]}^{j,k} = \begin{cases} \prod_{l=(i-1)C+k+1}^{(i-1)C+j} \alpha_l, & j \geq k, \\ 0, & j < k, \end{cases}$$

(where the empty product gives 1 on the diagonal). Let $\overleftarrow{\mathbf{\Phi}}_{[i]} \in \mathbb{R}^{C \times C}$ be the diagonal matrix with $j$-th diagonal entry

$$(\overleftarrow{\mathbf{\Phi}}_{[i]})_{jj} := \prod_{l=(i-1)C+1}^{(i-1)C+j} \alpha_l, \qquad j = 1, \ldots, C.$$

Then the chunk-states satisfy

$$\mathbf{A}_{[i]} = \overleftarrow{\mathbf{\Phi}}_{[i]}\mathbf{1}\,\mathbf{A}_{(i-1)C}^\top \; + \; \mathbf{M}_{[i]}\,\mathrm{diag}(\beta_{[i]})\mathbf{K}_{[i]}^{(2)}.$$

Consequently, to get $\tilde{\mathbf{Q}}_{[i]}$ we have the elementwise form

$$\tilde{\mathbf{Q}}_{[i]} = \mathbf{Q}_{[i]} \oslash \big(\mathbf{A}_{[i]} + \mathbf{\Lambda}\big),$$

where $\mathbf{\Lambda}$ is broadcast row-wise across the $C$ rows of $\mathbf{A}_{[i]}$.

### F.1.2. ATK (UNSTABLE) PRECONDITIONER RECURRENCE

Here, we show the chunkwise parallel form of the theoretically motivated ATK preconditioner recurrence derived from an application of the Sherman-Morrison identity discussed in Appendix C.1. Recall that we found this functional form difficult to train due to reasons discussed in Appendix E.3, but present it here for the sake of completeness.

For this recurrence we require

$$\frac{(\mathbf{A}_{t-1} + \mathbf{\Lambda})^{-1}\boldsymbol{k}_t}{1 + \boldsymbol{k}_t^\top (\mathbf{A}_{t-1} + \mathbf{\Lambda})^{-1}\boldsymbol{k}_t}.$$

Differing from the ATQ recurrence, we require $\mathbf{A}_{t-1}$, which means slightly shifted decay matrices. Let $\mathbf{A}_{[i]}^{\mathrm{pre}} \in \mathbb{R}^{C \times d_k}$ have row $j$ equal to $\mathbf{A}_{(i-1)C+j-1}$ (so the first row is $\mathbf{A}_{(i-1)C}$). Define the left-shifted decayed mask $\mathbf{M}_{[i]}^{<} \in \mathbb{R}^{C \times C}$ by

$$\mathbf{M}_{[i]}^{<\,j,k} = \begin{cases} \prod_{l=(i-1)C+k+1}^{(i-1)C+j-1} \alpha_l, & j > k, \\ 0, & j \leq k, \end{cases}$$

(where the empty product gives 1 on the first subdiagonal). Let $\overleftarrow{\mathbf{\Phi}}_{[i]}^{<} \in \mathbb{R}^{C \times C}$ be diagonal with $j$-th diagonal entry

$$(\overleftarrow{\mathbf{\Phi}}_{[i]}^{<})_{jj} := \prod_{l=(i-1)C+1}^{(i-1)C+j-1} \alpha_l, \qquad j = 1, \ldots, C,$$

(with $(\overleftarrow{\mathbf{\Phi}}_{[i]}^{<})_{11} = 1$). Then

$$\mathbf{A}_{[i]}^{\mathrm{pre}} = \overleftarrow{\mathbf{\Phi}}_{[i]}^{<}\mathbf{1}\,\mathbf{A}_{(i-1)C}^\top \; + \; \mathbf{M}_{[i]}^{<}\,\mathrm{diag}(\beta_{[i]})\mathbf{K}_{[i]}^{(2)}.$$

Let $\mathbf{P}_{[i]}^{\mathrm{pre}} := (\mathbf{A}_{[i]}^{\mathrm{pre}} + \mathbf{\Lambda})^{\odot(-1)}$ denote the elementwise reciprocal. Then

$$\tilde{\mathbf{K}}_{[i]} = \mathrm{diag}\Big(\big(\mathbf{1} + \big(\mathbf{P}_{[i]}^{\mathrm{pre}} \odot \mathbf{K}_{[i]}^{(2)}\big)\mathbf{1}\big)^{-1}\Big)\Big(\mathbf{P}_{[i]}^{\mathrm{pre}} \odot \mathbf{K}_{[i]}\Big),$$

where the $(-1)$ on the $C$-vector inside $\mathrm{diag}(\cdot)$ is an elementwise inverse.

F.1.3. ATK (STABLE) PRECONDITIONER RECURRENCE

Here, we show the chunkwise parallel form we employ in practice for our stable parameterization of the preconditioner given by $\mathbf{B}_t$. Recall from Appendix E.3 that we modify the ATK (unstable) form by dropping the normalization term and using the current time step $\mathbf{A}_t$ to precondition as opposed to the previous time step $\mathbf{A}_{t-1}$. As such, the chunkwise parallel form is the same as the one derived for ATQ above (no left shift), and we simply apply the elementwise squash $F$ on $\mathbf{A}_t$ to form $\mathbf{B}_t$:

$$\mathbf{B}_{[i]} := F(\mathbf{A}_{[i]}), \qquad \tilde{\mathbf{K}}_{[i]} = \mathbf{B}_{[i]} \odot \mathbf{K}_{[i]}.$$

## F.2. Chunkwise parallel forms for the preconditioned main recurrences

In this section, we derive the chunkwise parallel form for the following preconditioned recurrences: PDN, PGDN, PKDA, PLA, P-Mamba-2, and PGLA. While we refer to them here as preconditioned recurrences given the context of our work, we note that these chunkwise parallel forms hold for arbitrary key-side and query-side transformations.

F.2.1. CHUNKWISE PARALLEL FORMS FOR THE PRECONDITIONED DELTA-RULE AND ITS DECAYED VARIANTS

We note that the derivations here largely follow the one presented for the chunkwise parallel form in DeltaNet (Yang et al., 2024).

**PDN chunkwise parallel form.** Consider the PDN recurrence with a *read key* $\boldsymbol{k}_t$ and a distinct *write key* $\tilde{\boldsymbol{k}}_t$:

$$\mathbf{S}_t = \mathbf{S}_{t-1} - \beta_t(\mathbf{S}_{t-1}\boldsymbol{k}_t - \boldsymbol{v}_t)\tilde{\boldsymbol{k}}_t^\top, \qquad \boldsymbol{o}_t = \mathbf{S}_t\boldsymbol{q}_t,$$

where $\mathbf{S}_t \in \mathbb{R}^{d_v \times d_k}$, $\boldsymbol{k}_t, \tilde{\boldsymbol{k}}_t, \boldsymbol{q}_t \in \mathbb{R}^{d_k}$, $\boldsymbol{v}_t \in \mathbb{R}^{d_v}$, and $\beta_t \in \mathbb{R}$. Equivalently,

$$\mathbf{S}_t = \mathbf{S}_{t-1}\Big(\mathbf{I} - \beta_t\boldsymbol{k}_t\tilde{\boldsymbol{k}}_t^\top\Big) + \beta_t\boldsymbol{v}_t\tilde{\boldsymbol{k}}_t^\top. \tag{F.2.1}$$

**Chunking.** Split the sequence into chunks of length $C$. For chunk $[i]$, define the boundary state $\mathbf{S}_{[i]} := \mathbf{S}_{iC}$ and stack tokens (as rows)

$$\mathbf{Q}_{[i]} \in \mathbb{R}^{C \times d_k}, \quad \mathbf{K}_{[i]} \in \mathbb{R}^{C \times d_k}, \quad \tilde{\mathbf{K}}_{[i]} \in \mathbb{R}^{C \times d_k}, \quad \mathbf{V}_{[i]} \in \mathbb{R}^{C \times d_v}, \quad \mathbf{O}_{[i]} \in \mathbb{R}^{C \times d_v},$$

where row $j$ of $\mathbf{K}_{[i]}$ is $\boldsymbol{k}_{iC+j}^\top$, and row $j$ of $\tilde{\mathbf{K}}_{[i]}$ is $\tilde{\boldsymbol{k}}_{iC+j}^\top$. Let $\mathbf{M} \in \{0,1\}^{C \times C}$ be the causal (lower-triangular) mask.
**Unrolling within a chunk.** For within-chunk index $r \in \{1, \ldots, C\}$, define the per-step factors

$$\mathbf{A}_{[i],r} := \mathbf{I} - \beta_{iC+r}\,\boldsymbol{k}_{iC+r}\tilde{\boldsymbol{k}}_{iC+r}^\top, \qquad \mathbf{B}_{[i],r} := \beta_{iC+r}\,\boldsymbol{v}_{iC+r}\tilde{\boldsymbol{k}}_{iC+r}^\top.$$

Define the (within-chunk) product and contribution matrices

$$\mathbf{P}_{[i],r} := \prod_{t=1}^{r}\mathbf{A}_{[i],t} \in \mathbb{R}^{d_k \times d_k}, \qquad \mathbf{H}_{[i],r} := \sum_{t=1}^{r}\mathbf{B}_{[i],t}\left(\prod_{s=t+1}^{r}\mathbf{A}_{[i],s}\right) \in \mathbb{R}^{d_v \times d_k},$$

with the convention $\prod_{s=r+1}^{r}\mathbf{A}_{[i],s} = \mathbf{I}$. Then unrolling Eq. (F.2.1) over the $r$ tokens in chunk $[i]$ yields

$$\mathbf{S}_{iC+r} = \mathbf{S}_{[i]}\,\mathbf{P}_{[i],r} + \mathbf{H}_{[i],r}. \tag{F.2.2}$$

**Compact form of $\mathbf{P}_{[i],r}$ and $\mathbf{H}_{[i],r}$.** We show $\mathbf{P}_{[i],r}$ and $\mathbf{H}_{[i],r}$ admit rank-$r$ representations using the *write keys* $\tilde{\boldsymbol{k}}_{iC+t}$:

$$\mathbf{P}_{[i],r} = \mathbf{I} - \sum_{t=1}^{r}\boldsymbol{w}_{[i],t}\tilde{\boldsymbol{k}}_{iC+t}^\top, \qquad \mathbf{H}_{[i],r} = \sum_{t=1}^{r}\boldsymbol{u}_{[i],t}\tilde{\boldsymbol{k}}_{iC+t}^\top, \tag{F.2.3}$$

for some vectors $\boldsymbol{w}_{[i],t} \in \mathbb{R}^{d_k}$ and $\boldsymbol{u}_{[i],t} \in \mathbb{R}^{d_v}$. To obtain their recurrences, write

$$\mathbf{P}_{[i],r} = \mathbf{P}_{[i],r-1}\mathbf{A}_{[i],r} = \Big(\mathbf{I} - \sum_{t<r}\boldsymbol{w}_{[i],t}\tilde{\boldsymbol{k}}_{iC+t}^\top\Big)\Big(\mathbf{I} - \beta_{iC+r}\boldsymbol{k}_{iC+r}\tilde{\boldsymbol{k}}_{iC+r}^\top\Big),$$

and collect the coefficients of $\tilde{\boldsymbol{k}}_{iC+r}^\top$, yielding

$$\boldsymbol{w}_{[i],r} = \beta_{iC+r}\Big(\boldsymbol{k}_{iC+r} - \sum_{t<r}\boldsymbol{w}_{[i],t}\,(\tilde{\boldsymbol{k}}_{iC+t}^\top\boldsymbol{k}_{iC+r})\Big). \tag{F.2.4}$$

Similarly, using $\mathbf{H}_{[i],r} = \mathbf{H}_{[i],r-1}\mathbf{A}_{[i],r} + \mathbf{B}_{[i],r}$ and collecting $\tilde{\boldsymbol{k}}_{iC+r}^{\top}$ gives

$$\boldsymbol{u}_{[i],r} = \beta_{iC+r}\left(\boldsymbol{v}_{iC+r} - \sum_{t<r} \boldsymbol{u}_{[i],t}\,(\tilde{\boldsymbol{k}}_{iC+t}^{\top}\boldsymbol{k}_{iC+r})\right). \tag{F.2.5}$$

**Matrix form (UT transform).** Stack $\boldsymbol{w}_{[i],r}^{\top}$ and $\boldsymbol{u}_{[i],r}^{\top}$ as rows into

$$\mathbf{W}_{[i]} \in \mathbb{R}^{C \times d_k}, \qquad \mathbf{U}_{[i]} \in \mathbb{R}^{C \times d_v},$$

and let $\boldsymbol{\beta}_{[i]} \in \mathbb{R}^C$ be the vector with entries $(\boldsymbol{\beta}_{[i]})_r = \beta_{iC+r}$. Equations (F.2.4)–(F.2.5) are equivalent to the lower-triangular linear systems

$$\left(\mathbf{I} + \mathrm{tril}\big(\mathrm{diag}(\boldsymbol{\beta}_{[i]})\,\mathbf{K}_{[i]}\tilde{\mathbf{K}}_{[i]}^{\top}, -1\big)\right)\mathbf{W}_{[i]} = \mathrm{diag}(\boldsymbol{\beta}_{[i]})\,\mathbf{K}_{[i]},$$

$$\left(\mathbf{I} + \mathrm{tril}\big(\mathrm{diag}(\boldsymbol{\beta}_{[i]})\,\mathbf{K}_{[i]}\tilde{\mathbf{K}}_{[i]}^{\top}, -1\big)\right)\mathbf{U}_{[i]} = \mathrm{diag}(\boldsymbol{\beta}_{[i]})\,\mathbf{V}_{[i]}.$$

Define

$$\mathbf{T}_{[i]} := \left(\mathbf{I} + \mathrm{tril}\big(\mathrm{diag}(\boldsymbol{\beta}_{[i]})\,\mathbf{K}_{[i]}\tilde{\mathbf{K}}_{[i]}^{\top}, -1\big)\right)^{-1} \mathrm{diag}(\boldsymbol{\beta}_{[i]}), \tag{F.2.6}$$

so that

$$\mathbf{W}_{[i]} = \mathbf{T}_{[i]}\mathbf{K}_{[i]}, \qquad \mathbf{U}_{[i]} = \mathbf{T}_{[i]}\mathbf{V}_{[i]}. \tag{F.2.7}$$

**State update.** Taking $r = C$ in Eq. (F.2.2) and using Eq. (F.2.3) gives

$$\mathbf{S}_{[i+1]} = \mathbf{S}_{[i]}\left(\mathbf{I} - \mathbf{W}_{[i]}^{\top}\tilde{\mathbf{K}}_{[i]}\right) + \mathbf{U}_{[i]}^{\top}\tilde{\mathbf{K}}_{[i]} = \mathbf{S}_{[i]} + \left(\mathbf{U}_{[i]} - \mathbf{W}_{[i]}\mathbf{S}_{[i]}^{\top}\right)^{\top}\tilde{\mathbf{K}}_{[i]}.$$

**Outputs.** For token $r$ in chunk $[i]$, combining Eq. (F.2.2) with Eq. (F.2.3) yields

$$\mathbf{S}_{iC+r} = \mathbf{S}_{[i]} + \sum_{t \leq r} \left(\boldsymbol{u}_{[i],t} - \mathbf{S}_{[i]}\boldsymbol{w}_{[i],t}\right)\tilde{\boldsymbol{k}}_{iC+t}^{\top}.$$

Multiplying by $\boldsymbol{q}_{iC+r}$ and stacking over $r = 1, \ldots, C$ gives the exact chunkwise parallel form

$$\boxed{\mathbf{O}_{[i]} = \mathbf{Q}_{[i]}\mathbf{S}_{[i]}^{\top} + \left((\mathbf{Q}_{[i]}\tilde{\mathbf{K}}_{[i]}^{\top}) \odot \mathbf{M}\right)\left(\mathbf{U}_{[i]} - \mathbf{W}_{[i]}\mathbf{S}_{[i]}^{\top}\right)}$$

together with

$$\boxed{\mathbf{S}_{[i+1]} = \mathbf{S}_{[i]} + \left(\mathbf{U}_{[i]} - \mathbf{W}_{[i]}\mathbf{S}_{[i]}^{\top}\right)^{\top}\tilde{\mathbf{K}}_{[i]}, \qquad \mathbf{T}_{[i]}\ \text{given by (F.2.6)},\ \mathbf{W}_{[i]} = \mathbf{T}_{[i]}\mathbf{K}_{[i]},\ \mathbf{U}_{[i]} = \mathbf{T}_{[i]}\mathbf{V}_{[i]}.}$$

when $\tilde{\boldsymbol{k}}_t = \boldsymbol{k}_t$ (so $\tilde{\mathbf{K}}_{[i]} = \mathbf{K}_{[i]}$), this reduces to the standard DeltaNet chunkwise parallel form.

**Extensions to decayed variants.** As in Gated DeltaNet and Kimi Delta Attention, the same unrolling argument extends to (i) scalar decay and (ii) diagonal decay by inserting the corresponding decay factors into the within-chunk products.

**Scalar-decayed PDN (PGDN).** Let $\alpha_t \in (0,1)$ be a scalar decay and define, within chunk $t$ of length $C$,

$$\gamma_r[t] := \prod_{j=tC+1}^{tC+r} \alpha_j, \qquad \Gamma[t]_{ab} := \frac{\gamma_a[t]}{\gamma_b[t]}\,\mathbb{1}\{a \geq b\} \in \mathbb{R}^{C \times C}.$$

Using arrow notation,

$$\overleftarrow{\mathbf{Q}}_r[t] := \gamma_r[t]\mathbf{Q}_r[t], \qquad \overrightarrow{\mathbf{K}}_r[t] := \frac{\gamma_C[t]}{\gamma_r[t]}\tilde{\mathbf{K}}_r[t], \qquad \overrightarrow{\mathbf{S}}[t] := \gamma_C[t]\mathbf{S}[t],$$

the chunkwise parallel form becomes

$$\mathbf{S}[t+1] = \overrightarrow{\mathbf{S}}[t] + \left(\mathbf{U}[t] - \mathbf{W}[t]\mathbf{S}[t]^{\top}\right)^{\top}\overrightarrow{\mathbf{K}}[t], \qquad \mathbf{O}[t] = \overleftarrow{\mathbf{Q}}[t]\mathbf{S}[t]^{\top} + \left((\mathbf{Q}[t]\tilde{\mathbf{K}}[t]^{\top}) \odot \Gamma[t]\right)\left(\mathbf{U}[t] - \mathbf{W}[t]\mathbf{S}[t]^{\top}\right).$$

**Diagonally-decayed PDN (PKDA).** Let $\boldsymbol{\alpha}_t \in (0,1)^{d_k}$ be an elementwise decay and define cumulative products $\boldsymbol{b}_t := \prod_{j=1}^{t} \boldsymbol{\alpha}_j$ (elementwise). For chunk index $i$ and within-chunk position $r \in [1, C]$, define

$$\boldsymbol{\Lambda}_{iC+r} := \boldsymbol{b}_{iC+r} \oslash \boldsymbol{b}_{iC}, \qquad \boldsymbol{\Gamma}_{iC+r} := \boldsymbol{b}_{(i+1)C} \oslash \boldsymbol{b}_{iC+r}, \qquad \boldsymbol{\gamma}_{i+1} := \boldsymbol{b}_{(i+1)C} \oslash \boldsymbol{b}_{iC},$$

and stack $\boldsymbol{\Lambda}_{[i+1]}, \boldsymbol{\Gamma}_{[i+1]} \in \mathbb{R}^{C \times d_k}$ row-wise from $\boldsymbol{\Lambda}_{iC+r}, \boldsymbol{\Gamma}_{iC+r}$. Define

$$\mathbf{Q}_{[i+1]}^{\Lambda} := \mathbf{Q}_{[i+1]} \odot \boldsymbol{\Lambda}_{[i+1]}, \qquad \tilde{\mathbf{K}}_{[i+1]}^{\Gamma} := \tilde{\mathbf{K}}_{[i+1]} \odot \boldsymbol{\Gamma}_{[i+1]}, \qquad \bar{\tilde{\mathbf{K}}}_{[i+1]} := \tilde{\mathbf{K}}_{[i+1]} \oslash \boldsymbol{\Lambda}_{[i+1]},$$

and let $\mathbf{M} \in \{0,1\}^{C \times C}$ be the causal mask. Then

$$\mathbf{O}_{[i+1]} = \mathbf{Q}_{[i+1]}^{\Lambda} \mathbf{S}_{[i]}^{\top} + \left( \left( \mathbf{Q}_{[i+1]}^{\Lambda} \bar{\tilde{\mathbf{K}}}_{[i+1]}^{\top} \right) \odot \mathbf{M} \right) \left( \mathbf{U}_{[i+1]} - \mathbf{W}_{[i+1]} \mathbf{S}_{[i]}^{\top} \right),$$

$$\mathbf{S}_{[i+1]} = \mathbf{S}_{[i]} \operatorname{diag}(\boldsymbol{\gamma}_{i+1}) + \left( \mathbf{U}_{[i+1]} - \mathbf{W}_{[i+1]} \mathbf{S}_{[i]}^{\top} \right)^{\top} \tilde{\mathbf{K}}_{[i+1]}^{\Gamma}.$$

F.2.2. CHUNKWISE PARALLEL FORMS FOR PRECONDITIONED LINEAR ATTENTION AND ITS DECAYED VARIANTS

**PLA chunkwise parallel form.** The derivation for the chunkwise parallel form of PLA via ATQ follows exactly from the chunkwise parallel form of linear attention, but replacing the original queries with the transformed queries as follows: PLA maintains

$$\mathbf{S}_t = \mathbf{S}_{t-1} + \boldsymbol{v}_t \boldsymbol{k}_t^{\top}, \qquad \boldsymbol{o}_t = \mathbf{S}_t \tilde{\boldsymbol{q}}_t.$$

where $\tilde{\boldsymbol{q}}_t$ denotes the preconditioned query. Split the sequence into chunks of length $C$. For chunk $[i]$, define the boundary state $\mathbf{S}_{[i]} := \mathbf{S}_{iC}$ and stack tokens (as rows)

$$\tilde{\mathbf{Q}}_{[i]} \in \mathbb{R}^{C \times d_k}, \quad \mathbf{K}_{[i]} \in \mathbb{R}^{C \times d_k}, \quad \mathbf{V}_{[i]} \in \mathbb{R}^{C \times d_v}, \quad \mathbf{O}_{[i]} \in \mathbb{R}^{C \times d_v},$$

where row $j$ of $\mathbf{V}_{[i]}$ is $\boldsymbol{v}_{iC+j}^{\top}$, etc. Let $\mathbf{M} \in \{0,1\}^{C \times C}$ be the causal (lower-triangular) mask.
**State update.** Unrolling across the $C$ tokens in chunk $[i]$,

$$\mathbf{S}_{[i+1]} = \mathbf{S}_{[i]} + \sum_{j=1}^{C} \boldsymbol{v}_{iC+j} \boldsymbol{k}_{iC+j}^{\top} = \mathbf{S}_{[i]} + \mathbf{V}_{[i]}^{\top} \mathbf{K}_{[i]}.$$

**Outputs.** For token $j$ in chunk $[i]$,

$$\mathbf{S}_{iC+j} = \mathbf{S}_{[i]} + \sum_{r \le j} \boldsymbol{v}_{iC+r} \boldsymbol{k}_{iC+r}^{\top}, \qquad \boldsymbol{o}_{iC+j} = \mathbf{S}_{[i]} \tilde{\boldsymbol{q}}_{iC+j} + \sum_{r \le j} \boldsymbol{v}_{iC+r} \left( \boldsymbol{k}_{iC+r}^{\top} \tilde{\boldsymbol{q}}_{iC+j} \right).$$

Stacking over $j = 1, \ldots, C$ and using $(\tilde{\mathbf{Q}}_{[i]} \mathbf{K}_{[i]}^{\top})_{j,r} = \tilde{\boldsymbol{q}}_{iC+j}^{\top} \boldsymbol{k}_{iC+r}$ gives the exact chunkwise parallel form

$$\boxed{\mathbf{O}_{[i]} = \tilde{\mathbf{Q}}_{[i]} \mathbf{S}_{[i]}^{\top} + \left( (\tilde{\mathbf{Q}}_{[i]} \mathbf{K}_{[i]}^{\top}) \odot \mathbf{M} \right) \mathbf{V}_{[i]}} \qquad \boxed{\mathbf{S}_{[i+1]} = \mathbf{S}_{[i]} + \mathbf{V}_{[i]}^{\top} \mathbf{K}_{[i]}}.$$

**Scalar-decayed PLA (P-Mamba-2).** The scalar-decayed version of the PLA recurrence corresponds to a preconditioned version of Mamba-2 and follows readily by incorporating the scalar decay terms into the derivation (see (Yang et al., 2025) for details) as follows:
Let $\alpha_t \in (0, 1)$ be a scalar decay and define, within chunk $t$ of length $C$,

$$\gamma_r[t] := \prod_{j=tC+1}^{tC+r} \alpha_j, \qquad \Gamma[t]_{ij} := \frac{\gamma_i[t]}{\gamma_j[t]} \mathbb{1}\{i \ge j\} \in \mathbb{R}^{C \times C}.$$

Using the arrow notation (decay-to-first / decay-to-last within a chunk),

$$\overleftarrow{\tilde{\boldsymbol{q}}}_r[t] := \gamma_r[t] \tilde{\boldsymbol{q}}_r[t], \qquad \overrightarrow{\boldsymbol{k}}_r[t] := \frac{\gamma_C[t]}{\gamma_r[t]} \boldsymbol{k}_r[t], \qquad \overrightarrow{\mathbf{S}}_{[t]} := \gamma_C[t] \mathbf{S}_{[t]},$$

and defining the corresponding stacked blocks

$$
\overleftarrow{\tilde{\mathbf{Q}}}_{[t]} := \begin{bmatrix} \overleftarrow{\tilde{\boldsymbol{q}}}_1[t]^\top \\ \vdots \\ \overleftarrow{\tilde{\boldsymbol{q}}}_C[t]^\top \end{bmatrix}, \qquad \overrightarrow{\mathbf{K}}_{[t]} := \begin{bmatrix} \overrightarrow{\boldsymbol{k}}_1[t]^\top \\ \vdots \\ \overrightarrow{\boldsymbol{k}}_C[t]^\top \end{bmatrix},
$$

with $\tilde{\mathbf{Q}}_{[t]}, \mathbf{K}_{[t]}, \mathbf{V}_{[t]}, \mathbf{O}_{[t]}$ denoting the usual (unarrowed) stacked blocks, the chunkwise parallel form is

$$
\mathbf{S}_{[t+1]} = \overrightarrow{\mathbf{S}}_{[t]} + \mathbf{V}_{[t]}^\top \overrightarrow{\mathbf{K}}_{[t]}, \qquad \mathbf{O}_{[t]} = \overleftarrow{\tilde{\mathbf{Q}}}_{[t]} \mathbf{S}_{[t]}^\top + \left( (\tilde{\mathbf{Q}}_{[t]} \mathbf{K}_{[t]}^\top) \odot \Gamma[t] \right) \mathbf{V}_{[t]}.
$$

**Diagonally-decayed PLA (PGLA).** Similarly, the diagonally-decayed version of the PLA recurrence yields PGLA, which also follows readily from incorporating the diagonal decay terms into the derivation (see (Yang et al., 2023) for details) as follows:

Let $\boldsymbol{\alpha}_t \in (0,1)^{d_k}$ be an elementwise (diagonal) decay and define the cumulative products $\boldsymbol{b}_t := \prod_{j=1}^{t} \boldsymbol{\alpha}_j$ (elementwise). For chunk index $i$ and within-chunk position $j \in [1, C]$, define

$$
\boldsymbol{\Lambda}_{iC+j} := \boldsymbol{b}_{iC+j} \oslash \boldsymbol{b}_{iC}, \qquad \boldsymbol{\Gamma}_{iC+j} := \boldsymbol{b}_{(i+1)C} \oslash \boldsymbol{b}_{iC+j}, \qquad \boldsymbol{\gamma}_{i+1} := \boldsymbol{b}_{(i+1)C} \oslash \boldsymbol{b}_{iC},
$$

and stack $\boldsymbol{\Lambda}_{[i+1]}, \boldsymbol{\Gamma}_{[i+1]} \in \mathbb{R}^{C \times d_k}$ row-wise from $\boldsymbol{\Lambda}_{iC+j}, \boldsymbol{\Gamma}_{iC+j}$. Define the gate-adjusted blocks

$$
\tilde{\mathbf{Q}}_{[i+1]}^{\Lambda} := \tilde{\mathbf{Q}}_{[i+1]} \odot \boldsymbol{\Lambda}_{[i+1]}, \qquad \tilde{\mathbf{K}}_{[i+1]}^{\Gamma} := \mathbf{K}_{[i+1]} \odot \boldsymbol{\Gamma}_{[i+1]}, \qquad \bar{\mathbf{K}}_{[i+1]} := \mathbf{K}_{[i+1]} \oslash \boldsymbol{\Lambda}_{[i+1]},
$$

and let $\mathbf{M} \in \{0,1\}^{C \times C}$ be the causal mask. Then the chunkwise parallel form is

$$
\mathbf{O}_{[i+1]} = \tilde{\mathbf{Q}}_{[i+1]}^{\Lambda} \mathbf{S}_{[i]}^\top + \left( (\tilde{\mathbf{Q}}_{[i+1]}^{\Lambda} \bar{\mathbf{K}}_{[i+1]}^\top) \odot \mathbf{M} \right) \mathbf{V}_{[i+1]},
$$

$$
\mathbf{S}_{[i+1]} = \mathbf{S}_{[i]} \operatorname{diag}(\boldsymbol{\gamma}_{i+1}) + \mathbf{V}_{[i+1]}^\top \tilde{\mathbf{K}}_{[i+1]}^{\Gamma}.
$$

# G. Deriving recurrences using the preconditioned OCP framework

In this section, we show how preconditioned DeltaNet can be derived readily by changing the norm of the proximal term under the online convex programming (OCP) framework. Table 8 shows the online learning objectives of common recurrences, for convenience.

*Table 8.* Linear recurrences and corresponding online learning objectives. Table taken from (Yang et al., 2025) and extended to include preconditioner.

| Method | Online Learning Objective | Online Update |
|---|---|---|
| LA | $\|\mathbf{S}_t - \mathbf{S}_{t-1}\|_F^2 - 2\langle \mathbf{S}_t \boldsymbol{k}_t,\, \boldsymbol{v}_t \rangle$ | $\mathbf{S}_t = \mathbf{S}_{t-1} + \boldsymbol{v}_t \boldsymbol{k}_t^\top$ |
| Mamba-2 | $\|\mathbf{S}_t - \alpha_t \mathbf{S}_{t-1}\|_F^2 - 2\langle \mathbf{S}_t \boldsymbol{k}_t,\, \boldsymbol{v}_t \rangle$ | $\mathbf{S}_t = \alpha_t \mathbf{S}_{t-1} + \boldsymbol{v}_t \boldsymbol{k}_t^\top$ |
| Longhorn | $\|\mathbf{S}_t - \mathbf{S}_{t-1}\|_F^2 + \beta_t \|\mathbf{S}_t \boldsymbol{k}_t - \boldsymbol{v}_t\|^2$ | $\mathbf{S}_t = \mathbf{S}_{t-1}(I - \epsilon_t \boldsymbol{k}_t \boldsymbol{k}_t^\top) + \epsilon_t \boldsymbol{v}_t \boldsymbol{k}_t^\top, \quad \epsilon_t = \frac{\beta_t}{1+\beta_t \boldsymbol{k}_t^\top \boldsymbol{k}_t}$ |
| DeltaNet | $\|\mathbf{S}_t - \mathbf{S}_{t-1}\|_F^2 - 2\langle \mathbf{S}_t \boldsymbol{k}_t,\, \beta_t(\boldsymbol{v}_t - \mathbf{S}_{t-1}\boldsymbol{k}_t) \rangle$ | $\mathbf{S}_t = \mathbf{S}_{t-1}(I - \beta_t \boldsymbol{k}_t \boldsymbol{k}_t^\top) + \beta_t \boldsymbol{v}_t \boldsymbol{k}_t^\top$ |
| Gated DeltaNet | $\|\mathbf{S}_t - \alpha_t \mathbf{S}_{t-1}\|_F^2 - 2\langle \mathbf{S}_t \boldsymbol{k}_t,\, \beta_t(\boldsymbol{v}_t - \alpha_t \mathbf{S}_{t-1}\boldsymbol{k}_t) \rangle$ | $\mathbf{S}_t = \mathbf{S}_{t-1}\big(\alpha_t(I - \beta_t \boldsymbol{k}_t \boldsymbol{k}_t^\top)\big) + \beta_t \boldsymbol{v}_t \boldsymbol{k}_t^\top$ |
| P-Mamba-2 | $\|\mathbf{S}_t - \alpha_t \mathbf{S}_{t-1}\|_{\mathbf{P}^{-1}}^2 - 2\langle \mathbf{S}_t \boldsymbol{k}_t,\, \boldsymbol{v}_t \rangle$ | $\mathbf{S}_t = \alpha_t \mathbf{S}_{t-1} + \boldsymbol{v}_t \tilde{\boldsymbol{k}}_t^\top$ |
| P-Longhorn | $\|\mathbf{S}_t - \mathbf{S}_{t-1}\|_{\mathbf{P}^{-1}}^2 + \beta_t \|\mathbf{S}_t \boldsymbol{k}_t - \boldsymbol{v}_t\|^2$ | $\mathbf{S}_t = \mathbf{S}_{t-1}(I - \epsilon_t \boldsymbol{k}_t \tilde{\boldsymbol{k}}_t^\top) + \epsilon_t \boldsymbol{v}_t \tilde{\boldsymbol{k}}_t^\top, \quad \epsilon_t = \frac{\beta_t}{1+\beta_t \boldsymbol{k}_t^\top P \boldsymbol{k}_t}$ |
| PGDN | $\|\mathbf{S}_t - \alpha_t \mathbf{S}_{t-1}\|_{\mathbf{P}^{-1}}^2 - 2\langle \mathbf{S}_t \boldsymbol{k}_t,\, \beta_t(\boldsymbol{v}_t - \alpha_t \mathbf{S}_{t-1}\boldsymbol{k}_t) \rangle$ | $\mathbf{S}_t = \mathbf{S}_{t-1}\big(\alpha_t(I - \beta_t \boldsymbol{k}_t \tilde{\boldsymbol{k}}_t^\top)\big) + \beta_t \boldsymbol{v}_t \tilde{\boldsymbol{k}}_t^\top$ |

## G.1. Preconditioned Gated DeltaNet

Consider the modified preconditioned Gated DeltaNet learning objective:

$$
\mathcal{L}_{OCP} = \|\mathbf{S}_t - \alpha_t \mathbf{S}_{t-1}\|_{\mathbf{P}^{-1}}^2 - 2\langle \mathbf{S}_t \boldsymbol{k}_t,\, \beta_t(\boldsymbol{v}_t - \alpha_t \mathbf{S}_{t-1}\boldsymbol{k}_t) \rangle
$$

Where $\|\mathbf{X}\|_{\mathbf{A}}^2$ is taken as $\operatorname{tr}(\mathbf{X}\mathbf{A}\mathbf{X}^\top)$. We generally assume $\mathbf{P} \succ 0$.

The objective can be written as

$$
\min_{\mathbf{S}_t} \quad \operatorname{tr}\Big( (\mathbf{S}_t - \alpha_t \mathbf{S}_{t-1})\, \mathbf{P}^{-1} (\mathbf{S}_t - \alpha_t \mathbf{S}_{t-1})^\top \Big) - 2\left\langle \mathbf{S}_t \boldsymbol{k}_t,\, \beta_t(\boldsymbol{v}_t - \alpha_t \mathbf{S}_{t-1}\boldsymbol{k}_t) \right\rangle.
$$

Using the identities

$$\nabla_{\mathbf{S}_t} \operatorname{tr}\Big((\mathbf{S}_t - \alpha_t \mathbf{S}_{t-1})\,\mathbf{P}^{-1}(\mathbf{S}_t - \alpha_t \mathbf{S}_{t-1})^\top\Big) = 2(\mathbf{S}_t - \alpha_t \mathbf{S}_{t-1})\mathbf{P}^{-1},$$

and

$$\nabla_{\mathbf{S}_t}\Big(-2\,\langle \mathbf{S}_t \boldsymbol{k}_t,\ \beta_t(\boldsymbol{v}_t - \alpha_t \mathbf{S}_{t-1}\boldsymbol{k}_t)\rangle\Big) = -2\,\beta_t(\boldsymbol{v}_t - \alpha_t \mathbf{S}_{t-1}\boldsymbol{k}_t)\,\boldsymbol{k}_t^\top,$$

the first-order optimality condition gives

$$2(\mathbf{S}_t - \alpha_t \mathbf{S}_{t-1})\mathbf{P}^{-1} - 2\,\beta_t(\boldsymbol{v}_t - \alpha_t \mathbf{S}_{t-1}\boldsymbol{k}_t)\,\boldsymbol{k}_t^\top = 0.$$

Thus,

$$(\mathbf{S}_t - \alpha_t \mathbf{S}_{t-1})\mathbf{P}^{-1} = \beta_t(\boldsymbol{v}_t - \alpha_t \mathbf{S}_{t-1}\boldsymbol{k}_t)\,\boldsymbol{k}_t^\top \implies \mathbf{S}_t = \alpha_t \mathbf{S}_{t-1} + \beta_t(\boldsymbol{v}_t - \alpha_t \mathbf{S}_{t-1}\boldsymbol{k}_t)\,\boldsymbol{k}_t^\top \mathbf{P}.$$

$$\begin{aligned}
\mathbf{S}_t^\star &= \alpha_t \mathbf{S}_{t-1} + \beta_t(\boldsymbol{v}_t - \alpha_t \mathbf{S}_{t-1}\boldsymbol{k}_t)\,\boldsymbol{k}_t^\top \mathbf{P} \\
&= \alpha_t \mathbf{S}_{t-1} + \beta_t(\boldsymbol{v}_t - \alpha_t \mathbf{S}_{t-1}\boldsymbol{k}_t)\,\tilde{\boldsymbol{k}}_t^\top
\end{aligned}$$

The case for DeltaNet trivially follows by setting $\alpha_t = 1$.

## G.2. Preconditioned Longhorn

We have the modified Longhorn (Liu et al., 2024) objective:

$$\min_{\mathbf{S}_t}\ \|\mathbf{S}_t - \mathbf{S}_{t-1}\|^2_{\mathbf{P}^{-1}} + \beta_t\|\mathbf{S}_t \boldsymbol{k}_t - \boldsymbol{v}_t\|^2_2, \qquad \|\mathbf{X}\|^2_{\mathbf{P}^{-1}} := \operatorname{tr}(\mathbf{X}\,\mathbf{P}^{-1}\mathbf{X}^\top).$$

Setting the first-order optimality condition:

$$\begin{aligned}
0 &= \nabla_{\mathbf{S}_t}\Big(\operatorname{tr}\big((\mathbf{S}_t - \mathbf{S}_{t-1})\,\mathbf{P}^{-1}(\mathbf{S}_t - \mathbf{S}_{t-1})^\top\big) + \beta_t\|\mathbf{S}_t \boldsymbol{k}_t - \boldsymbol{v}_t\|^2_2\Big) \\
&= 2(\mathbf{S}_t - \mathbf{S}_{t-1})\mathbf{P}^{-1} + 2\beta_t(\mathbf{S}_t \boldsymbol{k}_t - \boldsymbol{v}_t)\boldsymbol{k}_t^\top \\
(\mathbf{S}_t - \mathbf{S}_{t-1})\mathbf{P}^{-1} &= -\beta_t(\mathbf{S}_t \boldsymbol{k}_t - \boldsymbol{v}_t)\boldsymbol{k}_t^\top \\
\mathbf{S}_t - \mathbf{S}_{t-1} &= -\beta_t(\mathbf{S}_t \boldsymbol{k}_t - \boldsymbol{v}_t)\boldsymbol{k}_t^\top \mathbf{P} \\
&= -\beta_t \mathbf{S}_t \boldsymbol{k}_t \boldsymbol{k}_t^\top \mathbf{P} + \beta_t \boldsymbol{v}_t \boldsymbol{k}_t^\top \mathbf{P} \\
\mathbf{S}_t\big(I + \beta_t \boldsymbol{k}_t \boldsymbol{k}_t^\top \mathbf{P}\big) &= \mathbf{S}_{t-1} + \beta_t \boldsymbol{v}_t \boldsymbol{k}_t^\top \mathbf{P} \\
\mathbf{S}_t^\star &= \big(\mathbf{S}_{t-1} + \beta_t \boldsymbol{v}_t \boldsymbol{k}_t^\top \mathbf{P}\big)\big(I + \beta_t \boldsymbol{k}_t \boldsymbol{k}_t^\top \mathbf{P}\big)^{-1}.
\end{aligned}$$

Applying Sherman-Morrison to the solution:

$$\begin{aligned}
\big(I + \beta_t \boldsymbol{k}_t \boldsymbol{k}_t^\top \mathbf{P}\big)^{-1} &= I - \frac{\beta_t}{1 + \beta_t\,\boldsymbol{k}_t^\top \mathbf{P}\boldsymbol{k}_t}\,\boldsymbol{k}_t \boldsymbol{k}_t^\top \mathbf{P} \\
\mathbf{S}_t^\star &= \big(\mathbf{S}_{t-1} + \beta_t \boldsymbol{v}_t \boldsymbol{k}_t^\top \mathbf{P}\big)\left(I - \frac{\beta_t}{1 + \beta_t\,\boldsymbol{k}_t^\top \mathbf{P}\boldsymbol{k}_t}\,\boldsymbol{k}_t \boldsymbol{k}_t^\top \mathbf{P}\right) \\
&= \mathbf{S}_{t-1} + \frac{\beta_t}{1 + \beta_t\,\boldsymbol{k}_t^\top \mathbf{P}\boldsymbol{k}_t}\,(\boldsymbol{v}_t - \mathbf{S}_{t-1}\boldsymbol{k}_t)\boldsymbol{k}_t^\top \mathbf{P}.\mathbf{S}_t^\star \\
&= \mathbf{S}_{t-1} + \underbrace{\frac{\beta_t}{1 + \beta_t\,\boldsymbol{k}_t^\top \mathbf{P}\boldsymbol{k}_t}}_{\text{modified gain, } \tilde{\beta}_t}(\boldsymbol{v}_t - \mathbf{S}_{t-1}\boldsymbol{k}_t)\,\underbrace{\boldsymbol{k}_t^\top \mathbf{P}}_{\tilde{\boldsymbol{k}}^\top} \\
&= \mathbf{S}_{t-1} + \tilde{\beta}_t(\boldsymbol{v}_t - \mathbf{S}_{t-1}\boldsymbol{k}_t)\tilde{\boldsymbol{k}}^\top
\end{aligned}$$

Note that in this case, the gain term contains the normalization $\frac{1}{1+\beta_t\,\boldsymbol{k}_t^\top \mathbf{P}\boldsymbol{k}_t}$. This is closer to our ATK formulation used in Eq. 5, whereas the preconditioned DeltaNet update derived from the OCP formulation in Appendix G.1 is closer to the version derived in Appendix C.3. To get exact correspondence between the OCP preconditioned GDN update and the OCP preconditioned Longhorn update, we need to use $\mathbf{P}^{-1}_{OCP,\text{Longhorn}} = \mathbf{G}_{t-1}$ and $\mathbf{P}^{-1}_{OCP,\text{GDN}} = \mathbf{G}_t$.

### G.3. Key-preconditioned Mamba-2

In this case, we consider **key-preconditioned** Mamba-2, found by modifying the proximal term of Mamba-2 to include the $\mathbf{P}^{-1}$ norm. As discussed in Section 5, the preconditioned OCP framework *does not properly expose the solve axis*, so we cannot recover the ATQ/ATK correspondence between query-preconditioned linear attention and key-preconditioned DeltaNet. Therefore, the recurrence exposed in this section *is distinct* from the P-Mamba-2 recurrence discussed in the previous section, which is query-preconditioned.

Consider the POCP given by:

$$\min_{\mathbf{S}_t} \ \|\mathbf{S}_t - \alpha_t \mathbf{S}_{t-1}\|_{\mathbf{P}^{-1}}^2 \ - \ 2\langle \mathbf{S}_t \boldsymbol{k}_t, \ \boldsymbol{v}_t\rangle, \qquad \|\mathbf{X}\|_{\mathbf{P}^{-1}}^2 := \mathrm{tr}\big(\mathbf{X}\,\mathbf{P}^{-1}\mathbf{X}^\top\big).$$

Setting the first-order optimality condition gives us

$$
\begin{aligned}
0 &= \nabla_{\mathbf{S}_t}\Big(\mathrm{tr}\big((\mathbf{S}_t - \alpha_t \mathbf{S}_{t-1})\,\mathbf{P}^{-1}(\mathbf{S}_t - \alpha_t \mathbf{S}_{t-1})^\top\big) - 2\langle \mathbf{S}_t \boldsymbol{k}_t, \boldsymbol{v}_t\rangle\Big) \\
&= 2(\mathbf{S}_t - \alpha_t \mathbf{S}_{t-1})\mathbf{P}^{-1} - 2\,\boldsymbol{v}_t \boldsymbol{k}_t^\top \\
(\mathbf{S}_t - \alpha_t \mathbf{S}_{t-1})\mathbf{P}^{-1} &= \boldsymbol{v}_t \boldsymbol{k}_t^\top \\
\mathbf{S}_t - \alpha_t \mathbf{S}_{t-1} &= \boldsymbol{v}_t \boldsymbol{k}_t^\top \mathbf{P}.
\end{aligned}
$$

and solving yields

$$
\begin{aligned}
\mathbf{S}_t^\star &= \alpha_t \mathbf{S}_{t-1} + \boldsymbol{v}_t \boldsymbol{k}_t^\top \mathbf{P} \\
&= \alpha_t \mathbf{S}_{t-1} + \boldsymbol{v}_t \tilde{\boldsymbol{k}}_t^\top
\end{aligned}
$$

The non-decayed case, standard linear attention with key-side preconditioning, follows directly.

