# OpenReview forum: "Preconditioned DeltaNet: Curvature-aware Sequence Modeling for Linear Recurrences"
_ICML.cc/2026/Conference — ICML 2026 regular_

### Official Review · Reviewer_FE9C · 2026-03-11

**Soundness:** 3
**Presentation:** 4
**Significance:** 3
**Originality:** 3
**Overall Recommendation:** 5
**Confidence:** 4

**Summary:**

The paper starts from the test-time regression view of linear attention variants, and note that DeltaNet (DN) and Linear Attention (LA) approximately solve the least square problem:  DN uses online sgd, while LN uses single step offline full batch gradient descent, both ignoring the curvature information (the key Gram matrix). Next, authors show by incorporating preconditioning in gradient descent, both DN and LN recover the exact least-square solutions. In order to allow chunk-wise parallel forms, the paper adopts diagonal preconditioning, similar to popular SGD optimizers. They train preconditioned versions of gated DN and Kimi Delta attention on language, and show improved downstream performance on common-sense reasoning and retrieval benchamarks, while incurring 20% cost in total throughput. Based on their analysis, they also propose a taxonomy of linear attention models based on the choice of decay, gain, preconditioner, and solve.

**Compliance With Llm Reviewing Policy:**

Affirmed.

**Final Justification:**

I recommend acceptance. The paper is very well written, technically strong, and supported by thorough empirical and implementation details. My main concern about the motivation for diagonal preconditioning was valid, and the authors fully acknowledged the issue in the rebuttal, provided more appropriate metrics, and committed to correcting this.

**Key Questions For Authors:**

1. Is it possible to provide comparison (or any insight) using the reported numbers from the MesaNet paper? How do the methods compare in terms of throughput?
2. Based on point 2 from the section above, could you demonstrate a sound measure of effectiveness of diagonal preconditioning?
3. in terms of computational complexity, applying ATQ in PLA seems lighter than ATK of PDN. Can you comment on this?
4. Can the authors provide their best guess as to why KDA achieves the best performance at larger scale, in retrieval?

**Limitations:**

Yes

**Strengths And Weaknesses:**

Strength:

The paper is exceptionally well-written and well-organized. The uthors managed to maintain a good balance between mathematical rigor and readability. The paper is very detailed yet a familiar reader can follow the arguments, as they lay the necessary groundwork for the methods. Empirical studies are thorough, as they match the training setup of prior work and compare downstream performance on popular base language models benchmarks. The implementation extends the flash-linear-attention triton codebase, and the author provide extensive discussion on practical implementation details.

Weakness:

1. The main weakness of the empirical evaluation is a lack of comparison with MesaNet, which is the direct contender to the proposed approach. Both methods are motivated by solving the exact online least square problem. The narrative of the paper discards MesaNet because of “numerical instabilities”, while the authors of MesaNet propose measures to avoid instabilities and similarly train a 1B MesaNet on 50B tokens.
2. I think the motivation on diagonalizing the preconditioner is inaccurate at some points:
    1. For example, from Fig.1 right, the fact that effective rank of $H=KK^\top$ is similar to that of $\text{diag}(H)$ does _not_ support the effectiveness of a diagonal preconditioner, as rank does not take into account the feature cross-correlation (i.e. rotation of the eigenbasis), which is exactly what a diagonal preconditioner cannot inverse. For example, given any rotation matrix, eigenvalues of $A$ and $R^\top A R$ are the same (and hence the effective ranks), while if a diagonal preconditioner can make $A$ isotropic, it is guaranteed not to do so for the rotated one.
    2. This sentence is also not accurate, and perhaps incorrect: “From the perspective of learning, a diagonal approximation is justified given the presence of anisotropy in pretrained models.“ anisotropy does not justify the effectiveness of “diagonal” preconditioner. This only makes sense when the eigenbasis is axis-aligned.
    3. Therefore, to reasonably measure the effectiveness of diagonal preconditioning, one should measure the level of diagonal dominance of H, or how axis-aligned the eigenbasis is. For example, you may plot the normalized norm of $||H - diag(H)||$, or the $L_\infty$ norm of (normalized) eigenvectors.
    4. I think the computational cost of full preconditioning, combined with the prevalent use of diagonal conditioning in popular optimizers is a strong enough justification on its own, which suggests there remains some gap that better preconditioning methods could fill.
3. minor issues:
    1. The ridge term in eq. 1 is missing.
    2. below Eq. 4: “This shows that preconditioning this step with the key Gram inverse recovers the exact least squares solution.” however, this is actually later shown in Theorem 3.1., which makes the text confusing.
    3. some notations are not introduced, e.g. ⊘
    4. line 270: “Unlike the Gated DeltaNet recurrence, it has no decay or write term.” why is a decay term required for A? Based on the mathematical framework provided, A should match diag of K^\top K.

---

> ### Author Rebuttal · Authors · 2026-03-31
>
> We thank the reviewer for their extremely thorough review and positive comments regarding our work! We respond to each of the points raised below:
>
> **(1) Lack of comparison to MesaNet**
>
> We agree that our work would benefit from a comparison to MesaNet, however we are unable to replicate the results given their implementation is not open-sourced. The only implementation (to the extent of our knowledge) is the one in flash-linear-attention (FLA) [here](https://github.com/fla-org/flash-linear-attention/blob/main/fla/layers/mesa_net.py). We attempted to train a 340M MesaNet model using this implementation, but encountered NaNs after roughly 5K steps of training. This is despite our best efforts to resolve this instability using the tricks discussed in Section F.5 of the MesaNet paper. Note that this training instability in the MesaNet layer implemented in FLA was also observed in [1].
>
> However, we are able to benchmark our preconditioned recurrences against MesaNet regarding throughput, the results for which can be found [here](https://ibb.co/k2PxF1p4). Since our initial submission, we have optimized the preconditioner recurrence sub-kernel that is invoked in both the PGDN and PKDA kernels, reducing the overhead relative to GDN and KDA from 20% to 13%. As such, we find that PGDN has higher throughput than MesaNet, while KDA is marginally slower. However, we note that a proper comparison between MesaNet and KDA would require MesaNet to use a diagonal decay $\alpha_t$ as opposed to its scalar decay which enables a fair comparison between PGDN and MesaNet out-of-the-box.
>
> **(2) Inaccurate motivation of diagonal preconditioner**
>
> This is an excellent point raised by the reviewer and we agree that the motivation of the diagonal preconditioner in our submission is inaccurate, particularly regarding the effective rank metric as pointed our by the reviewer's rotation counterexample. As suggested, we have computed the normalized norm and axis-alignment metrics which can be found [here](https://ibb.co/CKpgyPGv). We find that the off-diagonal coupling is certainly non-trivial, particularly in the inverse Gram, indicating the utility of exploring preconditioners that capture the off-diagonal coupling in a structured/efficient manner. We will be sure to remove the effective rank plot from the paper and add these more robust metrics. Additionally, we will point out the fact that there is still ground to be gained with non-diagonal preconditioners which constitutes an interesting direction of future work.
>
> **(3) Computational complexity of ATQ vs ATK**
>
> Yes, we agree that ATQ in PLA is presumably lighter than ATK in PDN since the base kernel for linear attention is more efficient than the base kernel for DeltaNet. We do not have optimized kernels for PLA as we implemented it using the general DPLR kernels and found that they perform worse than PDN (see Appendix E.3 for these results). This, along with Theorem 3.2, justifies why we chose to pursue PDN-style recurrences in this work.
>
> More generally, this is a quality-efficiency tradeoff as we know that DeltaNet models and its decayed variants (e.g. GDN and KDA) tend to outperform linear attention and its decayed variants (e.g. Mamba-2 and GLA).  Understanding how preconditioning interacts with each of these base recurrences and the Pareto frontier for recurrence design is certainly an interesting direction for future work.
>
> **(4) On the performance of KDA at the 1B scale**
>
> We assume the reviewer is referring to KDA outperforming PKDA at the 1B scale on retrieval in Table 1.
>
> We first note that since our initial submission, we have also extended our evaluation set. In particular, we have added evaluations on BoolQ and SciQ for commonsense reasoning, TQA and DROP for in-context retrieval (see results [here](https://ibb.co/1fWL8VTR)) and extended the RULER evaluations to 4K and 8K sequence lengths (see results [here](https://ibb.co/HLTnQsz7)). Taking these evaluations into account, we actually find that PKDA outperforms KDA in retrieval at the 1B scale. More generally, all additional evaluations that we have collected support the use of preconditioning in the recurrence families we have explored.
>
> However, for some intuition as to why preconditioning in KDA yields less pronounced improvements than preconditioning in the other recurrences, we refer the reviewer to our response to reviewer ifPA who posed a similar question. In particular, since KDA has a diagonal decay term, it implicitly models a diagonal transform on its write key; this is in contrast to DN and GDN which do not. Still, since this term in KDA is intended to model decay, it is less performant than having an explicit diagonal preconditioner as we show.
>
> **(5) Other improvements to paper**
>
> We have also fixed the minor issues pointed out by the reviewer. Thanks for finding these!
>
> **References**
>
> [1] Gated KalmaNet: A Fading Memory Layer Through Test-Time Ridge Regression (https://arxiv.org/pdf/2511.21016)

---

> > ### Author Rebuttal · Reviewer_FE9C · 2026-03-31
> >
> > Thank you for your detailed response and providing the new axis-alignment metrics. I have already recommended acceptance and will keep my score as is.

---

> > > ### Author Response · Authors · 2026-04-05
> > >
> > > We thank the reviewer for going over our rebuttal and maintaining their positive review of our work!

---

### Official Review · Reviewer_4aVK · 2026-03-12

**Soundness:** 3
**Presentation:** 3
**Significance:** 3
**Originality:** 3
**Overall Recommendation:** 4
**Confidence:** 4

**Summary:**

The authors show that linear attention and DeltaNet become equivalent under exact inverse-Gram preconditioning and derive practical approximations using a diagonal Gram estimate. Based on this insight, the paper introduces preconditioned variants of DeltaNet, GDN, and KDA, along with efficient chunkwise-parallel implementations. Experiments show improvements on associative recall benchmarks and some language modeling tasks.

**Compliance With Llm Reviewing Policy:**

Affirmed.

**Key Questions For Authors:**

Please refer to Weaknesses.

**Limitations:**

yes

**Strengths And Weaknesses:**

Strengths:

1. The paper provides an elegant interpretation of linear attention and DeltaNet as approximate solvers of an online least-squares problem. The equivalence between preconditioned linear attention and preconditioned DeltaNet offers a useful conceptual unification.
2. The proposed DGPS taxonomy highlights preconditioning alongside decay, gain, and solve strategies, which helps organize existing recurrence designs and may guide future architectures.
3. The authors derive chunkwise-parallel forms compatible with existing efficient kernels and provide Triton implementations that maintain scalability with only moderate overhead.
4. The proposed methods show consistent gains on associative recall and needle-in-a-haystack style benchmarks, suggesting that preconditioning can improve memory dynamics in linear recurrent models.

Weaknesses:
1. Theoretical formula is elegant, but the final implementation deviates significantly from the accurate preprocessing formula due to several engineering modifications (e.g., diagonal approximation, learning parameters, and compression in Equation (8)). It remains unclear whether the performance improvement stems from curvature-aware preprocessing or simply from more flexible dimensionality-wise write reweighting. Further ablation experiments will help clarify this, e.g. disabling the learned $\alpha_t^P$, $\beta_t^P$, using a fixed $B_t$ versus learned $B_t$.
2. Theorem 3.2 relies on some strong assumptions (e.g., keys/queries i.i.d, uniformly on the unit sphere), which are unlikely to hold true in representations learned in real LLMs. Therefore, its impact on the dynamics of real LLMs remains unclear. Empirical verification of these assumptions or their corollaries will help strengthen the argument.
3. This paper demonstrates that diagonal Gram matrices and full Gram matrices possess similar statistical properties (e.g., spectrum shape), but this does not necessarily imply that their inverse geometries are similar. Since the reliance on the inverse Gram matrix facilitates a more direct analysis of the approximate quality, this is helpful.
4. In Table 1, the improvements across different recursion families are inconsistent (e.g., KDA/PKDA shows unstable improvements in large-scale scenarios). Furthermore, this method introduces approximately 20% training overhead, which is not negligible for long-context models. In addition, the paper lacks a more comprehensive end-to-end latency or deployment analysis.

---

> ### Author Rebuttal · Authors · 2026-03-30
>
> We thank the reviewer for their thorough feedback and positive review of our paper! We respond to each point raised below:
>
> **(1) Ablations on functional form of preconditioner recurrence**
>
> We first note that we have conducted several ablations to justify the functional form of the precondtioner which can be found in Appendix E.3 of the paper.
>
> However, on the point of whether or not the "performance improvement stems from curvature-aware preprocessing or simply from more flexible dimensionality-wise write reweighting", we agree that we do not address this direction. In fact, as discussed in Section 4.1 of the paper, we propose this as a direction for future work. The reason we do not include this in the scope of this paper is that we maintain it constitutes a different recurrence. The purpose of our work and the recurrences we analyze is to understand how we can leverage the existing parameterizations of linear recurrences and improve performance via preconditioning (which adds effectively no parameters). It is entirely possible that learning the "preconditioner" directly by parameterizing $B_t$ (analogous to how KDA parameterizes $\alpha_t$) would improve performance. However, this comes at the cost of adding more parameters to the recurrence. Moreover, note that the functional form of the preconditioner recurrence as proposed in Equation (8) maintains the monotonicity induced by the exact curvature recurrence $$A_t = A_{t-1} + k_t \odot k_t$$ which at minimum demonstrates that we do get performance improvements from curvature-aware preprocessing.
>
> Regarding the other proposed ablation of "disabling the learned $\alpha_t^P, \beta_t^P$", is the reviewer proposing that we fix these as $1$ or something else? We are happy to run this ablation given some clarification.
>
> **(2) Theorem 3.2 relies on strong assumptions of isotropy**
>
> We agree with the reviewer that these assumptions are strong and do not hold in practice. In fact, if they did, then we would not need preconditioning as the key Gram $K^T K$ would reduce to the identity.  The purpose of this theorem is simply to offer some justification as to why we chose to explore preconditioning applied to DeltaNet-style recurrences as opposed to linear attention ones. However, exploring preconditioned linear attention models is certainly a viable direction for future work which we discuss in detail in Appendices B and G. We will make sure to revise the paper to clarify the scope of the theorem.
>
> **(3) Inverse Gram geometry differs from Gram geometry**
>
> We entirely agree with this point and will revise the paper to include analyses of the inverse Gram which can be found [here](https://ibb.co/CKpgyPGv). In particular, we find that the inverse Gram and diagonal inverse Gram eignespectra are highly correlated, while the effective rank is less so. However, we also agree with the point raised by reviewer FE9C on the effective rank of the Gram being a poor metric for justifying the diagonal preconditioner. As suggested by reviewer FE9C, we plan to replace the effective rank plot with the plot of the eigenvalue-weighted $\ell_{\infty}$ norm of the key Gram/inverse key Gram eigenvectors to demonstrate the degree to which the matrix is axis-aligned (see [here](https://ibb.co/CKpgyPGv)).
>
> **(4) Unstable improvements in evaluation metrics**
>
> Since the paper's initial submission, we have extended our evaluation set. In particular, we have added evaluations on BoolQ and SciQ for commonsense reasoning, TQA and DROP for in-context retrieval (see results [here](https://ibb.co/1fWL8VTR)) and extended the RULER evaluations to 4K and 8K sequence lengths (see results [here](https://ibb.co/HLTnQsz7)). Taking these evaluations into account, we see more stable/clear improvements when using preconditioning in the KDA recurrence. More generally, all additional evaluations that we have collected support the use of preconditioning in the recurrence families we have explored.
>
> **(5) Overhead from preconditioning and lack of more thorough profiling**
>
> Firstly, we note that since our initial submission, we have optimized the PGDN and PKDA kernels by improving the preconditioner recurrence subkernel via tiling along the key dimension and doing recomputation of the preconditioner states in the backward pass. This has reduced the overhead relative to GDN and PKDA from 20% to 13%. See the profiling results [here](https://ibb.co/k2PxF1p4).
>
> Secondly, we provide a more thorough analysis of memory, throughput and latency of the recurrences we explore in this work which can be found [here](https://ibb.co/GfbjKQ2q). We thank the reviewer for raising this point and will incorporate this into the final version of the paper.

---

> > ### Author Rebuttal · Reviewer_4aVK · 2026-04-03
> >
> > Thank the authors for the rebuttal. My concerns have been resolved. I'll keep my positive review of this paper.

---

> > > ### Author Response · Authors · 2026-04-05
> > >
> > > We thank the reviewer for going over our rebuttal and maintaining their positive review of our work!

---

### Official Review · Reviewer_ifPA · 2026-03-13

**Soundness:** 3
**Presentation:** 3
**Significance:** 3
**Originality:** 3
**Overall Recommendation:** 4
**Confidence:** 2

**Summary:**

The paper that under certain formulation the preconditioned linear attention and the preconditioned DeltaNet are equivalent. Beyond that, the paper also adopts certain optimization for better training efficiency and effect. Emprical stuides are conducted on 340M and 1B LLMs to show the effectiveness of the precondition terms.

**Compliance With Llm Reviewing Policy:**

Affirmed.

**Final Justification:**

The paper is overall solid and well-motivated by the inverse-Gram formulation. During the rebuttal the authors promise to make the theory to practice transition more explicit and add a clarification that "richer preconditioners capturing structured off-diagonal correlations are an important direction for future work". Also for questions regarding missing baselines and intuitions of why preconditioning works better on DN than KDA models, the authors provide satisfactory answers.

**Key Questions For Authors:**

1. It seems that preconditioning works better on DN models than DA models, are there any explanations for that?
2. Why does the comparison between DN and PDN disappears for the 1B models?

**Limitations:**

yes

**Strengths And Weaknesses:**

Strengths:
1. The theoretical intuition behind introducing the precondition terms is solid and well-motivated.
2. Optimization efforts for efficient training and practical concerns seem to improve the practicality of the proposed method.
3. Experimental results indeed show some gains for models with precondition terms.

Weaknesses:
1. The theory utilizes the exact inverse-Gram case, but the practical model uses a diagonal approximation plus additional trainability modifications, which introduces approximation.
2. In table 1 the training set size seems insufficient for pretraining of LLMs (even for LLMs as small as 340m, 15B tokens are not enough).

---

> ### Author Rebuttal · Authors · 2026-03-30
>
> We thank the reviewer for their thoughtful comments and positive outlook on our work! We address each of their points below:
>
> **(1) Exact inverse-Gram vs. practical diagonal approximation**
>
> We agree that the practical models introduce approximation relative to the exact inverse-Gram formulation analyzed in the theory. Our intent in presenting the exact case is to expose the preconditioning axis in linear recurrences and show that, under exact preconditioning, preconditioned linear attention and preconditioned DeltaNet are equivalent realizations of the same operator.
>
> The practical models then instantiate this axis using a diagonal preconditioner, which is the simplest and most efficient approximation. Diagonal preconditioning is a standard approximation in large-scale optimization, used in methods such as Adam, Adagrad, and RMSProp. In our setting, it is particularly attractive because it preserves chunkwise parallelism, numerical stability, and hardware efficiency, while still capturing useful coordinate-wise anisotropy of the key Gram as discussed in Section 3 of the paper.
>
> We will revise the paper to make this theory-to-practice transition more explicit, and to clarify that richer preconditioners capturing structured off-diagonal correlations are an important direction for future work.
>
> **(2) Training token budget / model scale**
>
> We agree that larger scale training is important. However, our goal in Table 1 is to show that preconditioning improves matched linear recurrent architectures under a controlled and standard experimental budget. In this sense, all methods are compared fairly at the same model size and token budget. More broadly, the scales considered here are consistent with prior work on novel linear recurrent architectures, where evaluation at 340M parameters trained on 15B tokens is canonical (e.g. DeltaNet, Gated DeltaNet, Comba, GLA).
>
> **(3) Preconditioning works better on DN models than DA models**
>
> We assume by “DA models” the reviewer is referring to KDA. Our intuition for this observation is that DeltaNet (DN) models do not include any built-in diagonal reweighting of the write key, whereas KDA already does so implicitly through its diagonal decay. Using $S_t \in \mathbb{R}^{d_v \times d_k}$, we can write a DN-style recurrence with distinct read and write keys as
>
> $$
> S_t = S_{t-1} + \beta_t \bigl(v_t - S_{t-1} k_t\bigr)\tilde{k}_t^\top .
> $$
>
> In prior work such as DN, GDN, and KDA, the read and write keys are typically tied, i.e. $\tilde{k}_t = k_t$. In our formulation, we instead consider a preconditioned write key of the form
>
> $$
> \tilde{k}_t = P_t k_t ,
> $$
>
> where $P_t$ is the preconditioner. When $P_t$ is diagonal, this explicitly rescales the key coordinates before writing into the state.
>
> By contrast, KDA augments the base DeltaNet recurrence with a diagonal decay term:
> $$
> S_t = \alpha_t S_{t-1} + \beta_t \bigl(v_t - \alpha_t  S_{t-1} k_t\bigr)\tilde{k}_t^\top
> $$
>
> If we rewrite this as
> $$
> S_t = \alpha_t  S_{t-1} + \beta_t \bigl(v_t - S_{t-1} k_t\bigr)(\alpha_t \tilde{k}_t)^\top
> $$
>
> we see that the decay induces an “effective” write key $(\alpha_t \tilde{k}_t)$. Thus, KDA already implicitly applies a diagonal coordinatewise rescaling to the write key, partially capturing the role of a diagonal preconditioner.
>
> However, in KDA this rescaling is tied to the recurrence dynamics through the decay, rather than being a separate preconditioning mechanism. This is precisely why adding an explicit diagonal preconditioner can still help KDA. At the same time, because vanilla DeltaNet lacks any analogous built-in diagonal key reweighting, it is natural that explicit preconditioning has a larger effect there.
>
> **(4) Lack of comparison between DN and PDN at 1B scale**
>
> We do not scale the DeltaNet (DN) model to 1B as they have been shown in prior work (e.g. Gated DeltaNet paper) to perform signicantly worse than their gated variants such as GDN and KDA due to the lack of a decay term in the recurrence. We also observe this in our results at the 340M scale.
>
> **(5) General improvements to the results**
>
> We additionally note some improvements we have made to the paper since its initial submission.
>
> Firstly, we have optimized the PGDN and PKDA kernels by improving the preconditioner recurrence subkernel via tiling along the key dimension and doing recomputation of the preconditioner states in the backward pass. This has reduced the overhead relative to GDN and PKDA from 20% to 13%. See the profiling results [here](https://ibb.co/k2PxF1p4).
>
> Secondly, we have extended our evaluation set. In particular, we have added evaluations on BoolQ and SciQ for commonsense reasoning, TQA and DROP for in-context retrieval (see results [here](https://ibb.co/1fWL8VTR)) and extended the RULER evaluations to 4K and 8K sequence lengths (see results [here](https://ibb.co/HLTnQsz7)). All additional evaluations further demonstrate that the preconditioned recurrences outperform their unpreconditioned counterparts.

---

> > ### Author Rebuttal · Reviewer_ifPA · 2026-04-03
> >
> > I thank the authors for the rebuttal, and I suggest clarifying weakness1 and question2 in the paper. I'll keep my positive review of this paper.

---

> > > ### Author Response · Authors · 2026-04-05
> > >
> > > We thank the reviewer for going over our rebuttal and maintaining their positive review of our work! We will be sure to clarify the points raised in the final version of the paper.

---

### Official Review · Reviewer_5KNf · 2026-03-14

**Soundness:** 2
**Presentation:** 3
**Significance:** 2
**Originality:** 2
**Overall Recommendation:** 3
**Confidence:** 3

**Summary:**

This paper introduces preconditioning to linear recurrences to account for the curvature of the least-squares objective in the test-time regression framework. The authors demonstrate that while exact preconditioning unifies linear attention and DeltaNet, a diagonal approximation allows for efficient, numerically stable, and parallelizable implementations. Empirically, the proposed preconditioned models (PDN, PGDN, PKDA) outperform their base versions in language modeling and long-context retrieval tasks at the 340M and 1B parameter scales.

**Compliance With Llm Reviewing Policy:**

Affirmed.

**Key Questions For Authors:**

How does the memory and latency overhead of maintaining the diagonal state $A_t$ scale as the head dimension ($d_k$) increases, particularly in comparison to the general DPLR kernels?

**Limitations:**

yes.

**Strengths And Weaknesses:**

Strengths:

1. The work provides a rigorous derivation showing that exactly preconditioned linear attention and DeltaNet are equivalent realizations of the same operator.

2. By utilizing a diagonal preconditioner, the authors develop chunkwise parallel algorithms that maintain high hardware efficiency and numerical stability compared to prior exact methods.

Weaknesses

1. While diagonal preconditioning is efficient, it only captures coordinate-wise anisotropy and ignores off-diagonal correlations in the key Gram matrix that full preconditioning would address.

2. Although the results are consistent, the experiments are conducted at relatively small scales (up to 1B parameters and 50B tokens), leaving the benefits of preconditioning at much larger scales unverified.

---

> ### Author Rebuttal · Authors · 2026-03-30
>
> We thank the reviewer for their feedback. Below we address each concern.
>
> **(1) Diagonal preconditioning ignores off-diagonal correlations**
>
> We agree that a diagonal approximation ignores off-diagonal correlations that full preconditioning would capture. However, diagonal preconditioning is a standard and well-motivated approximation in large-scale optimization, used in methods such as Adam, Adagrad, and RMSProp, where it provides a favorable tradeoff between quality and efficiency.
>
> In our setting, the diagonal approximation is particularly attractive because it preserves the key properties required for efficient chunkwise parallel implementations, while still capturing per-coordinate anisotropy of the key Gram. This enables a numerically stable and hardware-efficient realization that would not be possible with full preconditioning, as discussed in Section 3 of the paper.
>
> We will clarify this point in the paper and emphasize that our goal is to expose the preconditioning axis in linear recurrence design. The diagonal case represents the simplest and most practical point in this space, and our results suggest it already yields consistent gains. Exploring richer preconditioners that capture structured off-diagonal correlations is a promising direction for future work.
>
> **(2) Limited scale (≤1B parameters)**
>
> We agree that scaling to larger models is important. However, the scale considered in this work (up to 1B parameters and 50B tokens) is consistent with prior work. For example, MesaNet is evaluated in exactly this regime. Other works such as Gated DeltaNet and Comba scale to 1.3B parameters trained on 100B tokens.
>
> Nonetheless, to address the reviewer’s concern regarding scale, we are currently training 1.3B models trained on 100B tokens and will include these results in the final version of the paper. More broadly, scaling beyond this range is still relatively uncommon for new sequence mixer architectures due to the significant computational cost of training from scratch, and we believe our evaluation is well-aligned with current practice.
>
> **(3) Memory/latency overhead of diagonal state**
>
> We conduct a detailed profiling of memory and throughput speed as a function of head dimension for GDN, KDA, PGDN, PKDA, DPLR-GDN and DPLR-KDA which can be found [here](https://ibb.co/GfbjKQ2q).
>
> The additional state introduced by our diagonal preconditioner scales linearly with the head dimension, since it stores and updates only one scalar per coordinate. As such, the memory and speed overhead relative to the unpreconditioned base recurrence grows roughly linearly in the state. However, since the state of the base recurrence is $O(d^2)$, the total state (and thus time and memory requirements of the kernel) which is $O(d^2) + O(d)$ is dominated by the quadratic term.
>
> We observe that both PGDN and PKDA have significantly better memory and speed profiling relative to their naive DPLR counterparts, indicating the utility of the Triton kernels we wrote for our preconditioned recurrences.
>
> **(4) General improvements to the results**
>
> We additionally note some general improvements we have made to the paper since its initial submission.
>
> Firstly, we have optimized the PGDN and PKDA kernels by improving the preconditioner recurrence subkernel via tiling along the key dimension and doing recomputation of the preconditioner states in the backward pass. This has reduced the overhead relative to GDN and PKDA from 20% to 13%. See the new profiling results [here](https://ibb.co/k2PxF1p4).
>
> Secondly, we have extended our evaluation set. In particular, we have added evaluations on BoolQ and SciQ for commonsense reasoning, TQA and DROP for in-context retrieval (see results [here](https://ibb.co/1fWL8VTR)) and extended the RULER evaluations to 4K and 8K sequence lengths (see results [here](https://ibb.co/HLTnQsz7)). All additional evaluations further demonstrate that the preconditioned recurrences outperform their unpreconditioned counterparts.

---

### Decision · Program_Chairs · 2026-04-30

**Decision:**

Accept (regular)

**Comment:**

This paper introduces preconditioning into delta-rule linear recurrences. From the test-time regression view, exact inverse-Gram preconditioning makes linear attention and DeltaNet equivalent. The authors instantiate this with a practical diagonal approximation and provide efficient Triton kernels for preconditioned DeltaNet, GDN, and KDA, along with a DGPS (decay / gain / preconditioner / solve) taxonomy of recurrence designs. The preconditioned variants give consistent gains on synthetic recall, language modeling, and long-context retrieval at 340M and 1B scales.

Reviewers consistently praised the clarity and rigor of the writing, the elegance of the unifying perspective (preconditioning as a missing axis in delta-rule recurrences), the DGPS taxonomy as a useful organizing tool, and the careful Triton implementation extending the FLA codebase. The author response added throughput comparisons against MesaNet, expanded evaluations (BoolQ, SciQ, TQA, DROP, RULER at 4K/8K), more rigorous diagnostics for justifying the diagonal approximation (replacing a misleading effective-rank plot with axis-alignment metrics), and a clear explanation of why preconditioning helps DeltaNet/GDN more than KDA (KDA's diagonal decay already implicitly applies coordinate-wise rescaling to the write key).

Limitations remain: the diagonal approximation deviates from the exact theory and richer preconditioners are left to future work, and KDA gains at 1B are smaller than for DeltaNet / GDN. These do not undermine the contribution. The paper exposes a clean and previously underexplored design axis, gives a hardware-efficient instantiation, and provides convincing empirical support across multiple recurrence families.

Overall, I recommend acceptance.